# REGRET ANALYSIS OF RMSPROP AND ADAMNC FOR TRAINING DEEP INTERPOLATING NEURAL NETWORKS

## ABSTRACT

We provide a theoretical analysis for RMSProp and AdamNC (Adam without corrective terms) for training deep fully-connected neural networks with smooth activations, in the online learning setting. We focus on the binary classification tasks with logistic loss or exponential loss. We assume that the model can interpolate data, i.e., it can obtain an arbitrarily small loss $\varepsilon$, while the distance to the initialization is bounded by a decreasing function $g(\varepsilon)$. We show that the regret is upper bounded by $\mathcal{O}(\text{poly}[g(1/T)])$ provided that the width is at least $\mathcal{O}(\text{poly}[g(1/T)])$, where $T$ is the total iteration number. We further show that under NTK-separability, the regret is less than $\mathcal{O}(\text{poly}(\log T))$ when the width is larger than $\mathcal{O}(\text{poly}(\log T))$. We also provide a comparable regret bound for the scalar version of RMSProp and AdamNC, without requiring prior knowledge of problem parameters for learning rates. Our analysis can also be applied to smooth losses, leading to similar regret bounds.

## 1 INTRODUCTION

RMSProp (Tieleman & Hinton, 2012) and Adam (Kingma & Ba, 2015) are among the popular optimizers for training neural networks. They are variants of AdaGrad-type algorithms (Duchi et al., 2011) where the main novelties are to involve exponential moving averages (EMA) for the first and second moments of gradients, with the decay rates controlled by $\beta_1$ and $\beta_2$, respectively.

Numerous works have studied the regret bounds of RMSProp and Adam in online convex optimization (OCO) (Cesa-Bianchi et al., 2004; Hazan et al., 2016). Generally, in the online optimization setting, the goal is to minimize the following regret with respect to a static predictor $\boldsymbol{w} \in \mathcal{X} \subseteq \mathbb{R}^d$ given a sequence of loss functions $h_t(\cdot)$,

$$R(T) = \sum_{t=1}^{T} h_t(\boldsymbol{w}_t) - \inf_{\boldsymbol{w} \in \mathcal{X}} \sum_{t=1}^{T} h_t(\boldsymbol{w}). \tag{1}$$

At $t$-th iteration, the learner receives information related to $h_t$, and produces a predictor $\boldsymbol{w}_t \in \mathcal{X} \subseteq \mathbb{R}^{d'}$, which often represents a model's parameters or feature weights. A line of works such as (Reddi et al., 2018; Alacaoglu et al., 2020) showed $\mathcal{O}(\sqrt{T})$ regret bounds for RMSProp and AdamNC (Adam without corrective terms) in online convex optimization (i.e. $h_t$ is convex), usually relying on bounded iterates or bounded gradients. Other works (Mukkamala & Hein, 2017; Wang et al., 2019b) demonstrated $\mathcal{O}(\log T)$ regret bounds for variants of Adam in strongly convex optimization.

For the potentially non-convex non-smooth neural network optimization (Bolte & Pauwels, 2021), the regret analysis for RMSProp and Adam is limited to our knowledge. It is unclear whether RMSProp and Adam can achieve regret bounds comparable to those in standard online convex optimization settings. If such guarantees are possible, which conditions on the neural network architecture (e.g., width and depth) and algorithmic parameters (e.g., learning rates, $\beta_1$ and $\beta_2$) are required.

In this paper, we provide a regret analysis for RMSProp and AdamNC on training a deep fully-connected neural network with smooth activations. We focus on the binary classification task and consider two classes of loss functions: self-bounded losses such as logistic loss and exponential loss; and smooth losses such as logistic loss, $\ell_2$-loss and Huber loss.

We follow a common realizable setup in optimization analysis of neural networks training, assuming that the model can interpolate the data with an arbitrarily small loss $\varepsilon$ while the distance to initialization is controlled by a decreasing function $g(\varepsilon)$. This setup has been used in prior works (Chen et al., 2021; Taheri & Thrampoulidis, 2024) to analyze the convergence of Gradient Descent (GD) in training neural networks. They also indicated that the realizable setup can be implied by the linear separability of data, and the NTK-separability—the latter assumes that the neural tangent kernel (NTK) with respect to the model can separate data with a positive margin—under standard losses such as logistic loss. Building on this foundation, we summarize our main contributions as follows.

**Our contributions.** Under the interpolating realizable setup, we show that the regret for RMSProp and AdamNC is bounded by $\mathcal{O}(\text{poly}[g(1/T)])$ when training deep fully-connected neural networks with self-bounded losses, including logistic loss or exponential loss. The regret bound requires the network width to be at least $\mathcal{O}(\text{poly}[g(1/T)])$. Further assuming NTK-separability, we show that the regret is less than $\mathcal{O}(\text{poly}[\log T])$ with the minimal width of $\mathcal{O}(\text{poly}[\log T])$. The regret bound is comparable to $\mathcal{O}(\log T)$ regret bound for Online Gradient Descent (OGD) in strongly convex settings (Hazan et al., 2007). The EMA decay rates can be any constants such that $0 \leq \beta_1 \leq \beta_2 < 1$, covering a typical empirical setup of $\beta_1 = 0.9, \beta_2 = 0.999$. Unlike most existing analyses in online convex optimization, we do not assume bounded iterates. Instead, we show that the iterative sequences are automatically bounded, due to the interpolating assumption and well-behaved properties of neural networks. We also apply our analysis framework to smooth losses, leading to similar regret bounds as self-bounded losses.

We further show that for the scalar version of RMSProp and AdamNC, the regret can also be bounded by $\mathcal{O}(\text{poly}[g(1/T)])$ with polylogarithm order of width under the same setup. Interestingly, this result does not require prior knowledge of problem parameters for the learning rate.

In our analysis, we use the interpolating realizable setup and two loss function classes to derive the general regret bounds, and further deduce the result under NTK-separability with logistic or exponential losses. Our analysis starts from the online Bregman Proximal Gradient (BPG) with momentum and regularization terms, which covers RMSProp and AdamNC (and their projected forms). Central to our analysis is to provide the loss landscape of objective functions, which exhibits weakly-convex-like properties. The objective functions behave more like Lipschitz-continuous and convex functions as the width increases. Based on these properties, we use an induction argument to prove the bounded iterate and further derive the regret bound.

**Related works.** We postpone the brief discussions on related works in Section A from the appendix, mainly on the online learning analysis of adaptive gradient methods and the optimization analysis of neural network training.

**Notations.** For any positive integer $n$, we denote $[n] = \{1, 2, \cdots, n\}$. We use $\nabla_{\boldsymbol{w}}$ to denote the gradient and $\nabla_{\boldsymbol{w}}^2$ to denote the Hessian, with respect to $\boldsymbol{w}$. The $\|\cdot\|$ denotes the $\ell_2$-norm of vectors and the operator norm of matrices, and $\|\cdot\|_F$ denotes the Frobenius norm. For any positive semi-definite matrix $\boldsymbol{A} \in \mathbb{R}^{d \times d}$ and any vector $\boldsymbol{x} \in \mathbb{R}^d$, we denote $\|\boldsymbol{x}\|_{\boldsymbol{A}}^2 = \langle \boldsymbol{x}, \boldsymbol{A}\boldsymbol{x} \rangle$, and $\Pi_{\mathcal{X}}^{\boldsymbol{A}}(\boldsymbol{x}) = \min_{\boldsymbol{y} \in \mathcal{X}} \|\boldsymbol{x} - \boldsymbol{y}\|_{\boldsymbol{A}}^2, \mathcal{X} \subseteq \mathbb{R}^d$ for a closed convex set $\mathcal{X}$. Particularly, we denote $\Pi_{\mathcal{X}}(\boldsymbol{x}) = \min_{\boldsymbol{y} \in \mathcal{X}} \|\boldsymbol{x} - \boldsymbol{y}\|^2$. $\text{Diag}(\boldsymbol{x}) \in \mathbb{R}^{d \times d}$ denotes the diagonal matrix with diagonal entries given by $\boldsymbol{x}$. $\boldsymbol{x} \odot \boldsymbol{x}$ denotes the entry-wise product. We use $\mathcal{B}(\boldsymbol{x}, r)$ to denote the open ball $\{\boldsymbol{x}' \mid \|\boldsymbol{x}' - \boldsymbol{x}\| < r\}$.

## 2 PRELIMINARY

In this section, we introduce the problem setups. We focus on the binary classification task in the online learning setting. The goal is to minimize the following regret:

$$R(T) = \sum_{t=1}^{T} F_t(\boldsymbol{w}_t) - \inf_{\boldsymbol{w} \in \mathcal{X}} \sum_{t=1}^{T} F_t(\boldsymbol{w}). \tag{2}$$

Here, $(\boldsymbol{x}_t, y_t), \boldsymbol{x}_t \in \mathbb{R}^d, y_t \in \{\pm 1\}$ is the sample at $t$-th iteration, and $F_t(\boldsymbol{w}) = f(y_t \Phi(\boldsymbol{w}, \boldsymbol{x}_t))$ is the objective function with weight $\boldsymbol{w} \in \mathbb{R}^{d'}$. $\Phi(\boldsymbol{w}, \boldsymbol{x}_t)$ is the neural network output, and $f : \mathbb{R} \to \mathbb{R}^+$ is the loss, which is assumed to be twice differentiable, non-negative, and convex throughout the paper. $\mathcal{X}$ is a closed convex set. We next state some assumptions regarding models and losses.

**Assumption 1.** *For any* $(\boldsymbol{x}_t, y_t), t \geq 1$, $\|\boldsymbol{x}_t\| \leq R, y_t \in \{\pm 1\}$ *for some positive constant* $R$.

We note that Assumption 1 is standard in the optimization analysis. We also present two key assumptions on the convex loss $f : \mathbb{R} \to \mathbb{R}^+$.

**Assumption 2** (self-bounded)**.** *There exists* $\alpha_f > 0$ *such that* $|f'(u)| \leq \alpha_f f(u), \forall u \in \mathbb{R}, t \geq 1$.

**Assumption 3** (smooth)**.** *There exists* $L_f > 0$ *such that* $|f''(u)| \leq L_f, \forall u \in \mathbb{R}, t \geq 1$.

The logistic loss $f(u) = \log(1 + \mathrm{e}^{-u})$ is self-bounded with $\alpha_f = 1$ and smooth with $L_f = 1/4$. The exponential loss $f(u) = \mathrm{e}^{-u}$ is non-smooth and non-Lipschitz-continuous but self-bounded with $\alpha_f = 1$. For smooth losses, unlike previous works, e.g., (Allen-Zhu et al., 2019; Taheri & Thrampoulidis, 2024), we do not assume them to be Lipschitz continuous. We note that some smooth losses like squared loss $f(u) = (1-u)^2$ (Masnadi-Shirazi & Vasconcelos, 2008) and Huber loss $f(u) = \mathbf{1}_{u<1}(x)(1-u)^2$ are not Lipschitz continuous. In addition, the function $F_t(\boldsymbol{w})$ can still be non-convex and non-smooth with $\boldsymbol{w}$ even when $f$ is smooth and convex.

We consider the realizable setup that can show the ability of neural networks to interpolate the data.

**Assumption 4.** *Given* $T \geq 1$ *and* $\boldsymbol{w}_1 \in \mathbb{R}^{d'}$, *for any sufficiently small* $\varepsilon > 0$, *there exists* $\boldsymbol{w}^{(\varepsilon)} \in \mathcal{X}$ *and a non-increasing function* $g$ *such that* $\sum_{t=1}^{T} F_t(\boldsymbol{w}^{(\varepsilon)})/T \leq \varepsilon$ *and* $\|\boldsymbol{w}^{(\varepsilon)} - \boldsymbol{w}_1\| \leq g(\varepsilon)$.

The realizable setup indicates the power of models to perfectly interpolate the data, while the distance to the initialization is bounded by a function. Similar forms are also used in (Chen et al., 2021; Taheri & Thrampoulidis, 2024) for analyzing GD on training neural networks. We emphasize that many neural networks are proven to have sufficient expressive power to approximate the optimal solution (Poole et al., 2016; Raghu et al., 2017), thereby achieving arbitrarily small loss. For instance, neural networks with ReLU activation have the expressive capacity (Montúfar et al., 2014; Pascanu et al., 2013; Pan & Srikumar, 2016). Shallow neural networks with sigmoid activation can approximate any continuous function, a property called "universal approximators" (Hornik et al., 1989; Cybenko, 1989; Barron, 1993).

We note that the specific form of $g(\varepsilon)$ in Assumption 4 is not fully fixed. As shown in (Chen et al., 2021; Taheri & Thrampoulidis, 2024), Assumption 4 can be implied by the realizable setups of linear separability (Brutzkus et al., 2017; Wang et al., 2019a), or the following NTK-separability, provided that the network width is sufficiently large and $g(\varepsilon)$ is a logarithmic function or polynomial function. We will provide specific examples in Section 3.

**Assumption 5** (NTK-separability)**.** *Given any initialization* $\boldsymbol{w}_1 \in \mathcal{X}$. *There exists an unit vector* $\boldsymbol{w}^* \in \mathcal{X}$ *such that* $y_t \langle \nabla_{\boldsymbol{w}} \Phi(\boldsymbol{w}, \boldsymbol{x}_t), \boldsymbol{w}^* \rangle \geq \gamma > 0, \forall t \geq 1$.

The NTK-separability is used in several existing neural network optimization works, including (Ji & Telgarsky, 2020; Chen et al., 2021; Taheri & Thrampoulidis, 2024; Taheri et al., 2025). We further provide examples satisfying Assumptions 4 and 5 in Section D, including the cases of the linearly separable data and noisy XOR data distribution (Wei et al., 2019). Also, we perform some simple experiments in Section H to complement the interpolating assumption.

# 3 REGRET BOUNDS FOR RMSPROP AND ADAMNC

We consider the following $L$-layer fully connected neural network following (Du et al., 2019a; Taheri et al., 2025): $\boldsymbol{\varphi}^{(0)} = \boldsymbol{x} \in \mathbb{R}^d$, for any $i \in [L]$,

$$\boldsymbol{\varphi}^{(i)} = \frac{1}{\sqrt{m}} \boldsymbol{\sigma}\left(\boldsymbol{W}^{(i)} \boldsymbol{\varphi}^{(i-1)}\right) \in \mathbb{R}^m, \quad \Phi(\boldsymbol{w}, \boldsymbol{x}) = \frac{1}{\sqrt{m}} \boldsymbol{a}^\top \boldsymbol{\varphi}^{(L)} \in \mathbb{R}, \tag{3}$$

where $\boldsymbol{\sigma}(\boldsymbol{z}) = [\sigma(z_1), \cdots, \sigma(z_m)]^\top, \boldsymbol{z} = (z_i)_i \in \mathbb{R}^m$ is the entry-wise activation. $\boldsymbol{W}^{(j)} \in \mathbb{R}^{m \times m}$ is the weight matrix of the $j$-th layer and $\boldsymbol{a} \in \mathbb{R}^m$ is the output weight. We denote $\boldsymbol{w} = \left[\text{vec}\left(\boldsymbol{W}^{(1)}\right); \text{vec}\left(\boldsymbol{W}^{(2)}\right); \cdots; \text{vec}\left(\boldsymbol{W}^{(L)}\right)\right] \in \mathbb{R}^{d'}$, where $d' = m^2 L$, as the total trainable weight vector that combines the vectorization of each inner layer weight. Here, we fix the weight $\boldsymbol{a}$ in the last layer unchanged for the analysis simplicity, following from, e.g., (Li & Liang, 2018; Du et al., 2019b; Arora et al., 2019; Nitanda et al., 2019; Ji & Telgarsky, 2020).

For the activation, we assume that it's twice differentiable and make the following assumption.

---

**Algorithm 1** (Projected) AdamNC

---

**Input:** Horizon $T$, $\boldsymbol{w}_1 \in \mathcal{X} \subset \mathbb{R}^{d'}$, $C_0, \eta, \epsilon > 0$, $\boldsymbol{m}_0 = \boldsymbol{v}_0 = \boldsymbol{0}_{d'}$ and $\beta_1, \beta_2 \in [0, 1)$.
**for** $t = 1, \cdots, T$ **do**
$\quad$ Receive sample $(\boldsymbol{x}_t, y_t)$ and suffer loss $F_t(\cdot) = f(y_t(\cdot, \boldsymbol{x}_t))$;
$\quad$ Generate gradient $\boldsymbol{g}_t = \nabla F_t(\boldsymbol{w}_t)$;
$\quad$ $\boldsymbol{m}_t = \beta_1 \boldsymbol{m}_{t-1} + (1 - \beta_1)\boldsymbol{g}_t$;
$\quad$ $\boldsymbol{v}_t = (v_{t,i})_{i \in [d']} = \beta_2 \boldsymbol{v}_{t-1} + (1 - \beta_2)(\boldsymbol{g}_t \odot \boldsymbol{g}_t)$;
$\quad$ $\boldsymbol{\Lambda}_t = \eta^{-1} \mathrm{Diag}(\epsilon + \sqrt{v_{t,i}})_{i \in [d']}$ ;
$\quad$ $\boldsymbol{w}_{t+1} = \Pi_{\mathcal{X}}^{\boldsymbol{\Lambda}_t} \left( \boldsymbol{w}_t - C_0 \boldsymbol{\Lambda}_t^{-1} \boldsymbol{m}_t \right)$ .
**end for**

---

**Assumption 6.** *The activation satisfies that $|\sigma'(u)| \leq l, |\sigma''(u)| \leq \tilde{l}, \forall u \in \mathbb{R}$ for constants $l, \tilde{l} > 0$.*

Assumption 6 is mild including several typical activations such as Softplus $\sigma(u) = \log(1 + \mathrm{e}^u)$, Sigmoid $\sigma(u) = \frac{1}{1+\mathrm{e}^{-u}}$, Gaussian error linear unit (GELU) $\sigma(u) = \frac{u}{2}\left(1 + \mathrm{erf}(u/\sqrt{2})\right)$, and Hyperbolic Tangent function $\sigma(u) = \frac{\mathrm{e}^u - \mathrm{e}^{-u}}{\mathrm{e}^u + \mathrm{e}^{-u}}$. For simplicity, we do not consider the non-smooth ReLU, though noting that there are some smooth variants of ReLU, such as Swish activation $\sigma(u) = \frac{u}{1+\mathrm{e}^{-u}}$ (Ramachandran et al., 2017).

We use the following Gaussian initialization (He et al., 2015), where for each entry of the weight, $[\boldsymbol{a}]_i \sim \mathcal{N}\left(0, \frac{2}{m}\right)$, and

$$\left[\boldsymbol{W}_1^{(1)}\right]_{ij} \sim \mathcal{N}\left(0, \frac{2}{d}\right), \quad \left[\boldsymbol{W}_l^{(1)}\right]_{ij} \sim \mathcal{N}\left(0, \frac{2}{m}\right), \forall l \in \{L\},$$

$$\boldsymbol{w}_1 = \Pi_{\mathcal{X}}\left(\left[\mathrm{vec}\left(\boldsymbol{W}_1^{(1)}\right); \mathrm{vec}\left(\boldsymbol{W}_1^{(2)}\right); \cdots; \mathrm{vec}\left(\boldsymbol{W}_1^{(L)}\right)\right]\right). \tag{4}$$

Note that this initialization is not essential to our proof, and other initialization schemes, such as sub-Gaussian initialization with different variances, are possible.

We also recall AdamNC (Reddi et al., 2018) in Algorithm 1, and RMSProp is the special case of AdamNC with $\beta_1 = 0$. When $\mathcal{X} = \mathbb{R}^{d'}$, Algorithm 1 is the vanilla form without projection.

### 3.1 REGRET BOUNDS UNDER SELF-BOUNDED LOSSES AND SMOOTH LOSSES

We provide simplified regret bounds for Algorithm 1 under Assumptions 2 and 3, considering the interpolating realizable setup in Assumption 4. The detailed versions are in Section B.

**Theorem 1** (Under self-bounded losses). *Let $\{\boldsymbol{w}_s\}_{s \in [T]}$ be generated by Algorithm 1 with (4), and $\Phi(\boldsymbol{w}, \boldsymbol{x})$ be in (3). Let Assumptions 1, 2, 4 and 6 hold. For any $\epsilon > 0, T \geq 1$, we define $D_{\ell_2} = \frac{3}{2} + g(\frac{\epsilon}{T})$ and $r = 4D_{\ell_2} + 12\sqrt{2}$. If $0 < \beta_1 \leq \beta_2 < 1$,*

$$C_0\eta \leq \mathcal{O}\left(\frac{1}{g(\epsilon/T)L^2}\right), \quad and \quad m \geq \mathcal{O}\left(L^{3.5}g^4\left(\frac{\epsilon}{T}\right) + 1\right),$$

*then, with probability at least $1 - 2(L+1)\exp(-\frac{m}{2})$ with regard to the initialization,*

$$R(T) \leq \sum_{t=1}^{T} F_t(\boldsymbol{w}_t) \leq \mathcal{O}\left(L^2 g^3\left(\frac{\epsilon}{T}\right) + 1\right), \quad \forall s \in [T]. \tag{5}$$

**Theorem 2** (Under smooth losses). *Let $\{\boldsymbol{w}_s\}_{s \in [T]}$ be generated by Algorithm 1 with (4), and $\Phi(\boldsymbol{w}, \boldsymbol{x})$ be in (3). Let Assumptions 1, 3, 4 and 6 hold. For any $\epsilon_0 > 0, T \geq 1$ and $0 < \beta_1 \leq \beta_2 < 1$, we define*

$$D_{\ell_2}^2 = \frac{9}{4} + g^2\left(\frac{\epsilon_0}{T}\right) + \frac{10L_f}{\epsilon_0(1 - \beta_1)\beta_2} + \frac{32L_f L}{\epsilon_0(1 - \beta_1)^2(1 - \beta_2)^2} + \frac{4 + \sqrt{2}}{\epsilon_0},$$

*and $r = 4D_{\ell_2} + 12\sqrt{2}$. If $C_0\eta = \frac{\tilde{C}_0}{\sqrt{T}}, \epsilon = \frac{\epsilon_0}{\sqrt{T}}$,*

$$\tilde{C}_0 \leq \mathcal{O}\left(\frac{1}{g(\epsilon_0/T)L^{1.5}}\right), \quad and \quad m \geq \mathcal{O}\left(TL^{3.5}g^4\left(\frac{\epsilon_0}{T}\right) + T\right), \tag{6}$$

---

*then, with probability at least $1 - 2(L+1)\exp(-\frac{m}{2})$ with regard to the initialization,*

$$R(T) \leq \sum_{t=1}^{T} F_t(\boldsymbol{w}_t) \leq \mathcal{O}\left(L^{2.5} g^3 \left(\frac{\epsilon_0}{T}\right) + 1\right), \quad \forall s \in [T]. \tag{7}$$

We make some comments for the above two results. First, these results hold for any closed convex set $\mathcal{X}$, thus being applicable for both vanilla and projection forms of RMSProp and AdamNC. Second, unlike the analysis in online convex optimization with bounded iteration assumption, our results indicate that the iterate can be restricted with the norm less than $r \sim \mathcal{O}(g(\frac{1}{T}) + L)$. Lastly, our results allow constant setup of $\beta_1$ and $\beta_2$ (only requiring $0 < \beta_1 \leq \beta_2 < 1$), covering the typical empirical setting with $\beta_1 = 0.9$ and $\beta_2 = 0.999$, which is different from $\beta_1 \to 0$ in some results for online convex optimization (Reddi et al., 2018; Huang et al., 2019; Chen et al., 2020). Our results with constant setups of $\beta_1$ and $\beta_2$ do not contradict the negative examples in (Reddi et al., 2018) constructed for general online convex optimization, ignoring other benefit assumptions, where $\beta_1$ and $\beta_2$ are fixed first, and then problem instances are constructed in which the regret bounds diverge.

We further apply the above regret bounds under NTK-separability. We first show that NTK-separability can imply Assumption 4 with specific forms of $g(\cdot)$. We refer to Section D for the detailed proof of Proposition 1 and Remark 1.

**Proposition 1.** *Let Assumption 5 hold, $\boldsymbol{w}_1$ and $\boldsymbol{a}$ are as in (4), $|\Phi(\boldsymbol{w}_1, \boldsymbol{x}_t)| \leq C, \forall t \geq 1$, and Assumptions 1 and 6 hold. **(a).** If $f$ is logistic loss or exponential loss, and $m \geq \mathcal{O}(\log^4(1/\varepsilon))$, then, Assumption 4 holds with $g(\varepsilon) \sim \mathcal{O}(\log(1/\varepsilon))$. **(b).** If $f(u) = 1/u^\beta, \beta > 0$, and $m \geq \mathcal{O}((1/\varepsilon)^{4/\beta})$, then Assumption 4 holds with $g(\varepsilon) \sim \mathcal{O}((1/\varepsilon)^{1/\beta})$.*

**Remark 1.** *Under the assumptions of Proposition 1 and Theorem 1 (without Assumption 2), it holds that with high probability, $R(T) \leq \mathcal{O}(L^2(\log^3 T))$ when $m \geq \mathcal{O}(L^{3.5}(\log^4 T))$.*

We note that the regret bound is comparable to those in online strongly convex optimization, e.g., for OGD (Hazan et al., 2007), AdaGrad and RMSProp variants (Mukkamala & Hein, 2017), and Adam variants (Wang et al., 2019b). The reason arises from the benign properties of $F_t(\boldsymbol{w})$ induced by deep neural networks, as shown in Section 4.

## 3.2 EXTENSION FOR ANOTHER NEURAL NETWORK MODEL

We also apply our analysis to another structure of deep neural networks that has been commonly used in literature e.g., (Allen-Zhu et al., 2019; Zou & Gu, 2019; Chen et al., 2021), where the normalization factor $\frac{1}{\sqrt{m}}$ vanishes in the inner layer: $\boldsymbol{\varphi}^{(0)} = \boldsymbol{x} \in \mathbb{R}^d$, for any $i \in [L]$,

$$\boldsymbol{\varphi}^{(i)} = \boldsymbol{\sigma}\left(\boldsymbol{W}^{(i)}\boldsymbol{\varphi}^{(i-1)}\right) \in \mathbb{R}^m, \quad \Phi(\boldsymbol{w}, \boldsymbol{x}) = \frac{1}{\sqrt{m}}\boldsymbol{a}^\top \boldsymbol{\varphi}^{(L)} \in \mathbb{R}, \tag{8}$$

We then provide the regret bound for training (8) with Algorithm 1. We refer to Section F for proof of Theorem 3 and Remark 2.

**Theorem 3.** *Let $\{\boldsymbol{w}_s\}_{s\in[T]}$ be generated by Algorithm 1 with (4), and $\Phi(\boldsymbol{w}, \boldsymbol{x})$ be in (8) with $\sigma(0) = 0$. Suppose that $f$ is logistic loss, and Assumptions 1, 4 and 6 hold. If $C_0\eta \leq 1$, and $m \geq \mathcal{O}(g^L(\epsilon/T))$, then with probability at least $1 - 2(L+1)\exp(-\frac{m}{2})$ with regard to the initialization, $R(T) \leq \mathcal{O}(g^2(\epsilon/T) + 1)$.*

We also show that under (8) and NTK-separability, Assumption 4 can be implied with $g(\varepsilon)$ as a logarithm function. Consequently, we can apply Theorem 3 under NTK-separability.

**Remark 2.** *If assumptions of Theorem 3 (without Assumption 2) are satisfied, then with high probability $R(T) \leq \mathcal{O}(\log^2 T)$ when $m \geq \mathcal{O}(\log^L T)$.*

Our results show a comparable $\mathcal{O}(\text{poly}[\log T])$ regret bound to the ones for GD training deep neural networks under the same NTK-separability (Chen et al., 2021; Taheri et al., 2025). Chen et al. (2021) required a minimal width of $\mathcal{O}(\text{poly}(L, \log T))$ comparable to the one in Theorem 1 whereas Taheri & Thrampoulidis (2024) required at least $\mathcal{O}(\text{poly}(\log^L T))$ width comparable to the one in Theorem 3. We highlight that Theorem 3 only requires parameter-free learning rates, while two existing works required prior knowledge of problem parameters for setting the learning rate. This

is an adaptive advantage for Adam over GD/SGD that has been verified in stochastic optimization. Both papers considered the logistic loss and we further extend the regret bound to more general smooth or self-bounded losses. In addition, our regret bounds are implicitly better than GD when the gradient is sparse, which is also observed in AdaGrad (Duchi et al., 2011)[1].

**Experiments.** We provide some simple experiments in Section H to complement our theoretical results, particularly the interpolating data condition (Assumption 4), and the regret bounds.

## 4 TECHNICAL LEMMAS

We analyze the properties of the objective function. Their proofs are postponed in Sections C and D.

**Properties of models.** We provide the following results to locally bound the gradient and Hessian norm of the neural network (3). The results for the neural network (8) are in Section F.

**Lemma 1.** *Given any $r > 0$, let $R_1 = R + 2|\sigma(0)|$, and*

$$R_2 = R_1 l r \sqrt{L}, \quad R_3 = 2R_1 \left( \tilde{l} R_1 r (1 + 2lr) + 2l^2 r \right) \sqrt{L}.$$

*For any $\boldsymbol{w} \in \mathbb{R}^{d'}$, let $\boldsymbol{W}^{(i)} \in \mathbb{R}^{m \times m}$ be the weight at $i$-th layer. If Assumptions 1 and 6 hold, $\Phi(\boldsymbol{w}, \boldsymbol{x})$ is as in (3), $\|\boldsymbol{W}^{(i)}\| \leq \frac{r}{2}, \forall i \in [L], , \|\boldsymbol{a}\| \leq r$ and $\sqrt{m} \geq \max\{1, 2lr\}$, then,*

$$\|\nabla_{\boldsymbol{w}}\Phi(\boldsymbol{w}, \boldsymbol{x})\| \leq \frac{R_2}{m}, \quad \|\nabla_{\boldsymbol{w}}^2\Phi(\boldsymbol{w}, \boldsymbol{x})\| \leq \frac{R_3}{m}, \quad \forall \boldsymbol{x} \in \mathbb{R}^d.$$

We note that a similar $\mathcal{O}(\frac{1}{\sqrt{m}})$ bound for the Hessian norm is derived in (Liu et al., 2020; Taheri et al., 2025). We also obtain $\mathcal{O}(\frac{1}{\sqrt{m}})$ bounds for both graidents and Hessian of (8). For the Gaussian initialization, we have the following standard result to bound its norm.

**Lemma 2.** *Suppose that the initialization is generated by (4). Then, with probability at least $1 - 2(L+1)\exp(-m/2)$, $\left\|\boldsymbol{W}_1^{(i)}\right\| \leq 3\sqrt{2}, \forall i \in [L]$ and $\|\boldsymbol{a}\| \leq 3\sqrt{2}$.*

Based on Lemma 1, we provide a key lemma to bound the gradient and Hessian norm of $F_t(\boldsymbol{w})$ when Assumption 2 holds. We use $\lambda_{\min}$ to denote the minimum eigenvalue in the following lemmas.

**Lemma 3.** *Generally, let the same assumptions of Lemma 1 hold. If Assumption 2 holds, then*

$$\|\nabla F_t(\boldsymbol{w})\| \leq \frac{\alpha_f R_2}{m} \cdot F_t(\boldsymbol{w}), \quad \lambda_{\min}(\nabla^2 F_t(\boldsymbol{w})) \geq -\frac{\alpha_f R_3}{m} \cdot F_t(\boldsymbol{w}), \quad \forall t \geq 1.$$

Lemma 3 indicates the weakly-convex-like property of $F_t(\boldsymbol{w})$. In addition, with the increase of width, $F_t(\boldsymbol{w})$ behaves more like a Lipschitz-continuous and convex function. A similar result is derived in (Taheri et al., 2025) under logistic loss, which controls the norm of gradients and Hessian of (3) with a $\mathcal{O}(\frac{1}{\sqrt{m}})$ bound. When Assumption 3 holds, we can also derive upper bounds for the magnitude of the gradient and Hessian of $F_t(\boldsymbol{w})$ as follows.

**Lemma 4.** *Generally, let the same assumptions of Lemma 1 hold. If Assumption 3 holds, then*

$$\|\nabla F_t(\boldsymbol{w})\| \leq \frac{2L_f R_2}{m}\sqrt{F_t(\boldsymbol{w})}, \quad \lambda_{\min}(\nabla^2 F_t(\boldsymbol{w})) \geq -\frac{R_3\sqrt{2L_f F_t(\boldsymbol{w})}}{m}, \quad \forall t \geq 1.$$

Based on the weakly-convex-like properties in Lemmas 3 and 4, we have the following result.

**Lemma 5.** *Let the same assumptions of Lemma 1 hold. Also, let $\|\boldsymbol{W}^{(i)}\|, \|\boldsymbol{W}_t^{(i)}\| \leq \frac{r}{2}, \forall i \in [L]$ and $\|\boldsymbol{w}_t - \boldsymbol{w}\| \leq D$. If Assumption 2 holds and $m \geq \max\{1, 4l^2r^2, \frac{9\alpha_f D^2 R_3}{2}\}$, then,*

$$F_t(\boldsymbol{w}_t) - F_t(\boldsymbol{w}) \leq \langle \boldsymbol{w}_t - \boldsymbol{w}, \nabla F_t(\boldsymbol{w}_t) \rangle + \frac{F_t(\boldsymbol{w}_t) + F_t(\boldsymbol{w})}{8}.$$

*If Assumption 3 holds, and $m \geq \max\{1, 4l^2r^2, D^2 R_3\sqrt{L_f T}\}$, then,*

$$F_t(\boldsymbol{w}_t) - F_t(\boldsymbol{w}) \leq \langle \boldsymbol{w}_t - \boldsymbol{w}, \nabla F_t(\boldsymbol{w}_t) \rangle + \frac{F_t(\boldsymbol{w}_t) + F_t(\boldsymbol{w})}{8} + \frac{2 + 1/\sqrt{2}}{T}.$$

---

[1]A brief discussion is in Remark 4 of Section B.

## 5 PROOF OF THEOREM 1

We provide a detailed proof to derive Theorem 1. The proof for Theorems 2 and 3 can be found in Section F, which shares many similarities to Theorem 1. Also, all detailed proofs for lemmas in this section are in Section E.

**Bregman Proximal Gradient.** Our analysis starts from deriving a general result for online Bregman Proximal Gradient (BPG) (Censor & Lent, 1981) with momentum and regularization. For a closed convex set $\mathcal{X}$ with nonempty interior, we update $\boldsymbol{w}_t$ as follows,

$$\boldsymbol{g}_t = (g_{t,i})_{i \in [d']} = \nabla F_t(\boldsymbol{w}_t), \quad \boldsymbol{m}_t = \beta_1 \boldsymbol{m}_{t-1} + (1 - \beta_1)\boldsymbol{g}_t, \quad \beta_1 \in [0, 1),$$

$$\boldsymbol{w}_{t+1} = \arg\min_{\boldsymbol{w} \in \mathcal{X}} \left\{ \langle \boldsymbol{w}, \boldsymbol{m}_t \rangle + \phi(\boldsymbol{w}) + \frac{1}{C_0} B_{\psi_t}(\boldsymbol{w}, \boldsymbol{w}_t) \right\}, \tag{9}$$

where $\phi : \mathcal{X} \to \mathbb{R}^+$ is the regularization, $B_{\psi_t}(\boldsymbol{z}_1, \boldsymbol{z}_2) = \psi_t(\boldsymbol{z}_1) - \psi_t(\boldsymbol{z}_2) - \langle \boldsymbol{z}_1 - \boldsymbol{z}_2, \nabla \psi_t(\boldsymbol{z}_2) \rangle$ is the Bregman divergence with respect to a continuous differentiable function $\psi_t : \mathcal{X} \to \mathbb{R}$. We require $\psi_t$ to be 1-strongly convex with the semi-norm $\| \cdot \|_{\psi_t}$ following from (Duchi et al., 2011),

$$\psi_t(\boldsymbol{z}_1) - \psi_t(\boldsymbol{z}_2) - \langle \boldsymbol{z}_1 - \boldsymbol{z}_2, \nabla \psi_t(\boldsymbol{z}_2) \rangle \geq \frac{1}{2} \|\boldsymbol{z}_1 - \boldsymbol{z}_2\|_{\psi_t}^2, \quad \forall \boldsymbol{z}_1, \boldsymbol{z}_2 \in \mathcal{X}. \tag{10}$$

We use $\| \cdot \|_{\psi_t^*}$ to denote the dual norm of $\| \cdot \|_{\psi_t}$. Note that (9) provides a general form covering some first-order gradient methods. For example, by setting $\psi_t(\boldsymbol{z}) = \frac{1}{2}\|\boldsymbol{z}\|_{\boldsymbol{\Lambda}_t}^2$ with $\boldsymbol{\Lambda}_t$ defined in Algorithm 1, the update reduces to RMSProp and AdamNC. This choice of $\psi_t(\boldsymbol{z})$ also satisfies (10). Further details can be found in Proposition 3 from Section E. Then, we provide a general result for the online BPG with momentum, borrowing some techniques from (Alacaoglu et al., 2020).

**Proposition 2.** *Let $\{\boldsymbol{w}_t\}_{t \geq 1}$ be genenerated by (9), and (10) holds. For any $\boldsymbol{w} \in \mathcal{X}$, it holds that*

$$\sum_{s=1}^{t} \langle \boldsymbol{w}_s - \boldsymbol{w}, \nabla F_s(\boldsymbol{w}_s) \rangle \leq \frac{\beta_1}{1 - \beta_1} \langle \boldsymbol{w}_t - \boldsymbol{w}, \boldsymbol{m}_t \rangle + \underbrace{\frac{1}{C_0} \sum_{s=1}^{t} (B_{\psi_s}(\boldsymbol{w}, \boldsymbol{w}_s) - B_{\psi_s}(\boldsymbol{w}, \boldsymbol{w}_{s+1}))}_{(*)}$$

$$+ \sum_{s=1}^{t} (\phi(\boldsymbol{w}) - \phi(\boldsymbol{w}_{s+1})) + \frac{2C_0}{(1 - \beta_1)^2} \sum_{s=1}^{t} \|\boldsymbol{m}_s\|_{\psi_s^*}^2. \tag{11}$$

Proposition 2 serves as an important result for deriving the regret bound $R(T)$ since its LHS can be further lower bounded by $\sum_{s=1}^{t} (F_s(\boldsymbol{w}_s) - F_s(\boldsymbol{w}))$ through Lemma 5.

**Lemmas to estimate RHS of** (11). First, we note that the results in Section 4 are established locally, requiring that $\{\boldsymbol{w}_t\}_{t \geq 1}$ and $\boldsymbol{w}$ to be within $\mathcal{B}(\boldsymbol{0}_{d'}, r)$. Given this, we will provide three lemmas based on the following temporary assumptions.

**Assumption 7.** *(a). Suppose that $\Phi(\boldsymbol{w}, \boldsymbol{x})$ is as in (3), $\{\boldsymbol{w}_s\}$ is generated by Algorithm 1, or equally by (9) with $\psi_s(\boldsymbol{z}) = \frac{1}{2}\|\boldsymbol{z}\|_{\boldsymbol{\Lambda}_s}^2$ where $\boldsymbol{m}_s, \boldsymbol{v}_s$ and $\boldsymbol{\Lambda}_s$ are given in Algorithm 1.*
*(b). Suppose that the following inequalities hold with probability at least $1 - 2(L+1)\exp(-m/2)$ with regard to the initialization: given some constants $D_{\ell_2}, r > 0$ and some $\boldsymbol{w} \in \mathcal{X}$,*

$$\|\boldsymbol{w}_s - \boldsymbol{w}\| \leq D_{\ell_2}, \quad \|\boldsymbol{W}^{(i)}\| \leq \frac{r}{2}, \quad \|\boldsymbol{W}_s^{(i)}\| \leq \frac{r}{2}, \quad \forall s \in [t], i \in [L], \tag{12}$$

*Also, let $\sqrt{m} \geq \max\{1, 2lr\}$, and consequently Lemma 1 holds.*

**Lemma 6.** *Generally, let Assumption 7 hold. When Assumption 2 holds, it leads to that*

$$\frac{\beta_1}{1 - \beta_1} \langle \boldsymbol{w}_t - \boldsymbol{w}, \boldsymbol{m}_t \rangle \leq \frac{D_{\ell_2} \beta_1 \alpha_f R_2}{m} \sum_{s=1}^{t} F_s(\boldsymbol{w}_s).$$

*When Assumption 3 holds, it holds that*

$$\frac{\beta_1}{1 - \beta_1} \langle \boldsymbol{w}_t - \boldsymbol{w}, \boldsymbol{m}_t \rangle \leq \frac{4(D_{\ell_2}\beta_1)^2 L_f}{1 - \beta_1} \left( \frac{R_2}{m} \right)^2 + \sum_{s=1}^{t} \frac{F_s(\boldsymbol{w}_s)}{8}.$$

**Lemma 7.** *Generally, let Assumption 7 hold. When Assumption 2 holds, $(*)$ in (11) is bounded by*

$$(*) \leq \frac{D_{\ell_2}^2 \alpha_f R_2 \sqrt{1-\beta_2}}{C_0 \eta m (1-\sqrt{\beta_2})} \sum_{s=1}^t F_s(\boldsymbol{w}_s) + \frac{\epsilon \|\boldsymbol{w}-\boldsymbol{w}_1\|^2}{2C_0\eta} - \frac{\|\boldsymbol{w}-\boldsymbol{w}_{t+1}\|_{\boldsymbol{\Lambda}_t}^2}{2C_0}.$$

*When Assumption 3 holds, $(*)$ in (11) is bounded by*

$$(*) \leq \sum_{s=1}^t \frac{F_s(\boldsymbol{w}_s)}{8} + \frac{\epsilon \|\boldsymbol{w}-\boldsymbol{w}_1\|^2}{2C_0\eta} + \left(\frac{D_{\ell_2}^2 R_2}{C_0\eta m}\right)^2 \frac{L_f t}{\beta_2} - \frac{\|\boldsymbol{w}-\boldsymbol{w}_{t+1}\|_{\boldsymbol{\Lambda}_t}^2}{2C_0}.$$

**Lemma 8.** *Generally, let Assumption 7 hold, and $0 \leq \beta_1 \leq \beta_2 < 1$. When Assumption 2 holds, it leads to that*

$$\frac{2C_0}{(1-\beta_1)^2} \sum_{s=1}^t \|\boldsymbol{m}_s\|_{\boldsymbol{\Lambda}_s^{-1}}^2 \leq \frac{2C_0\eta \alpha_f R_2 \sqrt{d'}}{(1-\beta_1)(1-\sqrt{\beta_2})^2 m} \sum_{s=1}^t F_s(\boldsymbol{w}_s).$$

*When Assumption 3 holds, it leads to that*

$$\frac{2C_0}{(1-\beta_1)^2} \sum_{s=1}^t \|\boldsymbol{m}_s\|_{\boldsymbol{\Lambda}_s^{-1}}^2 \leq \sum_{s=1}^t \frac{F_s(\boldsymbol{w}_s)}{8} + \frac{16(C_0\eta)^2 L_f d' t}{(1-\beta_1)^2 (1-\beta_2)^2} \left(\frac{R_2}{m}\right)^2.$$

Based on three lemmas and Proposition 2, we can prove (12) and the main regret bounds through the following induction argument.

**Proof of Theorem 1 (Theorem 5).** First, we set $\boldsymbol{w} = \boldsymbol{w}^{(\epsilon/T)} \in \mathcal{X}$ satisfying Assumption 4.

**Case $k = 1$.** Using Assumption 4, Lemma 2 and the norm inequality, we get that $\|\boldsymbol{w} - \boldsymbol{w}_1\| \leq g(\epsilon/T) \leq D_{\ell_2}$ and $\|\boldsymbol{W}_1^{(l)}\|, \|\boldsymbol{W}^{(l)}\| \leq \frac{r}{4}, \forall l \in [L], \|\boldsymbol{a}^\top\| \leq r$ with $D_{\ell_2}$ and $r$ defined in Theorem 1.

**Case $k = t + 1$.** Suppose that (12) holds for some $t \in [T]$ with $D_{\ell_2}$, $r$ and $\boldsymbol{w}$ defined above. Based on the induction assumption, Assumption 7 is satisfied, and consequently Lemmas 5 to 8 hold under self-bounded losses with $D = D_{\ell_2}$ and $r$ defined in Theorem 1, and subsequent terms $R_1, R_2, R_3$. Using Proposition 2 (since $\boldsymbol{w} \in \mathcal{X}$) with $\phi(\cdot) = 0$, and combining Lemmas 5 to 8, and the requirements of $C_0, \eta, m$ in Theorem 1,[2] we get that

$$\sum_{s=1}^t \frac{F_s(\boldsymbol{w}_s)}{2} \leq \frac{9}{8} \sum_{s=1}^t F_s(\boldsymbol{w}) + \frac{\epsilon \|\boldsymbol{w}-\boldsymbol{w}_1\|^2}{2C_0\eta} - \frac{\|\boldsymbol{w}-\boldsymbol{w}_{t+1}\|_{\boldsymbol{\Lambda}_t}^2}{2C_0}. \tag{13}$$

Re-arranging (13), and using $F_s(\cdot) \geq 0$ and $\boldsymbol{\Lambda}_t \succeq (\epsilon/\eta)\mathbf{I}_{d' \times d'}$, we get that

$$\frac{\epsilon \|\boldsymbol{w}-\boldsymbol{w}_{t+1}\|^2}{2C_0\eta} \leq \frac{\|\boldsymbol{w}-\boldsymbol{w}_{t+1}\|_{\boldsymbol{\Lambda}_t}^2}{2C_0} \leq \frac{9}{8} \sum_{s=1}^t F_s(\boldsymbol{w}) + \frac{\epsilon \|\boldsymbol{w}-\boldsymbol{w}_1\|^2}{2C_0\eta} \leq \frac{9\epsilon}{8} + \frac{\epsilon \|\boldsymbol{w}-\boldsymbol{w}_1\|^2}{2C_0\eta},$$

where the last inequality uses $\sum_{s=1}^t F_s(\boldsymbol{w}) \leq \sum_{s=1}^T F_s(\boldsymbol{w}) \leq \epsilon$ from Assumption 4. Based on $C_0\eta \leq 1$ and the norm inequality, we get that for any $l \in [L]$,

$$\|\boldsymbol{W}^{(l)} - \boldsymbol{W}_{t+1}^{(l)}\| \leq \|\boldsymbol{w} - \boldsymbol{w}_{t+1}\| \leq D_{\ell_2} \leq \frac{r}{4}, \quad \|\boldsymbol{W}_{t+1}^{(l)}\| \leq \|\boldsymbol{W}^{(l)}\| + \|\boldsymbol{W}^{(l)} - \boldsymbol{W}_{t+1}^{(l)}\| \leq \frac{r}{2}.$$

We then prove that (12) holds for any $t \in [T]$. Based on (12) and $\inf_{\boldsymbol{w} \in \mathcal{X}} \sum_{s=1}^t F_s(\boldsymbol{w}) \geq 0$, we can use (13) to get the desired result in (5).

**Proof challenges.** We briefly list some of unique challenges for deriving regret bounds of RMSProp and AdamNC in neural network optimization. In Section B , we provide a more detailed discussion. **Challenge (I).** When considering GD and AdaGrad/AMSGrad, the summation term $(*)$ in (11) forms a telescoping sum due to the constant or non-increasing step-sizes. In comparison, step-sizes of RMSProp and AdamNC are non-monotonic given only constants $C_0\eta$ and $\beta_1, \beta_2$, leading to a challenge in controlling $(*)$. **Challenge (II).** Many existing regret bounds for Adam or other adaptive methods in online-convex optimization rely on the assumptions of bounded gradients and iterations, such as (Duchi et al., 2011; Alacaoglu et al., 2020). In comparison, we rely on neither of the assumptions, which may be closer to the practical setting.

To tackle **Challenge (I)**, we propose new bounds in Lemmas 7 and 18 as well as other related lemmas to control $(*)$ based on the EMA of AdamNC. We also tackle **Challenge (II)** through properties of gradient and Hessian norms in Lemmas 3, 4 and 15 and the induction argument.

---

[2]We refer to Theorem 5 for detailed requirements and definitions of $R_1, R_2, R_3$ in Section B.

## 6 REGRET BOUNDS FOR RMSPROP-NORM AND ADAMNC-NORM: FULL ADAPTIVITY OF THE LEARNING RATE

In this section, we investigate RMSProp-Norm and AdamNC-Norm, the scalar version of RMSProp and AdamNC, where we replace $\boldsymbol{\Lambda}_t$ by the following form in Algorithm 1,

$$\boldsymbol{\Lambda}_t = \eta^{-1}(\epsilon + \sqrt{v_t}) \cdot \mathbf{I}_{d' \times d'}, \quad v_0 = 0, v_t = \beta_2 v_{t-1} + (1 - \beta_2)\|\boldsymbol{g}_t\|^2, \quad \forall t \geq 1. \quad (14)$$

We provide the simplified version of regret bounds with detailed versions in Section G.

**Theorem 4** (Informal). *Let $\{\boldsymbol{w}_s\}_{s\in[T]}$ be generated by Algorithm 1 with* (14) *in replacement and* (4)*, and $\Phi(\boldsymbol{w}, \boldsymbol{x})$ be as in* (3)*. Also, let $0 < \beta_1 \leq \beta_2 < 1$, and $C_0, \eta$ be any positive constants. The following results hold with probability at least $1 - 2(L + 1)\exp(-\frac{m}{2})$ with initialization.*

*(a). Let the same assumptions in Theorem 1 hold. If $m \geq \mathcal{O}(L^{2.5}(g^4(\frac{1}{T}) + 1))$, then $R(T) \leq \mathcal{O}(g^2(\frac{1}{T}) + 1)$. (b). Let the same assumptions Theorem 2. If $m \geq \mathcal{O}\left(L^{2.5}\sqrt{T}(g^4(\frac{1}{T}) + 1)\right)$, then $R(T) \leq \mathcal{O}(g^2(\frac{1}{T}) + 1)$.*

**Remark 3** (Under NTK-separability). *Under the same assumptions of **(a)** in Proposition 1 and Theorem 4 (without Assumption 4), it holds with probability at least $1 - 2(L + 1)\exp(\frac{-m}{2})$ with regard to the initialization, $R(T) \leq \mathcal{O}(\log^2 T)$ when $m \geq \mathcal{O}(L^{2.5}(\log^4 T))$.*

Our results show that the regret bounds and the minimal width requirement of training (3) with RMSProp-Norm and AdamNC-Norm are also comparable to Theorems 1 and 2 for the coordinate-wise case. Moreover, the regret bounds do not require prior knowledge of problem parameters such as $\alpha_f$ and $L_f$ for setting learning rates $C_0$ and $\eta$. This adaptivity of learning rates verifies the advantage of adaptive methods over non-adaptive methods, as also shown in standard online convex optimization (Duchi et al., 2011; Reddi et al., 2018; Alacaoglu et al., 2020).

**A proof sketch.** We first note that Proposition 2 and Lemma 5 hold for any sequence generated by (9). In addition, all results in Section 4 are algorithm-independent. Second, we also assume that (12) holds for some $t \in [T]$ with $D_{\ell_2}, r > 0$ and $\boldsymbol{w} \in \mathbb{R}^{d'}$. Then, recalling the detailed proofs, we can easily show that Lemmas 6 and 7 remain unchanged for RMSProp-Norm and AdamNC-Norm. The major difference, and also the main reason of the full adaptivity of learning rates, comes from bounding $\frac{2C_0}{(1-\beta_1)^2}\sum_{s=1}^{t}\|\boldsymbol{m}_s\|^2_{\boldsymbol{\Lambda}_s^{-1}}$ without explicit $\mathcal{O}(d'), d' = m^2 L$ factor, as detailed in Lemma 20 from Section G. Then, we can use $m$ to adjust the upper bound of $\sum_{s=1}^{t}\|\boldsymbol{m}_s\|^2_{\boldsymbol{\Lambda}_s^{-1}}$ instead of the learning rate $C_0\eta$. The induction argument is also applied to ensure that (12) holds, which further leads to the regret bound. We refer to Section G for the detailed proof.

## 7 CONCLUSION

We provide an online regret analysis for RMSProp and AdamNC (possibly with projection), when training $L$-layer neural networks. We require the loss to be self-bounded or smooth, including two typical types, logistic loss and exponential loss. Under the NTK-separability, our results show a polylogarithmic order of regret bound, provided that the minimal width is at least of polylogarithmic order. We further show the similar regret bounds for scalar versions of RMSProp and Adam, without the need for prior knowledge of problem parameters for setting learning rates. During our proof, we analyze a more general Bregman Proximal Gradient and consider a weaker interpolating realizable setup that covers NTK-separability. Moreover, instead of directly assuming boundedness of the iterates, we employ an induction argument to prove their boundedness.

**Limitations.** The results rely on the NTK regime. Extending our results beyond this setting is left as a future problem. Also, it may be beneficial to provide more experiments in practical cases to verify the realizable assumptions and regret bounds.

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

## DECLARATION OF LLM USAGE

We used a large language model (LLM) only for language polishing (grammar, clarity, and style). The model did not generate ideas, analyses, results, or citations. The authors are fully responsible for all content.

## A    RELATED WORKS

We only briefly mention the most related works due to space and knowledge constraints.

### A.1    ADAPTIVE GRADIENT METHODS IN ONLINE LEARNING OPTIMIZATION

Existing works have shown that many adaptive gradient methods can achieve a regret bound of $\mathcal{O}(\sqrt{T})$ in online convex optimization, matching those for non-adaptive gradient methods, such as OGD (Hazan et al., 2016) and their variants (Zinkevich, 2003; Bartlett et al., 2007; Duchi et al., 2010). For example, Duchi et al. (2011) and Streeter & McMahan (2010) proposed AdaGrad-(Norm) and showed that it can even achieve a lower regret bound than non-adaptive gradient methods when the gradients are sparse. RMSProp and Adam were proposed to use an EMA to replace the cumulative gradient mechanism in AdaGrad. Reddi et al. (2018) pointed out an issue in the original regret analysis of Adam (Kingma & Ba, 2015), and further showed a $\mathcal{O}(\sqrt{T})$ regret bound for variants of Adam provided that $\beta_1 \to 0$, including AMSGrad and AdamNC. Many other variants of Adam have been designed with good practical properties, and most of them can achieve comparable regret bounds to AMSGrad, while still requiring $\beta_1 \to 0$, including AdaBound (Luo et al., 2019; Savarese, 2019) and Nostalgic Adam (Huang et al., 2019). To address the potential shortcoming of $\beta_1 \to 0$, Alacaoglu et al. (2020) provided a new framework to derive the regret bound of AMSGrad and AdamNC with constants $\beta_1$ and $\beta_2$.

In online strongly convex optimization, Wang et al. (2019b) provided a $\mathcal{O}(\log T)$ regret bound for SAdam, a variant of AdamNC, matching that for OGD (Hazan et al., 2007). Xiao et al. (2024) investigated the convergence of Adam in non-smooth optimization. They considered general non-smooth objective functions and provided almost sure convergence to the stationary point.

Many works investigated the convergence to stationary points of adaptive gradient methods in non-convex stochastic optimization; to name a few, AdaGrad by (Li & Orabona, 2019; Ward et al., 2020; Faw et al., 2022; Kavis et al., 2022; Liu et al., 2023; Wang et al., 2023b; Attia & Koren, 2023; Hong & Lin, 2024; Liu et al., 2025), RMSProp and Adam by (Shi et al., 2020; Défossez et al., 2022; Zhang et al., 2022; Li & Liu, 2023; Wang et al., 2023a; Hong & Lin, 2025; Li et al., 2025), and other variants by (Zaheer et al., 2018; Chen et al., 2019; Crawshaw et al., 2022; Xie et al., 2024). Most of the above works could provide the $\tilde{\mathcal{O}}(1/\sqrt{T})$ convergence rate.

### A.2    OPTIMIZATION ANALYSIS FOR NEURAL NETWORKS.

**Early research for global convergence of training neural networks.**    In early stage, many works (Tian, 2017; Brutzkus & Globerson, 2017; Li & Yuan, 2017; Zhong et al., 2017; Du & Lee, 2018; Soltanolkotabi et al., 2018; Zhang et al., 2019) usually assumed that the input data is Gaussian and that the label is generated according to a teacher neural network (i.e., an underlying ground-truth network). Based on this, they could derive the global convergence results for GD and SGD when training shallow neural networks.

Another line of works (Mei et al., 2018; Chizat & Bach, 2018; Sirignano & Spiliopoulos, 2020; Rotskoff & Vanden-Eijnden, 2018; Wei et al., 2019) employed mean field analysis to show that, for infinitely wide neural networks, the empirical distribution of parameters evolves as a Wasserstein gradient flow.

**Optimization analysis of training neural networks on $l_2$-regression.**    Many works focused on the finite-width ReLU neural networks trained by GD and SGD on $\ell_2$-regression. They typically assumed that the data satisfies some NTK-related restriction and the width grows polynomially with the sample size, motivated by the NTK theory (Jacot et al., 2018; Chizat & Bach, 2018). For instance, works like (Li & Liang, 2018; Allen-Zhu et al., 2019; Oymak & Soltanolkotabi, 2020;

Zou et al., 2020) assumed that the data is separable with a margin $\delta$, while other works (Du et al., 2019b; Arora et al., 2019; Du et al., 2019a; Wu et al., 2019) assumed the positive definiteness of the NTK with respect to data. The two assumptions are proved to be equivalent with a proper selection of $\delta$ (Zou & Gu, 2019). Based on the NTK theory and the strong convexity of $\ell_2$-loss, these works established strongly-convex-like properties for objective functions, such as the local Polyak-Łojasiewicz (PL) condition, as long as the iterate remains close to the initialization. These properties ensure linear convergence for GD and SGD. For adaptive gradient methods, to our knowledge, only Wu et al. (2019) analyzed a variant of AdaGrad-Norm known as AdaLoss, applied to a two-layer neural network with ReLU activation.

**Optimization analysis of training neural networks in classification tasks.** For classification tasks with logistic loss, several works considered using GD to train the shallow neural networks that satisfy NTK-separability. For example, Nitanda et al. (2019) proved that the average loss is less than $\tilde{O}(1/T)$, provided that the activation is smooth and the width is larger than $\mathcal{O}(\sqrt{T})$. Ji & Telgarsky (2020) improved the minimal width to $\mathcal{O}(\log T)$ when using ReLU activation. Taheri & Thrampoulidis (2024) considered smooth activations and the more general self-bounded losses, and derived a $\tilde{O}(1/T)$ bound for the average training loss with the minimal width of $\mathcal{O}(\log^4 T)$ order.

For training deep neural networks satisfying NTK-separability, Chen et al. (2021) and Taheri et al. (2025) showed the average loss is less than $\tilde{O}(1/T)$ as long as the width is at least polylogarithm order with $T$, considering ReLU activation or smooth activation, respectively. Inside the width requirement, Taheri et al. (2025) further required an exponential dependency with respect to the depth $L$ while Chen et al. (2021) only needed a polynomial dependency.

# B DETAILED MAIN RESULTS AND PROOF

## B.1 DETAILED REGRET BOUNDS

Here, we provide the detailed versions of Theorems 1 to 3.

**Theorem 5** (Detailed version of Theorem 1). *Let $\{\boldsymbol{w}_s\}_{s \in [T]}$ be generated by Algorithm 1 with* (4), *and $\Phi(\boldsymbol{w}, \boldsymbol{x})$ be in* (3). *Let Assumptions 1, 2, 4 and 6 hold. For any $\epsilon > 0, T \geq 1$, we define $D_{\ell_2} = \frac{3}{2} + g(\frac{\epsilon}{T})$, $r = 4D_{\ell_2} + 12\sqrt{2}$, $R_1 = R + 2|\sigma(0)|$,*

$$R_2 = R_1 lr\sqrt{L}, \quad and \quad R_3 = 2R_1\left(\tilde{l}R_1 r(1 + 2lr) + 2l^2 r\right)\sqrt{L}. \tag{15}$$

*Let $0 < \beta_1 \leq \beta_2 < 1$, $C_0\eta \leq \min\left\{1, \frac{(1-\sqrt{\beta_2})^2(1-\beta_1)}{8\alpha_f R_2\sqrt{L}}\right\}$, and*

$$m \geq \left\{8D_{\ell_2}\beta_1\alpha_f R_2, \frac{16D_{\ell_2}^2\alpha_f R_2\sqrt{1-\beta_2}}{C_0\eta(1-\sqrt{\beta_2})}, \frac{9\alpha_f D_{\ell_2}^2 R_3}{2}, 2\log(2(L+1)), 4l^2 r^2, d\right\}.$$

*Then, with probability at least $1 - 2L\exp(-\frac{m}{2})$ with regard to the initialization,*

$$R(T) \leq \sum_{t=1}^{T} F_t(\boldsymbol{w}_t) \leq \frac{9\epsilon}{4} + \frac{g^2(\epsilon/T)\epsilon}{C_0\eta}, \quad \forall s \in [T].$$

**Theorem 6** (Detailed version of Theorem 2). *Let $\{\boldsymbol{w}_s\}_{s \in [T]}$ be generated by Algorithm 1 with* (4). *Let Assumptions 1, 3, 4 and 6 hold. For any $\epsilon_0 > 0, T \geq 1$ and $0 < \beta_1 \leq \beta_2 < 1$, we define*

$$D_{\ell_2}^2 = \frac{9}{4} + g^2\left(\frac{\epsilon_0}{T}\right) + \frac{10L_f}{\epsilon_0(1-\beta_1)\beta_2} + \frac{32L_f L}{\epsilon_0(1-\beta_1)^2(1-\beta_2)^2} + \frac{4+\sqrt{2}}{\epsilon_0},$$

*Let $R_1, R_2, R_3$ be the same forms as* (15) *with $r = 4D_{\ell_2} + 12\sqrt{2}$. If $C_0\eta = \frac{\tilde{C}_0}{\sqrt{T}}$, $\epsilon = \frac{\epsilon_0}{\sqrt{T}}$ for some positive constant $\tilde{C}_0 \leq \min\{1, \frac{1}{R_2}\}$, and*

$$m \geq \max\left\{D_{\ell_2}^2 R_3\sqrt{L_f T}, D_{\ell_2} R_2\sqrt{\tilde{C}_0}, \left(\frac{D_{\ell_2}^2 R_2}{\sqrt{\tilde{C}_0}}\right)T, 2\log(2(L+1)), 4l^2 r^2, d\right\},$$

*then, with probability at least $1 - 2(L+1)\exp(-\frac{m}{2})$ with regard to the initialization,*

$$R(T) \leq \sum_{t=1}^{T} F_t(\boldsymbol{w}_t) \leq \frac{\epsilon_0 D_{\ell_2}^2}{\tilde{C}_0}, \quad \forall s \in [T].$$

**Theorem 7** (Detailed version of Theorem 3). *Let $\{\boldsymbol{w}_s\}_{s \in [T]}$ be generated by Algorithm 1 with (4), $\Phi(\boldsymbol{w}, \boldsymbol{x})$ be in (8) with $\sigma(0) = 0$. Let $f$ be logistic loss, and Assumptions 1, 4 and 6 hold. For any $\epsilon > 0, T \geq 1$, we define $D_{\ell_2} = \frac{3}{2} + g(\frac{\epsilon}{T})$, $r = 4D_{\ell_2} + 12\sqrt{2} + \frac{2}{l}$, $R_1' = R(lr)^L$,*

$$R_2' = \sqrt{L}R_1'(lr)^{L+1}, \quad R_3' = \sqrt{L}(lr)^{2L+1}\left[(lr)^{L-1}\tilde{l}r^2 R_1' + 1\right]. \tag{16}$$

*If $0 < \beta_1 \leq \beta_2 < 1$, $C_0\eta \leq 1$, and*

$$m \geq \max\left\{\frac{81(D_{\ell_2}^2 R_3')^2}{4}, (8D_{\ell_2}\beta_1 R_2')^2, \left(\frac{8D_{\ell_2}^2 R_2'\sqrt{1-\beta_2}}{C_0\eta(1-\sqrt{\beta_2})}\right)^2, \frac{4C_0\eta R_2'}{\epsilon(1-\beta_1)^2}, 2\log\left(2(L+1)\right)\right\}, \tag{17}$$

*then, with probability at least $1 - 2(L+1)\exp(-\frac{m}{2})$ with regard to the initialization,*

$$R(T) \leq \sum_{t=1}^{T} F_t(\boldsymbol{w}_t) \leq \frac{9\epsilon}{4} + \frac{g^2(\epsilon/T)\epsilon}{C_0\eta}, \quad \forall s \in [T].$$

### B.2 PROOF OF THEOREM 2 (THEOREM 6)

First, we set $\boldsymbol{w} = \boldsymbol{w}^{(\epsilon_0/T)} \in \mathcal{X}$ satisfying Assumption 4.

**Case $k = 1$.** With $D_{\ell_2}$ and $r$ defined in Theorem 2, we use Assumption 4, Lemma 2, and the norm inequality to get that $\|\boldsymbol{w} - \boldsymbol{w}_1\| \leq g\left(\frac{\epsilon_0}{T}\right) \leq D_{\ell_2}$ and $\|\boldsymbol{W}_1^{(i)}\|, \|\boldsymbol{W}^{(i)}\| \leq \frac{r}{4}, \forall i \in [L], \|\boldsymbol{a}^\top\| \leq r$.

**Case $k = t + 1$.** Suppose that (12) holds for some $t \in [T]$ with $D_{\ell_2}$, $r$ and $\boldsymbol{w}$ defined above. Also, Assumption 7 is satisfied under the induction assumption. Therefore, Lemmas 5 to 8 hold under smooth losses with $D = D_{\ell_2}$ and $r$ defined in Theorem 2, and subsequent terms $R_1, R_2, R_3$. Combining Proposition 2 with $\phi(\cdot) = 0$, and Lemmas 5 to 8, we get that with $C_0\eta = \tilde{C}_0/\sqrt{T}$ and $\epsilon = \epsilon_0/\sqrt{T}$,

$$\|\boldsymbol{w} - \boldsymbol{w}_{t+1}\|^2 \leq \frac{\tilde{C}_0}{\epsilon_0}\left(\sum_{s=1}^{t}\frac{9F_s(\boldsymbol{w})}{4} - F_s(\boldsymbol{w}_s)\right) + \|\boldsymbol{w} - \boldsymbol{w}_1\|^2 + \frac{8\tilde{C}_0(D_{\ell_2}\beta_1)^2 L_f}{\epsilon_0(1-\beta_1)}\left(\frac{R_2}{m}\right)^2$$

$$+ \left(\frac{D_{\ell_2}^2 R_2}{m}\right)^2 \frac{2L_f T t}{\epsilon_0\tilde{C}_0\beta_2} + \frac{32\tilde{C}_0^3 L_f L R_2^2 t}{\epsilon_0(1-\beta_1)^2(1-\beta_2)^2 T} + \frac{\tilde{C}_0(4+\sqrt{2})}{\epsilon_0}. \tag{18}$$

Similarly, using $\boldsymbol{\Lambda}_t \succeq (\epsilon/\eta)\mathbf{I}_{d' \times d'}$ and the learning rate setup, we have

$$\frac{\|\boldsymbol{w} - \boldsymbol{w}_{t+1}\|_{\boldsymbol{\Lambda}_t}^2}{2C_0} \geq \frac{\epsilon\|\boldsymbol{w} - \boldsymbol{w}_{t+1}\|^2}{2C_0\eta} = \frac{\epsilon_0\|\boldsymbol{w} - \boldsymbol{w}_{t+1}\|^2}{2\tilde{C}_0}.$$

Using $\sum_{s=1}^{t} F_s(\boldsymbol{w}) \leq \sum_{s=1}^{T} F_s(\boldsymbol{w}) \leq \epsilon_0$, the learning rate and the width requirement (detailed in Theorem 6 in Section B), one can get from (18) that for any $i \in [L]$,

$$\|\boldsymbol{W}^{(i)} - \boldsymbol{W}_{t+1}^{(i)}\|^2 \leq \|\boldsymbol{w} - \boldsymbol{w}_{t+1}\|^2 \leq D_{\ell_2}^2 - \frac{\tilde{C}_0}{\epsilon_0}\sum_{s=1}^{t} F_s(\boldsymbol{w}_s) \leq D_{\ell_2}^2. \tag{19}$$

It's then derived that $\|\boldsymbol{w} - \boldsymbol{w}_{t+1}\| \leq D_{\ell_2}$. Using the norm inequality, one can also easily derive that $\|\boldsymbol{W}_{t+1}^{(i)}\| \leq \frac{r}{2}, \forall i \in [L]$. With the induction, we prove that (12) holds for any $t \in [T]$ with $D_{\ell_2}$ and $r$ defined in Theorem 2. Then, we derive the desired regret bound (7) through (19).

**Remark 4.** *We notice that in the proof of Lemmas 7 and 8, the upper bounds are $\mathcal{O}(\sum_{i=1}^{d'}\sqrt{(1-\beta_2)\sum_{s=1}^{t}\beta_2^{t-s}g_{s,i}^2})$ order, such as (40) and (43). The similar terms also emerge in the proof of Lemmas 18 and 19. Note that when $\boldsymbol{g}_s$ is sparse, $\mathcal{O}(\sum_{i=1}^{d'}\sqrt{(1-\beta_2)\sum_{s=1}^{t}\beta_2^{t-s}g_{s,i}^2})$ can be smaller than $\mathcal{O}(\sqrt{T})$ obtained by GD/SGD as indicated by (Duchi et al., 2011).*

### B.3 Proof Challenges and Solutions

We list some of the unique challenges for deriving regret bounds of RMSProp and AdamNC in neural network optimization.

**Challenge (I). Non-monotonic step-sizes.** When considering GD and AdaGrad/AMSGrad, the summation term $(*)$ in (11) forms a telescoping term due to the constant or non-increasing step-sizes. In comparison, step-sizes of RMSProp and AdamNC are non-monotonic given only constants $C_0\eta$ and $\beta_1, \beta_2$, leading to a challenge in controlling $(*)$.

**Solution.** We propose new bounds in Lemmas 7 and 18, as well as other related terms, to control $(*)$. Based on the EMA in AdamNC, each distance term is roughly $B_{\psi_s}(\boldsymbol{w}, \boldsymbol{w}_s) \sim \mathcal{O}(D_{\ell_2}^2 \|\boldsymbol{g}_s\|)$ and the summation is roughly $\mathcal{O}(D_{\ell_2}^2 \sum_{s=1}^{t} \|\boldsymbol{g}_s\|)$, which can be further bounded through gradient upper bounds in Lemmas 3, 4 and 15.

**Challenge (II). Unbounded gradient and iterate norms.** Many existing regret bounds for Adam or other adaptive methods in online-convex optimization rely on the assumptions of bounded gradients and iteration, i.e., $\|\boldsymbol{g}_t\| \leq G$ and $\|\boldsymbol{w}_t - \boldsymbol{w}^*\| \leq D$ for any $t \in [T]$, such as (Duchi et al., 2011; Alacaoglu et al., 2020). In comparison, we rely on neither of the assumptions, which may be closer to the practical setting.

In neural network optimization, we also note that Taheri & Thrampoulidis (2024) and Taheri et al. (2025) use the similar induction argument to bound the iterate norm of GD. The key differences lie in the more complicated adaptive step-sizes of AdamNC compared to constant ones in GD. Also, the RHS of (11) can result in $\mathcal{O}(d') \sim \mathcal{O}(m^2)$ factor when bounding $\sum_{s=1}^{t} \|\boldsymbol{m}_s\|_{\boldsymbol{\Lambda}_s^{-1}}^2$ since RMSProp and AdamNC are both coordinate-wise algorithms (Défossez et al., 2022).

**Solution.** We tackle this challenge through the property of gradient and Hessian norms in Lemma 3 and a delicate induction argument. Under Assumption 7, we can locally bound RHS of (11) with a dominated order of $\mathcal{O}\left(\left(C_0\eta + \frac{1}{m}\right) \sum_{s=1}^{t} F_s(\boldsymbol{w}_s) - \|\boldsymbol{w}_{t+1} - \boldsymbol{w}^*\|_{\boldsymbol{\Lambda}_t}^2\right)$ while the LHS is lower bounded through the weakly-convex-like property in Lemma 5. Hence, with proper setups of $C_0\eta$ and $m$, we can use an induction argument to bound $\|\boldsymbol{w}_t - \boldsymbol{w}^*\|$.

To handle the additional $\mathcal{O}(d')$ factor, we provdie two solutions. For model (3), we derive a better $\mathcal{O}(\frac{1}{m})$ bound for the gradient norm instead of the $\mathcal{O}(1)$ in literature. For model (8), we can only derive $\mathcal{O}(\frac{1}{\sqrt{m}})$ but further derives a dimension-free bound for $\sum_{s=1}^{t} \|\boldsymbol{m}_s\|_{\boldsymbol{\Lambda}_s^{-1}}^2$. Both solutions can ensure that $\sum_{s=1}^{t} \|\boldsymbol{m}_s\|_{\boldsymbol{\Lambda}_s^{-1}}^2$ is less than $\frac{1}{4} \sum_{s=1}^{t} F_s(\boldsymbol{w}_s)$. The details can be found in Lemmas 8 and 19.

## C    Properties of deep neural networks

In this section, we will prove Lemma 1 regarding the gradient and Hessian norm of models. Fixed any point $\boldsymbol{w} = \left[\text{vec}\left(\boldsymbol{W}^{(1)}\right); \text{vec}\left(\boldsymbol{W}^{(2)}\right); \cdots; \text{vec}\left(\boldsymbol{W}^{(L)}\right)\right] \in \mathbb{R}^{d'}$, we follow the definition of $\boldsymbol{\varphi}^{(0)} = \boldsymbol{x}, \boldsymbol{\varphi}^{(i)} = \frac{1}{\sqrt{m}}\boldsymbol{\sigma}\left(\boldsymbol{W}^{(i)}\boldsymbol{\varphi}^{(i-1)}\right)$ in (3) and define

$$\boldsymbol{\psi}^{(i)} = \frac{\partial\Phi(\boldsymbol{w}, \boldsymbol{x})}{\partial\boldsymbol{\varphi}^{(i)}} \in \mathbb{R}^m, \quad \Sigma^{(i)} = \text{Diag}\left[\boldsymbol{\sigma}'\left(\boldsymbol{W}^{(i)}\boldsymbol{\varphi}^{(i-1)}\right)\right] \in \mathbb{R}^{m\times m} \quad \forall i \in [L], \qquad (20)$$

where $\boldsymbol{\sigma}'(\boldsymbol{z}) = [\sigma'(z_1), \cdots, \sigma'(z_m)], \forall \boldsymbol{z} \in \mathbb{R}^m$ denotes the entry-wise derivative of activation. For another fixed point $\tilde{\boldsymbol{w}} = \left[\text{vec}\left(\tilde{\boldsymbol{W}}^{(1)}\right); \text{vec}\left(\tilde{\boldsymbol{W}}^{(2)}\right); \cdots; \text{vec}\left(\tilde{\boldsymbol{W}}^{(L)}\right)\right] \in \mathbb{R}^{d'}$, we have

$$\tilde{\boldsymbol{\varphi}}^{(0)} = \boldsymbol{x} \in \mathbb{R}^d, \quad \tilde{\boldsymbol{\varphi}}^{(i)} = \frac{1}{\sqrt{m}}\boldsymbol{\sigma}\left(\tilde{\boldsymbol{W}}^{(i)}\tilde{\boldsymbol{\varphi}}^{(i-1)}\right), \forall i \in [L], \Phi(\tilde{\boldsymbol{w}}, \boldsymbol{x}) = \frac{1}{\sqrt{m}}\boldsymbol{a}^\top\tilde{\boldsymbol{\varphi}}^{(L)},$$

$$\tilde{\boldsymbol{\psi}}^{(i)} = \frac{\partial\Phi(\tilde{\boldsymbol{w}}, \boldsymbol{x})}{\partial\tilde{\boldsymbol{\varphi}}^{(i)}} \in \mathbb{R}^m, \quad \tilde{\Sigma}^{(i)} = \text{Diag}\left[\boldsymbol{\sigma}'\left(\tilde{\boldsymbol{W}}^{(i)}\tilde{\boldsymbol{\varphi}}^{(i-1)}\right)\right] \in \mathbb{R}^{m\times m} \quad \forall i \in [L].$$

Then, we have the following results for some terms related to the deep neural network.

**Lemma 9.** *Let* $\boldsymbol{w} \in \mathbb{R}^{d'}$ *such that* $\|\boldsymbol{W}^{(i)}\| \leq r, \forall i \in [L]$. *When* $\boldsymbol{\varphi}^{(i)}, \Sigma^{(i)}, \boldsymbol{\psi}^{(i)}$ *are defined in* (3) *and* $\|\boldsymbol{a}\| \leq r$, *for any* $i \in [L]$, *with* $R_1 = R + 2|\sigma(0)|$, *it holds that*

$$\left\|\boldsymbol{\varphi}^{(i)}\right\| \leq R_1, \quad \left\|\Sigma^{(i)}\right\| \leq l, \quad \left\|\boldsymbol{\psi}^{(i)}\right\| \leq \frac{r}{\sqrt{m}}.$$

*Proof.* For the model in (3), using the definition of $\boldsymbol{\varphi}^{(i)}$, $\|\boldsymbol{W}^{(i)}\| \leq r$ and $\sqrt{m} \geq 2lr$, we get that for any $i \in [L]$,

$$\left\|\boldsymbol{\varphi}^{(i)}\right\| = \frac{1}{\sqrt{m}}\left\|\boldsymbol{\sigma}\left(\boldsymbol{W}^{(i)}\boldsymbol{\varphi}^{(i-1)}\right)\right\| \leq \frac{1}{\sqrt{m}}\left\|\boldsymbol{\sigma}\left(\boldsymbol{W}^{(i)}\boldsymbol{\varphi}^{(i-1)}\right) - \boldsymbol{\sigma}(\boldsymbol{0}_m)\right\| + \frac{\|\boldsymbol{\sigma}(\boldsymbol{0}_m)\|}{\sqrt{m}}$$

$$\leq \frac{l}{\sqrt{m}}\left\|\boldsymbol{W}^{(i)}\boldsymbol{\varphi}^{(i-1)}\right\| + \frac{\|\boldsymbol{\sigma}(\boldsymbol{0}_m)\|}{\sqrt{m}} < \frac{lr}{\sqrt{m}}\left\|\boldsymbol{\varphi}^{(i-1)}\right\| + \frac{\|\boldsymbol{\sigma}(\boldsymbol{0}_m)\|}{\sqrt{m}}$$

$$\leq \left(\frac{lr}{\sqrt{m}}\right)^i \|\boldsymbol{x}\| + \frac{\|\boldsymbol{\sigma}(\boldsymbol{0}_m)\|}{\sqrt{m}} \sum_{j=0}^{i-1}\left(\frac{lr}{\sqrt{m}}\right)^j \leq R\left(\frac{lr}{\sqrt{m}}\right)^i + \frac{\|\boldsymbol{\sigma}(\boldsymbol{0}_m)\|}{\left(1 - \frac{lr}{\sqrt{m}}\right)\sqrt{m}}$$

$$\leq R + \frac{2\sqrt{m}|\sigma(0)|}{\sqrt{m}} = R_1,$$

where we use that $\sigma$ is $l$-Lipschitz continuous in the second inequality, and $\|\boldsymbol{x}\| \leq R$ in the third line. Using the definition of $\Sigma^{(i)}$ and that $\sigma$ is $l$-Lipschitz continuous, we get that

$$\left\|\Sigma^{(i)}\right\| = \left\|\boldsymbol{\sigma}'\left(\boldsymbol{W}^{(i)}\boldsymbol{\varphi}^{(i-1)}\right)\right\|_\infty \leq l.$$

Using the definition of $\boldsymbol{\psi}^{(i)}$, we get that

$$\boldsymbol{\psi}^{(i)} = \frac{1}{\sqrt{m}}\left(\boldsymbol{W}^{(i)}\right)^\top \Sigma^{(i)}\boldsymbol{\psi}^{(i+1)}, \quad \forall i \in [L-1], \quad \boldsymbol{\psi}^{(L)} = \frac{1}{\sqrt{m}}\boldsymbol{a}.$$

Then, using the norm inequality, $\|\boldsymbol{W}^{(i)}\| = \|\left(\boldsymbol{W}^{(i)}\right)^\top\|$, and $\sqrt{m} \geq 2lr$, for any $i \in [L-1]$,

$$\left\|\boldsymbol{\psi}^{(i)}\right\| \leq \frac{1}{\sqrt{m}}\left\|\boldsymbol{W}^{(i)}\right\|\left\|\Sigma^{(i)}\boldsymbol{\psi}^{(i+1)}\right\| \leq \frac{1}{\sqrt{m}}\left\|\boldsymbol{W}^{(i)}\right\|\left\|\Sigma^{(i)}\right\|\left\|\boldsymbol{\psi}^{(i+1)}\right\|$$

$$< \frac{lr}{\sqrt{m}}\left\|\boldsymbol{\psi}^{(i+1)}\right\| \leq \frac{1}{\sqrt{m}}\left(\frac{lr}{\sqrt{m}}\right)^{L-i}\|\boldsymbol{a}^\top\| = \frac{r}{\sqrt{m}}\left(\frac{lr}{\sqrt{m}}\right)^{L-i} \leq \frac{r}{\sqrt{m}}. \tag{21}$$

We further get that

$$\left\|\boldsymbol{\psi}^{(L)}\right\| = \frac{\|\boldsymbol{a}\|}{\sqrt{m}} < \frac{r}{\sqrt{m}}.$$

Combining the above, we obtain the upper bound for $\|\boldsymbol{\psi}^{(i)}\|$. $\qquad\square$

*Proof of Lemma 1.* By the chain rule, we have for any $i \in [L]$,

$$\frac{\partial\Phi(\boldsymbol{w}, \boldsymbol{x})}{\partial\boldsymbol{W}^{(i)}} = \frac{1}{\sqrt{m}}\Sigma^{(i)}\boldsymbol{\psi}^{(i)}\left(\boldsymbol{\varphi}^{(i-1)}\right)^\top \in \mathbb{R}^{m \times m}.$$

Then, using the norm inequality, we get that for any $i \in [L]$,

$$\left\|\frac{\partial\Phi(\boldsymbol{w}, \boldsymbol{x})}{\partial\boldsymbol{W}^{(i)}}\right\|_F \leq \frac{1}{\sqrt{m}}\left\|\Sigma^{(i)}\right\|\left\|\boldsymbol{\psi}^{(i)}\left(\boldsymbol{\varphi}^{(i-1)}\right)^\top\right\|_F \leq \frac{1}{\sqrt{m}}\left\|\Sigma^{(i)}\right\|\left\|\boldsymbol{\psi}^{(i)}\right\|\left\|\boldsymbol{\varphi}^{(i-1)}\right\|.$$

Using $\sqrt{m} \geq 2lr$ and Lemma 9 since $\|\boldsymbol{W}^{(i)}\| \leq r/2$, we get that

$$\max_{i \in [L]}\left\|\frac{\partial\Phi(\boldsymbol{w}, \boldsymbol{x})}{\partial\boldsymbol{W}^{(i)}}\right\|_F < \max_{i \in [L]}\frac{l}{\sqrt{m}} \cdot \frac{r}{\sqrt{m}} \cdot R_1 = \frac{lrR_1}{m}.$$

Note that $w$ is the vectorization of all trainable weights. Hence, $\nabla_{\boldsymbol{w}} \Phi(\boldsymbol{w}, \boldsymbol{x})$ is a vector and its $\ell_2$-norm is bounded by

$$\|\nabla_{\boldsymbol{w}} \Phi(\boldsymbol{w}, \boldsymbol{x})\| = \sqrt{\sum_{i=1}^{L} \left\| \frac{\partial \Phi(\boldsymbol{w}, \boldsymbol{x})}{\partial \boldsymbol{W}^{(i)}} \right\|_F^2} < \frac{lr R_1 \sqrt{L}}{m} = \frac{R_2}{m}.$$

For the upper bound of the Hessian norm, we have the following proof. We will calculate the variation of the gradient. For any $\boldsymbol{w}$ such that $\|\boldsymbol{W}^{(i)}\| \leq r/2, \forall i \in [L]$, we let $\tilde{\boldsymbol{w}} \in \mathcal{B}(\boldsymbol{w}, \Delta)$, meaning that $\|\boldsymbol{w} - \tilde{\boldsymbol{w}} < \Delta$. In addition, we let $\Delta \leq r/2$, which leads to that

$$\|\tilde{\boldsymbol{W}}^{(i)}\| \leq \|\boldsymbol{W}^{(i)} - \tilde{\boldsymbol{W}}^{(i)}\| + \|\boldsymbol{W}^{(i)}\| \leq \|\boldsymbol{w} - \tilde{\boldsymbol{w}}\| + \|\boldsymbol{W}^{(i)}\| \leq r, \forall i \in [L].$$

Using the definition and the triangle inequality, we have for each layer $i \in [L]$,

$$\begin{aligned}
\left\| \frac{\partial \Phi(\boldsymbol{w}, \boldsymbol{x})}{\partial \boldsymbol{W}^{(i)}} - \frac{\partial \Phi(\tilde{\boldsymbol{w}}, \boldsymbol{x})}{\partial \tilde{\boldsymbol{W}}^{(i)}} \right\|_F &= \frac{1}{\sqrt{m}} \left\| \Sigma^{(i)} \boldsymbol{\psi}^{(i)} \left( \boldsymbol{\varphi}^{(i-1)} \right)^\top - \tilde{\Sigma}^{(i)} \tilde{\boldsymbol{\psi}}^{(i)} \left( \tilde{\boldsymbol{\varphi}}^{(i-1)} \right)^\top \right\|_F \\
&\leq \frac{1}{\sqrt{m}} \left\| \left( \Sigma^{(i)} - \tilde{\Sigma}^{(i)} \right) \boldsymbol{\psi}^{(i)} \left( \boldsymbol{\varphi}^{(i-1)} \right)^\top \right\|_F \\
&\quad + \frac{1}{\sqrt{m}} \left\| \tilde{\Sigma}^{(i)} \left( \boldsymbol{\psi}^{(i)} - \tilde{\boldsymbol{\psi}}^{(i)} \right) \left( \boldsymbol{\varphi}^{(i-1)} \right)^\top \right\|_F \\
&\quad + \frac{1}{\sqrt{m}} \left\| \tilde{\Sigma}^{(i)} \tilde{\boldsymbol{\psi}}^{(i)} \left( \boldsymbol{\varphi}^{(i-1)} - \tilde{\boldsymbol{\varphi}}^{(i-1)} \right)^\top \right\|_F.
\end{aligned} \tag{22}$$

Using the definition of $\boldsymbol{\varphi}^{(i)}$ from (3) and the norm inequality, we get that for any $i \in [L]$,

$$\begin{aligned}
\left\| \boldsymbol{\varphi}^{(i)} - \tilde{\boldsymbol{\varphi}}^{(i)} \right\| &= \frac{1}{\sqrt{m}} \left\| \boldsymbol{\sigma} \left( \boldsymbol{W}^{(i)} \boldsymbol{\varphi}^{(i-1)} \right) - \boldsymbol{\sigma} \left( \tilde{\boldsymbol{W}}^{(i)} \tilde{\boldsymbol{\varphi}}^{(i-1)} \right) \right\| \leq \frac{l}{\sqrt{m}} \left\| \boldsymbol{W}^{(i)} \boldsymbol{\varphi}^{(i-1)} - \tilde{\boldsymbol{W}}^{(i)} \tilde{\boldsymbol{\varphi}}^{(i-1)} \right\| \\
&\leq \frac{l}{\sqrt{m}} \left\| \boldsymbol{W}^{(i)} \left( \boldsymbol{\varphi}^{(i-1)} - \tilde{\boldsymbol{\varphi}}^{(i-1)} \right) \right\| + \frac{l}{\sqrt{m}} \left\| \left( \boldsymbol{W}^{(i)} - \tilde{\boldsymbol{W}}^{(i)} \right) \tilde{\boldsymbol{\varphi}}^{(i-1)} \right\| \\
&\leq \frac{lr}{\sqrt{m}} \left\| \boldsymbol{\varphi}^{(i-1)} - \tilde{\boldsymbol{\varphi}}^{(i-1)} \right\| + \frac{l\Delta R_1}{\sqrt{m}} \leq \frac{l\Delta R_1}{\sqrt{m}} \sum_{j=0}^{i-1} \left( \frac{lr}{\sqrt{m}} \right)^j,
\end{aligned} \tag{23}$$

where we use that $\sigma$ is $l$-Lipschitz continuous in the first inequality, Lemma 9 in the third inequality, and $\boldsymbol{\varphi}^{(0)} = \tilde{\boldsymbol{\varphi}}^{(0)} = \boldsymbol{x}$ in the last one. With $\sqrt{m} \geq 2lr$, we thereby derive that

$$\max_{i \in [L]} \left\| \boldsymbol{\varphi}^{(i)} - \tilde{\boldsymbol{\varphi}}^{(i)} \right\| \leq \frac{2l\Delta R_1}{\sqrt{m}}.$$

Using that $\sigma'(\cdot)$ is $\tilde{l}$-Lipschitz continuous and $\Sigma^{(i)}$ is a digonal matrix, we get that

$$\begin{aligned}
\left\| \Sigma^{(i)} - \tilde{\Sigma}^{(i)} \right\| &= \left\| \sigma' \left( \boldsymbol{W}^{(i)} \boldsymbol{\varphi}^{(i-1)} \right) - \sigma' \left( \tilde{\boldsymbol{W}}^{(i)} \tilde{\boldsymbol{\varphi}}^{(i-1)} \right) \right\|_\infty \leq \tilde{l} \left\| \boldsymbol{W}^{(i)} \boldsymbol{\varphi}^{(i-1)} - \tilde{\boldsymbol{W}}^{(i)} \tilde{\boldsymbol{\varphi}}^{(i-1)} \right\|_\infty \\
&\leq \tilde{l} \left\| \boldsymbol{W}^{(i)} \boldsymbol{\varphi}^{(i-1)} - \tilde{\boldsymbol{W}}^{(i)} \tilde{\boldsymbol{\varphi}}^{(i-1)} \right\| \leq 2\tilde{l} \Delta R_1,
\end{aligned}$$

where the last inequality uses a similar deduction as (23). Using the definition of $\boldsymbol{\psi}^{(i)}$ from (20) and the norm inequality, we get that for any $i \in [L-1]$,

$$
\begin{aligned}
\left\| \boldsymbol{\psi}^{(i)} - \tilde{\boldsymbol{\psi}}^{(i)} \right\| &\leq \frac{1}{\sqrt{m}} \left( \left\| \left( \Sigma^{(i)} - \tilde{\Sigma}^{(i)} \right) \boldsymbol{W}^{(i)} \boldsymbol{\psi}^{(i+1)} \right\| + \left\| \tilde{\Sigma}^{(i)} \left( \boldsymbol{W}^{(i)} - \tilde{\boldsymbol{W}}^{(i)} \right) \boldsymbol{\psi}^{(i+1)} \right\| \right) \\
&\quad + \frac{1}{\sqrt{m}} \left\| \tilde{\Sigma}^{(i)} \tilde{\boldsymbol{W}}^{(i)} \left( \boldsymbol{\psi}^{(i+1)} - \tilde{\boldsymbol{\psi}}^{(i+1)} \right) \right\| \\
&\leq \frac{1}{\sqrt{m}} \left( \frac{2\tilde{l}r^2 \Delta R_1}{\sqrt{m}} + \frac{l\Delta r}{\sqrt{m}} + lr \left\| \boldsymbol{\psi}^{(i+1)} - \tilde{\boldsymbol{\psi}}^{(i+1)} \right\| \right) \\
&\leq \left( \frac{2\tilde{l}r^2 \Delta R_1}{m} + \frac{lr\Delta}{m} \right) \sum_{j=0}^{L-i-1} \left( \frac{lr}{\sqrt{m}} \right)^j + \left( \frac{lr}{\sqrt{m}} \right)^{L-i} \left\| \boldsymbol{\psi}^{(L)} - \tilde{\boldsymbol{\psi}}^{(L)} \right\| \\
&= \frac{2 \left( 2\tilde{l}r^2 R_1 + lr \right) \Delta}{m} + \frac{1}{\sqrt{m}} \left( \frac{lr}{\sqrt{m}} \right)^{L-i} \| \boldsymbol{a} - \boldsymbol{a} \| \\
&\leq \frac{2 \left( 2\tilde{l}r^2 R_1 + lr \right) \Delta}{\sqrt{m}}.
\end{aligned}
$$

For $i = L$, we get that

$$
\left\| \boldsymbol{\psi}^{(L)} - \tilde{\boldsymbol{\psi}}^{(L)} \right\| = \frac{1}{\sqrt{m}} \| \boldsymbol{a} - \boldsymbol{a} \| = 0.
$$

Combining the above, we get that $\max_{i \in [L]} \left\| \boldsymbol{\psi}^{(i)} - \tilde{\boldsymbol{\psi}}^{(i)} \right\| \leq \frac{2(2\tilde{l}r^2 R_1 + lr)\Delta}{\sqrt{m}}$. Based on the above results and (22), we then derive that for any $i \in [L]$,

$$
\begin{aligned}
\left\| \frac{\partial \Phi(\boldsymbol{w}, \boldsymbol{x})}{\partial \boldsymbol{W}^{(i)}} - \frac{\partial \Phi(\tilde{\boldsymbol{w}}, \boldsymbol{x})}{\partial \tilde{\boldsymbol{W}}^{(i)}} \right\|_F &\leq \frac{2\tilde{l}R_1^2 r \Delta}{m} + \frac{2lR_1 \left( 2\tilde{l}r^2 R_1 + lr \right) \Delta}{m} + \frac{2l^2 r R_1 \Delta}{m\sqrt{m}} \\
&\leq \frac{2R_1 \left( \tilde{l}R_1 r (1 + 2lr) + 2l^2 r \right) \Delta}{m}.
\end{aligned}
$$

Then, combining all the layers, we get that for any $\tilde{\boldsymbol{w}} \in \mathcal{B}(\boldsymbol{w}, \Delta)$ with $\Delta \leq \frac{r}{2}$,

$$
\begin{aligned}
\| \nabla_{\boldsymbol{w}} \Phi(\boldsymbol{w}, \boldsymbol{x}) - \nabla_{\tilde{\boldsymbol{w}}} \Phi(\tilde{\boldsymbol{w}}, \boldsymbol{x}) \|_F &\leq \sqrt{ \sum_{i=1}^{L} \left\| \frac{\partial \Phi(\boldsymbol{w}, \boldsymbol{x})}{\partial \boldsymbol{W}^{(i)}} - \frac{\partial \Phi(\tilde{\boldsymbol{w}}, \boldsymbol{x})}{\partial \tilde{\boldsymbol{W}}^{(i)}} \right\|_F^2 } \\
&\leq \frac{\Delta \sqrt{L} \left[ 2R_1 \left( \tilde{l}R_1 r (1 + 2lr) + 2l^2 r \right) \right]}{m} \\
&= \frac{\Delta R_3}{m}. \quad (24)
\end{aligned}
$$

Using the definition of Hessian matrix, we get that for any unit vector $\boldsymbol{v} \in \mathbb{R}^{d'}$,

$$
\nabla_{\boldsymbol{w}}^2 \Phi(\boldsymbol{w}, \boldsymbol{x}) \boldsymbol{v} = \lim_{t \to 0} \frac{\nabla \Phi_{\boldsymbol{w}}(\boldsymbol{w} + t\boldsymbol{v}, \boldsymbol{x}) - \nabla_{\boldsymbol{w}} \Phi(\boldsymbol{w}, \boldsymbol{x})}{t}.
$$

Then, taking the $\ell_2$-norm on both sides and using (24), we get that for any unit vector $\boldsymbol{v} \in \mathbb{R}^{d'}$,

$$
\left\| \nabla_{\boldsymbol{w}}^2 \Phi(\boldsymbol{w}, \boldsymbol{x}) \boldsymbol{v} \right\| = \lim_{t \to 0} \frac{\| \nabla_{\boldsymbol{w}} \Phi(\boldsymbol{w} + t\boldsymbol{v}, \boldsymbol{x}) - \nabla_{\boldsymbol{w}} \Phi(\boldsymbol{w}, \boldsymbol{x}) \|}{t} = \frac{R_3}{m},
$$

Recalling the definition of the induced $\ell_2$-norm, we get that

$$
\left\| \nabla_{\boldsymbol{w}}^2 \Phi(\boldsymbol{w}, \boldsymbol{x}) \right\| = \sup_{\|\boldsymbol{v}\|=1} \left\| \nabla_{\boldsymbol{w}}^2 \Phi(\boldsymbol{w}, \boldsymbol{x}) \boldsymbol{v} \right\| \leq \frac{R_3}{m}.
$$

$\square$

# D  PROOF OF RESULTS IN SECTIONS 3 AND 4

In this section, we provide the detailed proofs for the results in Sections 3 and 4 and some complementary lemmas for the objective function $F_t(\boldsymbol{w})$.

## D.1  DETAILED PROOF FOR RESULTS IN SECTION 3

We provide the following example from (Taheri & Thrampoulidis, 2024, Remark 1), satisfying Assumption 4 under the linear separable data.

**Example 1.** *Suppose that there exists $\boldsymbol{v}^* \in \mathbb{R}^d, \|\boldsymbol{v}^*\| \leq 1$ such that $y_i \langle \boldsymbol{v}^*, \boldsymbol{x}_i \rangle \geq \gamma > 0, \forall i \in [n]$. If $L = 1$ in (3), $f$ is the logistic loss function, $\sigma$ is tanh activation, $\boldsymbol{w}_0 = \boldsymbol{0}_{d'}$ and $m \geq 4 \log^2(1/\varepsilon)$, then Assumption 4 holds with $g(1/\varepsilon) = 2 \log(1/\varepsilon)/\gamma$.*

We also introduce specific examples satisfying Assumption 5, which is related to the noisy XOR data distribution (Wei et al., 2019), an uniform distribution on the following $2^d$ points:

$$\boldsymbol{x} = (x_i)_i \in \mathbb{R}^{d'}, y \in \mathbb{R}, \quad \text{where} \quad (x_1, x_2, y) \times (x_3, \cdots, x_d) \in$$

$$\left\{ \left( \frac{1}{\sqrt{d-1}}, 0, 1 \right), \left( 0, \frac{1}{\sqrt{d-1}}, -1 \right), \left( \frac{-1}{\sqrt{d-1}}, 0, 1 \right), \left( 0, \frac{-1}{\sqrt{d-1}}, -1 \right) \right\}$$

$$\times \left\{ \frac{-1}{\sqrt{d-1}}, \frac{1}{\sqrt{d-1}} \right\}^{d-2}, \tag{25}$$

and $\times$ denotes the Cartesian product. The following results show that NTK-separability is satisfied when $\{(\boldsymbol{x}_i, y_i)\}_{i \in [n]}$ is sampled from this data distribution.

**Example 2.** *(Ji & Telgarsky, 2020, Proposition 5.3) Define the Hilbert space $\mathcal{H} :=$ $\left\{ \boldsymbol{v} : \mathbb{R}^{d'} \to \mathbb{R}^{d'} \mid \int \|\boldsymbol{v}(\boldsymbol{z})\|^2 d\mu_{\mathcal{N}}(\boldsymbol{z}) < \infty \right\}$ where $\mu_{\mathcal{N}}$ is the Gaussian measure. If $\sigma$ is ReLU activation, $d \geq 3$ and $\Phi(\cdot, \cdot)$ is as in (3), then, for any $(\boldsymbol{x}, y)$ drawn from (25), there exists $\boldsymbol{w}^* \in \mathcal{H}$ such that $y \langle \nabla_{\boldsymbol{w}_1} \Phi(\boldsymbol{w}_1, \boldsymbol{x}), \boldsymbol{w}^* \rangle_{\mathcal{H}} \geq \frac{1}{60d}$.*

**Example 3.** *(Taheri & Thrampoulidis, 2024, Proposition 7) If $\sigma$ is $l$-Lipschitz continuous, $\mu$-strongly convex over the interval $[-2, 2]$, each entry of $\boldsymbol{w}_1$ is i.i.d. from $\mathcal{N}(0, 1)$ and $m \geq \frac{80^2 d^3 l^2 \log(2/\delta)}{2\mu^2}$, then with probability at least $1 - \delta$, Assumption 5 is satisfied with $\gamma = \frac{\mu}{80d}$.*

Based on Lemma 1, we can prove Proposition 1.

*Proof of Proposition 1.* We borrow some proof ideas from (Taheri & Thrampoulidis, 2024). First, we consider $f$ to be logistic loss or exponential loss. Setting $\boldsymbol{w}^{(\varepsilon)} = \boldsymbol{w}_1 + \frac{\boldsymbol{w}^*}{\gamma}(2C + \log(1/\varepsilon))$ where $\boldsymbol{w}^*$ is satisfies Assumption 5. Also, set $r = \frac{4(2C + \log(1/\varepsilon))}{\gamma} + 12\sqrt{2}$. Let $\boldsymbol{W}^{(\varepsilon,i)}$ denotes the $i$-th layer weight of $\boldsymbol{w}^{(\varepsilon)}$. We first derive from Lemma 2 that $\|\boldsymbol{W}_1^{(i)}\| \leq 3\sqrt{2} \leq \frac{r}{4}$. In addition, we have

$$\|\boldsymbol{w}^{(\varepsilon)} - \boldsymbol{w}_1\| = \frac{2C + \log(1/\varepsilon)}{\gamma}, \quad \|\boldsymbol{W}^{(\varepsilon,i)}\| \leq \|\boldsymbol{W}_1^{(i)}\| + \|\boldsymbol{W}^{(\varepsilon,i)} - \boldsymbol{W}_1^{(i)}\| \leq \frac{r}{2}.$$

Then, for any $\boldsymbol{w} \in [\boldsymbol{w}^{(\varepsilon)}, \boldsymbol{w}_1]$, we get that $\|\boldsymbol{W}^{(i)}\| = \|\lambda \boldsymbol{W}^{(\varepsilon,i)} + (1 - \lambda)(\boldsymbol{W}_1^{(i)})\| \leq \frac{r}{2}, \forall i \in [L], \lambda \in [0, 1]$. From Lemmas 1 and 2, if $\sqrt{m} \geq \max\{1, 2lr\}$ and $\|\boldsymbol{a}\| \leq 3\sqrt{2} < r$, then,

$$\|\nabla_{\boldsymbol{w}} \Phi(\boldsymbol{w}, \boldsymbol{x}_t)\| < \frac{R_2}{m}, \quad \|\nabla_{\boldsymbol{w}}^2 \Phi(\boldsymbol{w}, \boldsymbol{x}_t)\| < \frac{R_3}{m}, \quad \forall t \geq 1,$$

where $R_2, R_3$ are as in Lemma 1 with $r$ defined above. Combining Taylor's expansion theorem, there exists $\boldsymbol{w}' \in [\boldsymbol{w}^{(\varepsilon)}, \boldsymbol{w}_1]$ such that for any $t \in [T]$,

$$
\begin{aligned}
y_t \Phi(\boldsymbol{w}^{(\varepsilon)}, \boldsymbol{x}_t) &= y_t \Phi(\boldsymbol{w}_1, \boldsymbol{x}_t) + y_t \Big\langle \nabla_{\boldsymbol{w}_1} \Phi(\boldsymbol{w}_1, \boldsymbol{x}_t), \boldsymbol{w}^{(\varepsilon)} - \boldsymbol{w}_1 \Big\rangle \\
&\quad + \frac{y_t}{2} \Big\langle \boldsymbol{w}^{(\varepsilon)} - \boldsymbol{w}_1, \nabla_{\boldsymbol{w}'}^2 \Phi(\boldsymbol{w}', \boldsymbol{x}_t)(\boldsymbol{w}^{(\varepsilon)} - \boldsymbol{w}_1) \Big\rangle \\
&\geq - |y_t \Phi(\boldsymbol{w}_1, \boldsymbol{x}_t)| + y_t \Big\langle \nabla_{\boldsymbol{w}_1} \Phi(\boldsymbol{w}_1, \boldsymbol{x}_t), \boldsymbol{w}^{(\varepsilon)} - \boldsymbol{w}_1 \Big\rangle \\
&\quad - \frac{|y_t|}{2} \left\| \nabla_{\boldsymbol{w}'}^2 \Phi(\boldsymbol{w}', \boldsymbol{x}_t) \right\| \left\| \boldsymbol{w}^{(\varepsilon)} - \boldsymbol{w}_1 \right\|^2 \\
&\geq -C + 2C + \log(1/\varepsilon) - \frac{R_3}{2\gamma^2 m}(2C + \log(1/\varepsilon))^2 \geq \log(1/\varepsilon),
\end{aligned}
$$

where we use $m \geq \frac{R_3}{2\gamma^2 C}(2C + \log(1/\varepsilon))^2$ in the last inequality. When $f$ is logistic loss function, we derive that $F_t(\boldsymbol{w}^{(\varepsilon)}) = f(y_t \Phi(\boldsymbol{w}^{(\varepsilon)}, \boldsymbol{x}_t)) \leq \log(1 + \varepsilon) \leq \varepsilon$. When $f$ is exponential loss function, we derive that $F_t(\boldsymbol{w}^{(\varepsilon)}) = f(y_t \Phi(\boldsymbol{w}^{(\varepsilon)}, \boldsymbol{x}_t)) = \mathrm{e}^{-y_t \Phi(\boldsymbol{w}^{(\varepsilon)}, \boldsymbol{x}_t)} \leq \mathrm{e}^{-\log(1/\varepsilon)} \leq \varepsilon$. Then, we can derive that $\sum_{t=1}^{T} F_t(\boldsymbol{w})/T \leq \varepsilon, \forall T \geq 1$. Finally, with $R_3 \sim \mathcal{O}(r^2 \sqrt{L}) \sim \mathcal{O}(\sqrt{L} \log^2(1/\varepsilon))$, we get the desired result.

When $f(u) = 1/u^\beta$, we can follow the above proof with $\log(1/\varepsilon)$ replaced by $(\frac{1}{\varepsilon})^{1/\beta}$. Setting $\boldsymbol{w}^{(\varepsilon)} = \boldsymbol{w}_1 + \frac{\boldsymbol{w}^*}{\gamma}(2C + (\frac{1}{\varepsilon})^{1/\beta})$ where $\boldsymbol{w}^*$ is satisfies Assumption 5, we get that

$$
\|\boldsymbol{w}^{(\varepsilon)} - \boldsymbol{w}_1\| = \frac{2C + (1/\varepsilon)^{1/\beta}}{\gamma}.
$$

Let

$$
r = \frac{4(2C + (1/\varepsilon)^{1/\beta})}{\gamma} + 12\sqrt{2}, \quad m \geq \frac{R_3(2C + (1/\varepsilon)^{1/\beta})^2}{2\gamma^2 C}.
$$

Then, we also have $\|\boldsymbol{W}^{(\varepsilon, i)}\|, \|\boldsymbol{W}_1^{(i)}\| \leq \frac{r}{2}, \forall i \in [L]$. Based on Lemma 1, we can use a similar deduction as the proof of Proposition 1 to get that $y_t \Phi(\boldsymbol{w}^{(\varepsilon)}, \boldsymbol{x}_t) \geq (1/\varepsilon)^{1/\beta}$, which leads to that $f(y_t \Phi(\boldsymbol{w}^{(\varepsilon)}, \boldsymbol{x}_t)) \leq \varepsilon$. With $R_3 \sim \mathcal{O}(r^2 \sqrt{L})$, we get that $m \geq \mathcal{O}(\sqrt{L}(1/\beta)^{4/\beta})$.

$\square$

*Proof of Remarks 1 and 3.* Here, we prove the two remarks together. When the NTK-separability holds, and $|\Phi(\boldsymbol{w}_1, \boldsymbol{x})| \leq C$, we get from Proposition 1 and Lemma 2 that

$$
g\left(\frac{\epsilon}{T}\right) = \mathcal{O}\left(\log\left(\frac{T}{\epsilon}\right)\right), \quad \text{when} \quad m \geq \mathcal{O}\left(\frac{\log^4(T/\epsilon) L^{1.5}}{\gamma^2}\right).
$$

Then, combining Theorem 1 since logistic loss and exponential loss are both self-bounded, we can prove the desired result in Remark 1,

$$
R(T) \leq \mathcal{O}(L^2(\log^3 T)), \quad \text{when,} \quad m \geq \mathcal{O}(L^{3.5} \log^4 T).
$$

Also, combining Theorem 8, we can derive the desired result in Remark 3,

$$
R(T) \leq \mathcal{O}(\log^2 T), \quad \text{when,} \quad m \geq \mathcal{O}(L^{2.5} \log^4 T).
$$

$\square$

### D.2 DETAILED PROOF FOR RESULTS IN SECTION 4

We highlight the notations $F_t(\boldsymbol{w}) = f(y_t \Phi(\boldsymbol{w}, \boldsymbol{x}_t))$, $\nabla F_t(\boldsymbol{w}) = \nabla_{\boldsymbol{w}} f(y_t \Phi(\boldsymbol{w}, \boldsymbol{x}_t))$ and $\nabla^2 F_t(\boldsymbol{w}) = \nabla_{\boldsymbol{w}}^2 f(y_t \Phi(\boldsymbol{w}, \boldsymbol{x}_t))$.

*Proof of Lemma 2.* For any $i \in [L] \setminus [1]$, let $\tilde{\boldsymbol{W}}_1^{(i)} = (\tilde{W}_{ij})_{ij} = \frac{\sqrt{m}\boldsymbol{W}_1^{(i)}}{\sqrt{2}}$. Then, each entry $\tilde{W}_{ij} \sim \mathcal{N}(0, 1)$. Using (Vershynin, 2010, Corollary 5.35), it leads to that with probability at least $1 - 2\exp(-m/2)$,

$$\sqrt{\frac{m}{2}} \left\| \boldsymbol{W}_1^{(i)} \right\| = \left\| \tilde{\boldsymbol{W}}_1^{(i)} \right\| \leq 3\sqrt{m},$$

which leads to that $\left\| \boldsymbol{W}_1^{(i)} \right\| \leq 3\sqrt{2}$. Using (Vershynin, 2010, Corollary 5.35) and $m \geq d \geq 1$, it leads to that with probability at least $1 - 2\exp(-m/2)$,

$$\left\| \boldsymbol{W}_1^{(1)} \right\| \leq \sqrt{\frac{2}{m}} \left( 2\sqrt{m} + \sqrt{d} \right) \leq 3\sqrt{2}.$$

Similarly, we set $\tilde{\boldsymbol{a}} = \sqrt{\frac{m}{2}}\boldsymbol{a}$ which also satisfies standard Gaussian. Note that $\boldsymbol{a} \in \mathbb{R}^m$ is a vector. Then, we get that with proability at least $1 - 2\exp(-m/2)$, $\|\tilde{\boldsymbol{a}}\| \leq 2\sqrt{m}$, which leads to that $\|\boldsymbol{a}\| \leq 4 < 3\sqrt{2}$. $\qquad\square$

*Proof of Lemma 3.* We use $F_t'(\boldsymbol{w}) = f_t'(y_t\Phi(\boldsymbol{w}, \boldsymbol{x}_t))$ and $F_t''(\boldsymbol{w}) = f_t''(y_t\Phi(\boldsymbol{w}, \boldsymbol{x}_t))$ for the notation simplicity. Recalling the objective function in (2), we derive that for any $t \geq 1$,

$$\nabla F_t(\boldsymbol{w}) = F_t'(\boldsymbol{w})y_t\nabla_{\boldsymbol{w}}\Phi(\boldsymbol{w}, \boldsymbol{x}_t). \tag{26}$$

Taking norm on both sides of (26), then using $y_t \in \{\pm 1\}$, Assumption 2 and Lemma 1, we get that for any $t \geq 1$,

$$\|\nabla F_t(\boldsymbol{w})\| \leq |F_t'(\boldsymbol{w})| \|\nabla_{\boldsymbol{w}}\Phi(\boldsymbol{w}, \boldsymbol{x}_t)\| \leq \frac{\alpha_f R_2}{m} \cdot F_t(\boldsymbol{w}).$$

For the Hessian matrix, note that

$$\nabla^2 F_t(\boldsymbol{w}) = \underbrace{F_t''(\boldsymbol{w})\nabla_{\boldsymbol{w}}\Phi(\boldsymbol{w}, \boldsymbol{x}_t)\nabla_{\boldsymbol{w}}\Phi(\boldsymbol{w}, \boldsymbol{x}_t)^\top}_{\mathbf{A}} + \underbrace{F_t'(\boldsymbol{w})y_t\nabla_{\boldsymbol{w}}^2\Phi(\boldsymbol{w}, \boldsymbol{x}_t)}_{\mathbf{B}}, \quad \forall t \geq 1. \tag{27}$$

Note that when $f$ is convex, $\mathbf{A}$ in (27) is positive semi-definite since $F_t''(\boldsymbol{w}) \geq 0$. Then, using Weyl's inequality and $\mathbf{B}$ is Hermitian, we derive that

$$\lambda_{\min}(\nabla^2 F_t(\boldsymbol{w})) \geq \lambda_{\min}(\mathbf{B}) \geq -\sqrt{\|y_t F_t'(\boldsymbol{w})\nabla_{\boldsymbol{w}}^2\Phi(\boldsymbol{w}, \boldsymbol{x}_t)\|^2}$$

$$\geq -\sqrt{|y_t F_t'(\boldsymbol{w})|^2 \|\nabla_{\boldsymbol{w}}^2\Phi(\boldsymbol{w}, \boldsymbol{x}_t)\|^2} \geq -\frac{\alpha_f R_3}{m} \cdot F_t(\boldsymbol{w}),$$

where the last inequality applies Lemma 1 and Assumption 2. $\qquad\square$

The following lemma is a standard result for smooth losses.

**Lemma 10.** *Let $f$ be any non-negative function satisfying Assumption 3. Then,*

$$|f'(u)| \leq \sqrt{2L_f f(u)}, \quad \forall u \in \text{dom}(f).$$

*Proof of Lemma 4.* Following the notations in the proof of Lemma 3, we can take squared norm on both sides of (26), and using $y_t \in \{\pm 1\}$, Assumption 3, Lemma 1 and Lemma 10 to get that

$$\|\nabla F_t(\boldsymbol{w})\| \leq \sqrt{(F_t'(\boldsymbol{w}))^2 \|\nabla_{\boldsymbol{w}}\Phi(\boldsymbol{w}, \boldsymbol{x}_t)\|^2} \leq \frac{R_2}{m} \cdot \sqrt{2L_f F_t(\boldsymbol{w})}. \tag{28}$$

Note that when $f$ is convex, $\mathbf{A}$ in (27) is positive semi-definite since $F_t''(\boldsymbol{w}) \geq 0$. Then, using Weyl's inequality and $\mathbf{B}$ is Hermitian, we derive that

$$\lambda_{\min}(\nabla^2 F_t(\boldsymbol{w})) \geq \lambda_{\min}(\mathbf{B}) \geq -\sqrt{\|y_t F_t'(\boldsymbol{w})\nabla_{\boldsymbol{w}}^2\Phi(\boldsymbol{w}, \boldsymbol{x}_t)\|^2}$$

$$\geq -\sqrt{|y_t F_t'(\boldsymbol{w})|^2 \|\nabla_{\boldsymbol{w}}^2\Phi(\boldsymbol{w}, \boldsymbol{x}_t)\|^2} \geq -\frac{R_3\sqrt{2L_f F_t(\boldsymbol{w})}}{m},$$

where the last inequality applies Lemma 1 and Lemma 10. $\qquad\square$

Before proving Lemma 5, we will provide the following lemma to estimate the maximum value of $F_t(\boldsymbol{w})$ along a segment. For any two vectors $\boldsymbol{w}_1, \boldsymbol{w}_2 \in \mathbb{R}^{d'}$, we denote $[\boldsymbol{w}_1, \boldsymbol{w}_2] = \{\boldsymbol{w} : \boldsymbol{w} = \alpha\boldsymbol{w}_1 + (1-\alpha)\boldsymbol{w}_2, \alpha \in [0,1]\}$ and $(\boldsymbol{w}_1, \boldsymbol{w}_2) = \{\boldsymbol{w} : \boldsymbol{w} = \alpha\boldsymbol{w}_1 + (1-\alpha)\boldsymbol{w}_2, \alpha \in (0,1)\}$.

**Lemma 11.** *Generally, let the same assumptions of Lemma 5 hold. If Assumption 2 holds, and* $m \geq \max\left\{1, 4l^2r^2, \frac{\alpha_f D^2 R_3}{2}\right\}$, *then*

$$\max_{\boldsymbol{w}\in[\boldsymbol{w}_1,\boldsymbol{w}_2]} F_t(\boldsymbol{w}) \leq \left(1 - \frac{\alpha_f D^2 R_3}{2m}\right)^{-1} \max\left\{F_t(\boldsymbol{w}_1), F_t(\boldsymbol{w}_2)\right\}.$$

*If Assumption 3 holds, and* $m \geq \max\{1, 4l^2r^2\}$, *then*

$$\max_{\boldsymbol{w}\in[\boldsymbol{w}_1,\boldsymbol{w}_2]} F_t(\boldsymbol{w}) \leq 2\max\left\{F_t(\boldsymbol{w}_1), F_t(\boldsymbol{w}_2)\right\} + \frac{D^4 L_f R_3^2}{m^2}.$$

*Proof.* Letting $\boldsymbol{w}_a = \arg\max_{\boldsymbol{w}\in[\boldsymbol{w}_1,\boldsymbol{w}_2]} F_t(\boldsymbol{w})$ and using Taylor's expansion theorem, we get that there exists $\boldsymbol{w}_b \in [\boldsymbol{w}_1, \boldsymbol{w}_a]$ such that

$$F_t(\boldsymbol{w}_1) = F_t(\boldsymbol{w}_a) + \langle \nabla F_t(\boldsymbol{w}_a), \boldsymbol{w}_1 - \boldsymbol{w}_a \rangle + \frac{1}{2}\langle \boldsymbol{w}_1 - \boldsymbol{w}_a, \nabla^2 F_t(\boldsymbol{w}_b)(\boldsymbol{w}_1 - \boldsymbol{w}_a)\rangle.$$

If $\boldsymbol{w}_a = \boldsymbol{w}_1$ or $\boldsymbol{w}_a = \boldsymbol{w}_2$, the result is then trivial. Suppose that $\boldsymbol{w}_a \in (\boldsymbol{w}_1, \boldsymbol{w}_2)$. Since $F_t(\boldsymbol{w})$ is a differentiable function, we could derive from the optimal condition that $\langle \nabla F_t(\boldsymbol{w}_a), \boldsymbol{w}_1 - \boldsymbol{w}_a\rangle = 0$. Then, we have

$$F_t(\boldsymbol{w}_1) \geq F_t(\boldsymbol{w}_a) + \frac{\|\boldsymbol{w}_1 - \boldsymbol{w}_a\|^2}{2}\lambda_{\min}(\nabla^2 F_t(\boldsymbol{w}_b)). \tag{29}$$

Then, using Lemma 3 into (29), and denoting $M_1 = \frac{\alpha_f R_3}{m}$,

$$F_t(\boldsymbol{w}_1) \geq F_t(\boldsymbol{w}_a) - \frac{M_1\|\boldsymbol{w}_1 - \boldsymbol{w}_a\|^2}{2} \cdot F_t(\boldsymbol{w}_b)$$

$$\geq F_t(\boldsymbol{w}_a) - \frac{M_1 D^2}{2} \cdot F_t(\boldsymbol{w}_a) = \left(1 - \frac{M_1 D^2}{2}\right)F_t(\boldsymbol{w}_a).$$

Using the same deduction, we can also get that

$$F_t(\boldsymbol{w}_2) \geq \left(1 - \frac{M_1 D^2}{2}\right)F_t(\boldsymbol{w}_a).$$

Combining the above, we get the desired result. When Assumption 3 holds, using Lemma 4 into (29), and denoting $M_2 = \frac{R_3\sqrt{2L_f}}{m}$, we get that

$$F_t(\boldsymbol{w}_1) \geq F_t(\boldsymbol{w}_a) - \frac{M_2\|\boldsymbol{w}_1 - \boldsymbol{w}_a\|^2}{2} \cdot \sqrt{F_t(\boldsymbol{w}_b)}$$

$$\geq F_t(\boldsymbol{w}_a) - \frac{M_2 D^2}{2} \cdot \sqrt{F_t(\boldsymbol{w}_a)} = \left(\sqrt{F_t(\boldsymbol{w}_a)} - \frac{M_2 D^2}{4}\right)^2 - \frac{M_2^2 D^4}{16}.$$

Hence, we derive that

$$\sqrt{F_t(\boldsymbol{w}_a)} \leq \frac{M_2 D^2}{4} + \sqrt{F_t(\boldsymbol{w}_1) + \frac{M_2^2 D^4}{16}} \leq \sqrt{F_t(\boldsymbol{w}_1)} + \frac{M_2 D^2}{2}.$$

Using the same deduction, we also derive that $\sqrt{F_t(\boldsymbol{w}_a)} \leq \sqrt{F_t(\boldsymbol{w}_2)} + \frac{M_2 D^2}{2}$. Thus, combining the two inequalities and using Young's inequality, we prove that

$$\max_{\boldsymbol{w}\in[\boldsymbol{w}_1,\boldsymbol{w}_2]} F_t(\boldsymbol{w}) = F_t(\boldsymbol{w}_a) \leq \left(\max\left\{\sqrt{F_t(\boldsymbol{w}_1)}, \sqrt{F_t(\boldsymbol{w}_2)}\right\} + \frac{M_2 D^2}{2}\right)^2$$

$$\leq 2\max\left\{F_t(\boldsymbol{w}_1), F_t(\boldsymbol{w}_2)\right\} + \frac{M_2^2 D^4}{2},$$

which can lead to the desired result. $\qquad\square$

*Proof of Lemma 5.* When Assumption 2 holds, based on Taylor's expansion theorem and Lemma 3, there exists $\boldsymbol{w}_a \in [\boldsymbol{w}_t, \boldsymbol{w}]$ such that

$$
F_t(\boldsymbol{w}) = F_t(\boldsymbol{w}_t) + \langle \nabla F_t(\boldsymbol{w}_t), \boldsymbol{w} - \boldsymbol{w}_t \rangle + \frac{1}{2} \left\langle \boldsymbol{w} - \boldsymbol{w}_t, \nabla^2 F_t(\boldsymbol{w}_a)(\boldsymbol{w} - \boldsymbol{w}_t) \right\rangle
$$

$$
\geq F_t(\boldsymbol{w}_t) + \langle \nabla F_t(\boldsymbol{w}_t), \boldsymbol{w} - \boldsymbol{w}_t \rangle + \frac{\|\boldsymbol{w} - \boldsymbol{w}_t\|^2}{2} \lambda_{\min} \left( \nabla^2 F_t(\boldsymbol{w}_a) \right)
$$

$$
\geq F_t(\boldsymbol{w}_t) + \langle \nabla F_t(\boldsymbol{w}_t), \boldsymbol{w} - \boldsymbol{w}_t \rangle - \frac{\|\boldsymbol{w} - \boldsymbol{w}_t\|^2}{2} \cdot \frac{\alpha_f R_3}{m} \cdot F_t(\boldsymbol{w}_a)
$$

$$
\geq F_t(\boldsymbol{w}_t) + \langle \nabla F_t(\boldsymbol{w}_t), \boldsymbol{w} - \boldsymbol{w}_t \rangle - \frac{\|\boldsymbol{w} - \boldsymbol{w}_t\|^2}{2} \cdot \frac{\alpha_f R_3}{m} \cdot \max_{\boldsymbol{w} \in [\boldsymbol{w}, \boldsymbol{w}_t]} F_t(\boldsymbol{w}).
$$

Then, using $\|\boldsymbol{w}_t - \boldsymbol{w}\| \leq D$ and Lemma 11 under self-bounded losses, we obtain that

$$
F_t(\boldsymbol{w}) \geq F_t(\boldsymbol{w}_t) + \langle \nabla F_t(\boldsymbol{w}_t), \boldsymbol{w} - \boldsymbol{w}_t \rangle
$$

$$
- \frac{D^2 \alpha_f R_3}{2m} \cdot \left( 1 - \frac{\alpha_f D^2 R_3}{2m} \right)^{-1} F_t(\boldsymbol{w}_t) - \frac{D^2 \alpha_f R_3}{2m} \cdot \left( 1 - \frac{\alpha_f D^2 R_3}{2m} \right)^{-1} F_t(\boldsymbol{w}).
$$

We get that when $m \geq \frac{9\alpha_f D^2 R_3}{2}$,

$$
\frac{D^2 \alpha_f R_3}{2m} \cdot \left( 1 - \frac{\alpha_f D^2 R_3}{2m} \right)^{-1} x \leq \frac{x}{8}.
$$

Then, combining the above, we get the desired result. When Assumption 3 holds, based on Taylor's expansion theorem and Lemma 4, there exists $\boldsymbol{w}_a \in [\boldsymbol{w}_t, \boldsymbol{w}]$ such that

$$
F_t(\boldsymbol{w}) = F_t(\boldsymbol{w}_t) + \langle \nabla F_t(\boldsymbol{w}_t), \boldsymbol{w} - \boldsymbol{w}_t \rangle + \frac{1}{2} \left\langle \boldsymbol{w} - \boldsymbol{w}_t, \nabla^2 F_t(\boldsymbol{w}_a)(\boldsymbol{w} - \boldsymbol{w}_t) \right\rangle
$$

$$
\geq F_t(\boldsymbol{w}_t) + \langle \nabla F_t(\boldsymbol{w}_t), \boldsymbol{w} - \boldsymbol{w}_t \rangle + \frac{\|\boldsymbol{w} - \boldsymbol{w}_t\|^2}{2} \lambda_{\min} \left( \nabla^2 F_t(\boldsymbol{w}_a) \right)
$$

$$
\geq F_t(\boldsymbol{w}_t) + \langle \nabla F_t(\boldsymbol{w}_t), \boldsymbol{w} - \boldsymbol{w}_t \rangle - \frac{\|\boldsymbol{w} - \boldsymbol{w}_t\|^2}{2} \cdot \frac{R_3 \sqrt{2L_f}}{m} \cdot \sqrt{F_t(\boldsymbol{w}_a)}
$$

$$
\geq F_t(\boldsymbol{w}_t) + \langle \nabla F_t(\boldsymbol{w}_t), \boldsymbol{w} - \boldsymbol{w}_t \rangle - \frac{\|\boldsymbol{w} - \boldsymbol{w}_t\|^2}{2} \cdot \frac{R_3 \sqrt{2L_f}}{m} \cdot \max_{\boldsymbol{w} \in [\boldsymbol{w}_t, \boldsymbol{w}]} \sqrt{F_t(\boldsymbol{w})}.
$$

Then, using Lemma 11 under smooth losses, we obtain that

$$
F_t(\boldsymbol{w}) \geq F_t(\boldsymbol{w}_t) + \langle \nabla F_t(\boldsymbol{w}_t), \boldsymbol{w} - \boldsymbol{w}_t \rangle
$$

$$
- \frac{D^2 R_3 \sqrt{2L_f}}{2m} \sqrt{2 F_t(\boldsymbol{w}_t) + \frac{D^4 L_f R_3^2}{m^2}} - \frac{D^2 R_3 \sqrt{2L_f}}{2m} \sqrt{2 F_t(\boldsymbol{w}) + \frac{D^4 L_f R_3^2}{m^2}}.
$$

Using that $m^2 \geq D^4 L_f R_3^2 T$, we get that

$$
\frac{D^2 R_3 \sqrt{2L_f}}{2m} \sqrt{2x + \frac{D^4 L_f R_3^2}{m^2}} \leq \frac{x}{8} + \frac{(2 + 1/\sqrt{2}) D^4 L_f R_3^2}{m} \leq \frac{x}{8} + \frac{2 + 1/\sqrt{2}}{T}.
$$

Combining the above, we get the desired result. $\qquad \square$

# E    DETAILED PROOF OF RESULTS IN SECTION 5

We first provide a result for the Bregman divergence.

**Lemma 12.** *(Chen & Teboulle, 1993) Let $S \subset \mathbb{R}^{d'}$ be an open set with closure $\bar{S}$. Also, let $\psi : \bar{S} \to \mathbb{R}$ be continuous differentiable on $S$. Then, for any $\boldsymbol{a}, \boldsymbol{b} \in S$ and $\boldsymbol{c} \in \bar{S}$,*

$$
B_\psi(\boldsymbol{c}, \boldsymbol{a}) + B_\psi(\boldsymbol{a}, \boldsymbol{b}) - B_\psi(\boldsymbol{c}, \boldsymbol{b}) = \langle \nabla \psi(\boldsymbol{b}) - \nabla \psi(\boldsymbol{a}), \boldsymbol{c} - \boldsymbol{a} \rangle.
$$

Then, we prove a proposition showing that (9) can reduce to SGD, RMSProp, AdamNC and their projected form with different selections of $\psi_t$.

**Proposition 3.** *Let $\{w_t\}_{t \geq 1}$ be generated by (9) with $\mathcal{X} \subseteq \mathbb{R}^{d'}$, $\phi(w) = 0$ and $\psi_t(z) = \frac{1}{2}\|z\|^2_{\Lambda_t}$, where $\Lambda_t \in \mathbb{R}^{d' \times d'}$ is a symmetric positive-definite matrix. Then, $\psi_t(z)$ satisfies (10), and (9) reduces to*

$$w_{t+1} = \Pi_{\mathcal{X}}^{\Lambda_t} \left( w_t - C_0 \Lambda_t^{-1} m_t \right). \tag{30}$$

*When $\mathcal{X} = \mathbb{R}^{d'}$, and $\epsilon$ is a positive constant, the following results hold:*
*(a). If $\Lambda_t = I_{d' \times d'}$, then (9) becomes SGD with momentum;*
*(b). If $\beta_1 = 0, \Lambda_t = \eta_t^{-1}\text{Diag}\left(\epsilon + \sqrt{v_{t,i}}\right)_{i \in [d']}$ where $v_{t,i} = v_{t-1,i} + g_{t,i}^2$, then (9) becomes AdaGrad;*
*(c). If $\Lambda_t$ is as in Algorithm 1, then (9) becomes AdamNC. Further setting $\beta_1 = 0$, (9) reduces to RMSProp.*
*(d). When $\mathcal{X} \subset \mathbb{R}^{d'}$, the above cases reduce to their respective projected forms.*

*Proof of Proposition 3.* Note that when $\psi_t(z) = \frac{1}{2}\|z\|^2_{\Lambda_t}$, $B_{\psi_t}(z_1, z_2) = \frac{1}{2}\|z_1 - z_2\|^2_{\Lambda_t}$. Then, we get that

$$\psi_t(z_1) - \psi_t(z_2) - \langle z_1 - z_2, \nabla\psi_t(z_2)\rangle = \frac{1}{2}\|z_1\|^2_{\Lambda_t} + \frac{1}{2}\|z_2\|^2_{\Lambda_t} - \langle z_1, \Lambda_t z_2\rangle$$

$$= \frac{1}{2}\|z_1 - z_2\|^2_{\Lambda_t},$$

which shows that $\psi_t(z) = \frac{1}{2}\|z\|^2_{\Lambda_t}$ satisfies (10). When $\mathcal{X}$ is a closed convex set, we note that (30) is equal to that

$$w_{t+1} = \arg\min_{w \in \mathcal{X}} \left\|w - w_t + C_0\Lambda_t^{-1}m_t\right\|^2_{\Lambda_t}$$

$$= \arg\min_{w \in \mathcal{X}} \left[\|w - w_t\|^2_{\Lambda_t} + 2\langle w - w_t, C_0 m_t\rangle + \left\|C_0\Lambda_t^{-1}m_t\right\|^2_{\Lambda_t}\right]$$

$$= \arg\min_{w \in \mathcal{X}} \left[\frac{1}{2C_0}\|w - w_t\|^2_{\Lambda_t} + \langle w - w_t, m_t\rangle\right],$$

which is exactly the optimization problem in (9) with $B_{\psi_t}(w, w_t) = \frac{1}{2}\|w - w_t\|^2_{\Lambda_t}$. Then, with different setups of $\Lambda_t$, (9) reduces to the corresponding algorithms based on their definitions. $\qquad\square$

*Proof of Proposition 2.* Using the definition of $m_s$ from (9), we get that for any $s \geq 1$,

$$-\langle w_s - w, m_s\rangle = -(1 - \beta_1)\langle w_s - w, g_s\rangle - \beta_1\langle w_s - w, m_{s-1}\rangle$$

$$= -(1 - \beta_1)\langle w_s - w, g_s\rangle - \beta_1\langle w_s - w_{s-1}, m_{s-1}\rangle - \beta_1\langle w_{s-1} - w, m_{s-1}\rangle.$$

We then get that

$$\sum_{s=1}^{t}\langle w_s - w, g_s\rangle = \underbrace{\frac{\beta_1}{1 - \beta_1}\sum_{s=1}^{t}\left(\langle w_s - w, m_s\rangle - \langle w_{s-1} - w, m_{s-1}\rangle\right)}_{I_1}$$

$$+ \underbrace{\sum_{s=1}^{t}\langle w_s - w, m_s\rangle}_{I_2} - \underbrace{\frac{\beta_1}{1 - \beta_1}\sum_{s=1}^{t}\langle w_s - w_{s-1}, m_{s-1}\rangle}_{I_3}. \tag{31}$$

**Bounding $I_1$.** Using the telescoping sum and $m_0 = 0_{d'}$, we get that

$$I_1 = \frac{\beta_1}{1 - \beta_1}\left(\langle w_t - w, m_t\rangle - \langle w_0 - w, m_0\rangle\right) = \frac{\beta_1}{1 - \beta_1}\langle w_t - w, m_t\rangle. \tag{32}$$

**Bounding $I_2$.** From the optimality implied by (9), we get that for any $w \in \mathcal{X}$ and $\dot{\phi} \in \partial\phi(w_{s+1})$, where $\partial\phi(w_{s+1})$ is the sub-differential of of $\phi(w_{s+1})$,

$$\left\langle w - w_{s+1}, m_s + \dot{\phi} + \frac{1}{C_0}\left(\nabla\psi_s(w_{s+1}) - \nabla\psi_s(w_s)\right)\right\rangle \geq 0.$$

We further derive that

$$\langle \boldsymbol{w}_s - \boldsymbol{w}, \boldsymbol{m}_s \rangle = \langle \boldsymbol{w}_{s+1} - \boldsymbol{w}, \boldsymbol{m}_s \rangle + \langle \boldsymbol{w}_s - \boldsymbol{w}_{s+1}, \boldsymbol{m}_s \rangle$$

$$\leq \frac{1}{C_0} \langle \boldsymbol{w} - \boldsymbol{w}_{s+1}, \nabla \psi_s(\boldsymbol{w}_{s+1}) - \nabla \psi_s(\boldsymbol{w}_s) \rangle + \langle \boldsymbol{w}_s - \boldsymbol{w}_{s+1}, \boldsymbol{m}_s \rangle + \langle \boldsymbol{w} - \boldsymbol{w}_{s+1}, \dot{\boldsymbol{\phi}} \rangle$$

$$\leq \frac{1}{C_0} \left( B_{\psi_s}(\boldsymbol{w}, \boldsymbol{w}_s) - B_{\psi_s}(\boldsymbol{w}, \boldsymbol{w}_{s+1}) - B_{\psi_s}(\boldsymbol{w}_{s+1}, \boldsymbol{w}_s) \right)$$

$$+ \langle \boldsymbol{w}_s - \boldsymbol{w}_{s+1}, \boldsymbol{m}_s \rangle + \phi(\boldsymbol{w}) - \phi(\boldsymbol{w}_{s+1}),$$

where the last inequality applies Lemma 12 and the convexity of $\phi(\boldsymbol{w})$. Using Fenchel's inequality and Young's inequality, we get that

$$\langle \boldsymbol{w}_s - \boldsymbol{w}_{s+1}, \boldsymbol{m}_s \rangle \leq \frac{\|\boldsymbol{w}_s - \boldsymbol{w}_{s+1}\|_{\psi_s}^2}{4C_0} + C_0 \|\boldsymbol{m}_s\|_{\psi_s^*}^2 \leq \frac{B_{\psi_s}(\boldsymbol{w}_{s+1}, \boldsymbol{w}_s)}{2C_0} + C_0 \|\boldsymbol{m}_s\|_{\psi_s^*}^2,$$

where the last inequality applies that $\psi_s$ is 1-strongly convex with respect to $\| \cdot \|_{\psi_s}$. Combining the above, we get that

$$\langle \boldsymbol{w}_s - \boldsymbol{w}, \boldsymbol{m}_s \rangle \leq \frac{1}{C_0} \left( B_{\psi_s}(\boldsymbol{w}, \boldsymbol{w}_s) - B_{\psi_s}(\boldsymbol{w}, \boldsymbol{w}_{s+1}) - \frac{B_{\psi_s}(\boldsymbol{w}_{s+1}, \boldsymbol{w}_s)}{2} \right)$$

$$+ C_0 \|\boldsymbol{m}_s\|_{\psi_s^*}^2 + \phi(\boldsymbol{w}) - \phi(\boldsymbol{w}_{s+1}). \tag{33}$$

Summing up both sides of (33) over $s \in [t]$, we derive that

$$I_2 \leq \frac{1}{C_0} \sum_{s=1}^{t} \left( B_{\psi_s}(\boldsymbol{w}, \boldsymbol{w}_s) - B_{\psi_s}(\boldsymbol{w}, \boldsymbol{w}_{s+1}) \right) - \frac{1}{2C_0} \sum_{s=1}^{t} B_{\psi_s}(\boldsymbol{w}_{s+1}, \boldsymbol{w}_s)$$

$$+ C_0 \sum_{s=1}^{t} \|\boldsymbol{m}_s\|_{\psi_s^*}^2 + \sum_{s=1}^{t} \left( \phi(\boldsymbol{w}) - \phi(\boldsymbol{w}_{s+1}) \right). \tag{34}$$

**Bounding $I_3$.** Using Fenchel's inequality and Young's inequality, we get that

$$-\frac{\beta_1}{1-\beta_1} \langle \boldsymbol{w}_s - \boldsymbol{w}_{s-1}, \boldsymbol{m}_{s-1} \rangle \leq \frac{\|\boldsymbol{w}_s - \boldsymbol{w}_{s-1}\|_{\psi_{s-1}}^2}{4C_0} + \frac{C_0 \beta_1^2}{(1-\beta_1)^2} \|\boldsymbol{m}_{s-1}\|_{\psi_{s-1}^*}^2$$

$$\leq \frac{B_{\psi_{s-1}}(\boldsymbol{w}_s, \boldsymbol{w}_{s-1})}{2C_0} + \frac{C_0 \beta_1^2}{(1-\beta_1)^2} \|\boldsymbol{m}_{s-1}\|_{\psi_{s-1}^*}^2,$$

where the last inequality is also due to the 1-strongly convexity of $\psi_{s-1}(\cdot)$ with $\| \cdot \|_{\psi_{s-1}}$. Note that $\boldsymbol{m}_0 = \boldsymbol{0}_{d'}$. Then, we can sum up the above over $s = 2, 3, \cdots, t$ to get that

$$I_3 \leq \frac{1}{2C_0} \sum_{s=2}^{t} B_{\psi_{s-1}}(\boldsymbol{w}_s, \boldsymbol{w}_{s-1}) + \frac{C_0 \beta_1^2}{(1-\beta_1)^2} \sum_{s=2}^{t} \|\boldsymbol{m}_{s-1}\|_{\psi_{s-1}^*}^2$$

$$\leq \frac{1}{2C_0} \sum_{s=1}^{t} B_{\psi_s}(\boldsymbol{w}_{s+1}, \boldsymbol{w}_s) + \frac{C_0 \beta_1^2}{(1-\beta_1)^2} \sum_{s=1}^{t} \|\boldsymbol{m}_s\|_{\psi_s^*}^2, \tag{35}$$

where the second inequality uses that $B_{\psi_t}(\boldsymbol{w}_t, \boldsymbol{w}_{t+1}) \geq \frac{\|\boldsymbol{w}_{s+1} - \boldsymbol{w}_s\|_{\psi_s}^2}{2} \geq 0$. Combining (32), (34) and (35) into (31), and using $\beta_1 \in [0, 1)$, we prove the result. $\qquad \square$

*Proof of Lemma 6.* Recalling $\boldsymbol{m}_s$ in Algorithm 1, we get that for any $s \geq 1$,

$$\boldsymbol{m}_s = (1 - \beta_1) \sum_{s=1}^{t} \beta_1^{t-s} \boldsymbol{g}_s, \quad \boldsymbol{g}_s = \nabla F_s(\boldsymbol{w}_s). \tag{36}$$

When Assumption 2 holds, we use Cauchy-Schwarz inequality, $\|\boldsymbol{w} - \boldsymbol{w}_1\| \leq D_{\ell_2}$ from Assumption 7, (36), $\beta_1 \in [0, 1)$ and Lemma 3 to get that

$$\frac{\beta_1}{1-\beta_1} \langle \boldsymbol{w}_t - \boldsymbol{w}, \boldsymbol{m}_t \rangle \leq \frac{D_{\ell_2} \beta_1}{1-\beta_1} \left\| (1 - \beta_1) \sum_{s=1}^{t} \beta_1^{t-s} \boldsymbol{g}_s \right\|$$

$$\leq D_{\ell_2} \beta_1 \sum_{s=1}^{t} \|\boldsymbol{g}_s\| \leq \frac{D_{\ell_2} \beta_1 \alpha_f R_2}{m} \sum_{s=1}^{t} F_s(\boldsymbol{w}_s).$$

When Assumption 3 holds, we can use Cauchy-Schwarz inequality, $\|\boldsymbol{w} - \boldsymbol{w}_1\| \leq D_{\ell_2}$ from Assumption 7, (36) and Lemma 4 to get that

$$\frac{\beta_1}{1-\beta_1}\langle \boldsymbol{w}_t - \boldsymbol{w}, \boldsymbol{m}_t\rangle \leq \frac{D_{\ell_2}\beta_1}{1-\beta_1}\sqrt{\left\|(1-\beta_1)\sum_{s=1}^{t}\beta_1^{t-s}\boldsymbol{g}_s\right\|^2} \leq \frac{D_{\ell_2}\beta_1}{1-\beta_1}\sqrt{(1-\beta_1)\sum_{s=1}^{t}\beta_1^{t-s}\|\boldsymbol{g}_s\|^2}$$

$$\leq \frac{D_{\ell_2}\beta_1}{1-\beta_1}\sqrt{2L_f\left(\frac{R_2}{m}\right)^2(1-\beta_1)\sum_{s=1}^{t}\beta_1^{t-s}F_s(\boldsymbol{w}_s)}$$

$$\leq \frac{4(D_{\ell_2}\beta_1)^2 L_f}{1-\beta_1}\left(\frac{R_2}{m}\right)^2 + \sum_{s=1}^{t}\frac{F_s(\boldsymbol{w}_s)}{8},$$

where the last inequality uses Young's inequality and $\beta_1 \in [0,1)$. $\qquad\square$

*Proof of Lemma 7.* With $\psi_s(\boldsymbol{x}) = \frac{1}{2}\|\boldsymbol{x}\|_{\boldsymbol{\Lambda}_s}^2$, we get that $B_{\psi_s}(\boldsymbol{x}, \boldsymbol{y}) = \frac{1}{2}\|\boldsymbol{x} - \boldsymbol{y}\|_{\boldsymbol{\Lambda}_s}^2$. Then,

$$\underbrace{\frac{1}{C_0}\sum_{s=1}^{t}\left(B_{\psi_s}(\boldsymbol{w}, \boldsymbol{w}_s) - B_{\psi_s}(\boldsymbol{w}, \boldsymbol{w}_{s+1})\right)}_{(*)} = \frac{1}{2C_0}\sum_{s=1}^{t}\left(\|\boldsymbol{w} - \boldsymbol{w}_s\|_{\boldsymbol{\Lambda}_s}^2 - \|\boldsymbol{w} - \boldsymbol{w}_{s+1}\|_{\boldsymbol{\Lambda}_s}^2\right)$$

$$= \frac{\|\boldsymbol{w} - \boldsymbol{w}_1\|_{\boldsymbol{\Lambda}_1}^2}{2C_0} - \frac{\|\boldsymbol{w} - \boldsymbol{w}_{t+1}\|_{\boldsymbol{\Lambda}_t}^2}{2C_0} + \frac{1}{2C_0}\sum_{s=2}^{t}\|\boldsymbol{w} - \boldsymbol{w}_s\|_{\boldsymbol{\Lambda}_s - \boldsymbol{\Lambda}_{s-1}}^2. \tag{37}$$

Let $\boldsymbol{\Lambda}_t = \mathrm{Diag}(\Lambda_{t,i})_{i\in[d']}$. Recalling $\boldsymbol{\Lambda}_t$ and $\boldsymbol{v}_t = (v_{t,i})_{i\in[d']}$ defined in Algorithm 1, we get that for any $t \geq 1, i \in [d']$,

$$v_{t,i} = (1-\beta_2)\sum_{s=1}^{t}\beta_2^{t-s}g_{s,i}^2, \quad \Lambda_{t,i} = \frac{\epsilon + \sqrt{v_{t,i}}}{\eta}. \tag{38}$$

Using $\|\boldsymbol{w} - \boldsymbol{w}_1\| \leq D_{\ell_2}$ from Assumption 7, we have

$$\frac{\|\boldsymbol{w} - \boldsymbol{w}_1\|_{\boldsymbol{\Lambda}_1}^2}{2C_0} \leq \frac{\|\boldsymbol{w} - \boldsymbol{w}_1\|_{\boldsymbol{\Lambda}_1}^2}{2C_0}\max_{i\in[d']}\Lambda_{1,i} \leq \frac{\epsilon\|\boldsymbol{w} - \boldsymbol{w}_1\|^2}{2C_0\eta} + \frac{\|\boldsymbol{g}_1\|D_{\ell_2}^2\sqrt{1-\beta_2}}{2C_0\eta}. \tag{39}$$

Using (38), we get that

$$\max_{i\in[d']}|\Lambda_{s,i} - \Lambda_{s-1,i}| = \eta^{-1}\max_{i\in[d']}\left|\sqrt{v_{s,i}} - \sqrt{v_{s-1,i}}\right| \leq \eta^{-1}\max_{i\in[d']}\sqrt{|v_{s,i} - v_{s-1,i}|}$$

$$\leq \eta^{-1}\max_{i\in[d']}\sqrt{(1-\beta_2)\left|v_{s-1,i} - g_{s,i}^2\right|}$$

$$\leq \frac{\sqrt{1-\beta_2}}{\eta}\max_{i\in[d']}\sqrt{\left|(1-\beta_2)\sum_{j=1}^{s-1}\beta_2^{s-1-j}g_{j,i}\right|^2 + g_{s,i}^2}$$

$$\leq \frac{\sqrt{1-\beta_2}}{\eta}\max_{i\in[d']}\sqrt{(1-\beta_2)\sum_{j=1}^{s-1}\beta_2^{s-1-j}g_{j,i}^2 + g_{s,i}^2}$$

$$\leq \frac{\sqrt{1-\beta_2}}{\eta}\sqrt{\sum_{j=1}^{s-1}\beta_2^{s-1-j}\|\boldsymbol{g}_j\|^2 + \|\boldsymbol{g}_s\|^2}, \tag{40}$$

where we use $\left|\sqrt{a} - \sqrt{b}\right| \le \sqrt{|a-b|}, \forall a, b \ge 0$ in the first line and the convexity of $|\cdot|^2$ in the last second inequality. From (40) and $\|\boldsymbol{w} - \boldsymbol{w}_1\| \le D_{\ell_2}$ from Assumption 7, we know that

$$\frac{1}{2C_0} \sum_{s=2}^{t} \|\boldsymbol{w} - \boldsymbol{w}_s\|_{\boldsymbol{\Lambda}_s - \boldsymbol{\Lambda}_{s-1}}^2 \le \frac{1}{2C_0} \sum_{s=2}^{t} \|\boldsymbol{w} - \boldsymbol{w}_s\|^2 \max_{i \in [d']} |\Lambda_{s,i} - \Lambda_{s-1,i}|$$

$$\le \frac{D_{\ell_2}^2 \sqrt{1-\beta_2}}{2C_0\eta} \sum_{s=2}^{t} \sqrt{\sum_{j=1}^{s-1} \beta_2^{s-1-j} \|\boldsymbol{g}_j\|^2 + \|\boldsymbol{g}_s\|^2}. \quad (41)$$

Further, we get that

$$\|\boldsymbol{g}_1\| + \sum_{s=2}^{t} \sqrt{\sum_{j=1}^{s-1} \beta_2^{s-1-j} \|\boldsymbol{g}_j\|^2 + \|\boldsymbol{g}_s\|^2} \le \|\boldsymbol{g}_1\| + \sum_{s=2}^{t} \sqrt{\sum_{j=1}^{s-1} \beta_2^{s-1-j} \|\boldsymbol{g}_j\|^2} + \sum_{s=2}^{t} \|\boldsymbol{g}_s\|$$

$$\le \sum_{s=2}^{t} \sum_{j=1}^{s-1} \beta_2^{\frac{s-j-1}{2}} \|\boldsymbol{g}_j\| + \sum_{s=1}^{t} \|\boldsymbol{g}_s\|$$

$$= \sum_{j=1}^{t-1} \|\boldsymbol{g}_j\| \sum_{s=j+1}^{t} \beta_2^{\frac{s-j-1}{2}} + \sum_{s=1}^{t} \|\boldsymbol{g}_s\|$$

$$\le \frac{1}{1-\sqrt{\beta_2}} \sum_{s=1}^{t-1} \|\boldsymbol{g}_s\| + \sum_{s=1}^{t} \|\boldsymbol{g}_s\|$$

$$\le \frac{2}{1-\sqrt{\beta_2}} \sum_{s=1}^{t} \|\boldsymbol{g}_s\|, \quad (42)$$

where the last inequality uses $\beta_2 \in [0,1)$. Then, summing up (39) and (41), and using (42) and (37), we have

$$(*) \le \frac{\epsilon \|\boldsymbol{w} - \boldsymbol{w}_1\|^2}{2C_0\eta} + \frac{D_{\ell_2}^2 \sqrt{1-\beta_2}}{C_0\eta(1-\sqrt{\beta_2})} \sum_{s=1}^{t} \|\boldsymbol{g}_s\| - \frac{\|\boldsymbol{w} - \boldsymbol{w}_{t+1}\|_{\boldsymbol{\Lambda}_t}^2}{2C_0}$$

$$\le \frac{\epsilon \|\boldsymbol{w} - \boldsymbol{w}_1\|^2}{2C_0\eta} + \frac{D_{\ell_2}^2 \sqrt{1-\beta_2}}{C_0\eta(1-\sqrt{\beta_2})} \cdot \frac{\alpha_f R_2}{m} \sum_{s=1}^{t} F_s(\boldsymbol{w}_s) - \frac{\|\boldsymbol{w} - \boldsymbol{w}_{t+1}\|_{\boldsymbol{\Lambda}_t}^2}{2C_0},$$

where the second inequality applies Lemma 3. When Assumption 3 holds, we use $\beta_2 \in [0,1)$ and Lemma 4 to get that

$$\|\boldsymbol{g}_1\| + \sum_{s=2}^{t} \sqrt{\sum_{j=1}^{s-1} \beta_2^{s-1-j} \|\boldsymbol{g}_j\|^2 + \|\boldsymbol{g}_s\|^2} \le \|\boldsymbol{g}_1\| + \sum_{s=2}^{t} \sqrt{\frac{1}{\beta_2} \sum_{j=1}^{s} \beta_2^{s-j} \|\boldsymbol{g}_j\|^2} \le \sum_{s=1}^{t} \sqrt{\frac{1}{\beta_2} \sum_{j=1}^{s} \beta_2^{s-j} \|\boldsymbol{g}_j\|^2}$$

$$\le \sum_{s=1}^{t} \sqrt{\frac{1}{\beta_2} \sum_{j=1}^{s} \beta_2^{s-j} \cdot 2L_f \left(\frac{R_2}{m}\right)^2 F_j(\boldsymbol{w}_j)}$$

$$\le \sqrt{\sum_{s=1}^{t} \sum_{j=1}^{s} \beta_2^{s-j} F_j(\boldsymbol{w}_j)} \cdot \sqrt{\sum_{s=1}^{t} 2L_f \left(\frac{R_2}{m}\right)^2 \frac{1}{\beta_2}}$$

$$\le \sqrt{\frac{\sum_{s=1}^{t} F_s(\boldsymbol{w}_s)}{1-\beta_2}} \cdot \frac{R_2}{m} \cdot \sqrt{\frac{2L_f t}{\beta_2}}.$$

Then, we also sum up (39) and (41), and use Young's inequality,

$$(*) \le \frac{\epsilon \|\boldsymbol{w} - \boldsymbol{w}_1\|^2}{2C_0\eta} + \frac{D_{\ell_2}^2 \sqrt{1-\beta_2}}{2C_0\eta} \sqrt{\frac{\sum_{s=1}^{t} F_s(\boldsymbol{w}_s)}{1-\beta_2}} \cdot \frac{R_2}{m} \cdot \sqrt{\frac{2L_f t}{\beta_2}} - \frac{\|\boldsymbol{w} - \boldsymbol{w}_{t+1}\|_{\boldsymbol{\Lambda}_t}^2}{2C_0}$$

$$\le \frac{\epsilon \|\boldsymbol{w} - \boldsymbol{w}_1\|^2}{2C_0\eta} + \left(\frac{D_{\ell_2}^2 R_2}{C_0\eta m}\right)^2 \frac{L_f t}{\beta_2} + \sum_{s=1}^{t} \frac{F_s(\boldsymbol{w}_s)}{8} - \frac{\|\boldsymbol{w} - \boldsymbol{w}_{t+1}\|_{\boldsymbol{\Lambda}_t}^2}{2C_0}.$$

$\square$

*Proof of Lemma 8.* Let $\boldsymbol{m}_s = (m_{s,i})_{i \in [d']}$. Recalling (36), we get that $m_{s,i} = (1 - \beta_1) \sum_{j=1}^{s} \beta_1^{s-j} g_{j,i}$. Further using (38), we have

$$\|\boldsymbol{m}_s\|_{\boldsymbol{\Lambda}_s^{-1}}^2 = \sum_{i=1}^{d'} \frac{\eta m_{s,i}^2}{\epsilon + \sqrt{v_{s,i}}} \leq \sum_{i=1}^{d'} \frac{\eta \left| (1-\beta_1) \sum_{j=1}^{s} \beta_1^{s-j} g_{j,i} \right|^2}{\sqrt{(1-\beta_2) \sum_{j=1}^{s} \beta_2^{s-j} g_{j,i}^2}} \leq \frac{\eta(1-\beta_1)}{\sqrt{1-\beta_2}} \sum_{i=1}^{d'} \frac{\sum_{j=1}^{s} \beta_1^{s-j} g_{j,i}^2}{\sqrt{\sum_{j=1}^{s} \beta_2^{s-j} g_{j,i}^2}}$$

$$\leq \frac{\eta(1-\beta_1)}{\sqrt{1-\beta_2}} \sum_{i=1}^{d'} \sqrt{\sum_{j=1}^{s} \beta_2^{s-j} g_{j,i}^2} \leq \frac{\eta(1-\beta_1)}{\sqrt{1-\beta_2}} \sum_{i=1}^{d'} \sum_{j=1}^{s} \beta_2^{\frac{s-j}{2}} |g_{j,i}|, \tag{43}$$

where the second inequality uses the convexity of $|\cdot|^2$ and $\beta_1 \in [0, 1)$, and the third one applies $\beta_1 \leq \beta_2$. Then, summing up both sides over $s \in [t]$, and using Lemma 3,

$$\sum_{s=1}^{t} \|\boldsymbol{m}_s\|_{\boldsymbol{\Lambda}_s^{-1}}^2 \leq \frac{\eta(1-\beta_1)}{\sqrt{1-\beta_2}} \sum_{i=1}^{d'} \sum_{s=1}^{t} \sum_{j=1}^{s} \beta_2^{\frac{s-j}{2}} |g_{j,i}| = \frac{\eta(1-\beta_1)}{\sqrt{1-\beta_2}} \sum_{i=1}^{d'} \sum_{j=1}^{t} |g_{j,i}| \sum_{s=j}^{t} \beta_2^{\frac{s-j}{2}}$$

$$\leq \frac{\eta(1-\beta_1)}{\sqrt{1-\beta_2}(1-\sqrt{\beta_2})} \sum_{i=1}^{d'} \sum_{s=1}^{t} |g_{s,i}| \leq \frac{\eta(1-\beta_1)\sqrt{d'}}{(1-\sqrt{\beta_2})^2} \sum_{s=1}^{t} \|\boldsymbol{g}_s\|$$

$$\leq \frac{\eta(1-\beta_1)\sqrt{d'}}{(1-\sqrt{\beta_2})^2} \cdot \frac{\alpha_f R_2}{m} \sum_{s=1}^{t} F_s(\boldsymbol{w}_s).$$

Then, multiplying $\frac{2C_0}{(1-\beta_1)^2}$ on both sides, we get the desired result. When Assumption 3 holds, we get from (43) that

$$\sum_{s=1}^{t} \|\boldsymbol{m}_s\|_{\boldsymbol{\Lambda}_s^{-1}}^2 \leq \frac{\eta(1-\beta_1)}{\sqrt{1-\beta_2}} \sum_{i=1}^{d'} \sum_{s=1}^{t} \sqrt{\sum_{j=1}^{s} \beta_2^{s-j} g_{j,i}^2}$$

$$\leq \frac{\eta(1-\beta_1)}{\sqrt{1-\beta_2}} \sum_{i=1}^{d'} \sqrt{\sum_{s=1}^{t} \sum_{j=1}^{s} \beta_2^{s-j} g_{j,i}^2} \cdot \sqrt{t}$$

$$\leq \frac{\eta(1-\beta_1)\sqrt{t}}{\sqrt{1-\beta_2}} \sqrt{\sum_{i=1}^{d'} \sum_{s=1}^{t} \sum_{j=1}^{s} \beta_2^{s-j} g_{j,i}^2} \cdot \sqrt{d'}$$

$$\leq \frac{\eta(1-\beta_1)\sqrt{td'}}{\sqrt{1-\beta_2}} \sqrt{\sum_{s=1}^{t} \sum_{j=1}^{s} \beta_2^{s-j} \|\boldsymbol{g}_j\|^2}$$

$$\leq \frac{\eta(1-\beta_1)\sqrt{td'}}{\sqrt{1-\beta_2}} \sqrt{\sum_{s=1}^{t} \sum_{j=1}^{s} \beta_2^{s-j} \cdot 2L_f \left(\frac{R_2}{m}\right)^2 F_j(\boldsymbol{w}_j)}$$

$$\leq \frac{\eta(1-\beta_1)\sqrt{2L_f td'}}{\sqrt{1-\beta_2}} \sqrt{\frac{\sum_{j=1}^{t} F_j(\boldsymbol{w}_j)}{1-\beta_2} \left(\frac{R_2}{m}\right)^2}.$$

Then, multiplying $\frac{2C_0}{(1-\beta_1)^2}$ on both sides, and using Young's inequality, we get that

$$\frac{2C_0}{(1-\beta_1)^2} \sum_{s=1}^{t} \|\boldsymbol{m}_s\|_{\boldsymbol{\Lambda}_s^{-1}}^2 \leq \frac{2C_0 \eta \sqrt{2L_f td'}}{(1-\beta_1)(1-\beta_2)} \sqrt{\sum_{j=1}^{t} F_j(\boldsymbol{w}_j) \left(\frac{R_2}{m}\right)^2}$$

$$\leq \sum_{j=1}^{t} \frac{F_j(\boldsymbol{w}_j)}{8} + \frac{16(C_0\eta)^2 L_f d' t}{(1-\beta_1)^2 (1-\beta_2)^2} \cdot \left(\frac{R_2}{m}\right)^2.$$

$\square$

## F   DETAILED PROOF OF THEOREM 3 AND REMARK 2

The proof for Theorem 3 shares many similarities as the ones for Theorems 1 and 2. To start with, we also need to establish the gradient and Hessian norm of the model. We re-state the neural network model in (8) as follows,

$$\varphi_{\text{new}}^{(0)} = \boldsymbol{x} \in \mathbb{R}^d, \quad \varphi_{\text{new}}^{(i)} = \boldsymbol{\sigma}\left(\boldsymbol{W}^{(i)}\varphi_{\text{new}}^{(i-1)}\right) \in \mathbb{R}^m, \forall i \in [L],$$

$$\Phi_{\text{new}}(\boldsymbol{w}, \boldsymbol{x}) = \frac{1}{\sqrt{m}}\boldsymbol{a}^\top \varphi_{\text{new}}^{(L)} \in \mathbb{R}. \tag{44}$$

We also define for any $i \in [L]$,

$$\psi_{\text{new}}^{(i)} = \frac{\partial \Phi_{\text{new}}(\boldsymbol{w}, \boldsymbol{x})}{\partial \varphi_{\text{new}}^{(i)}} \in \mathbb{R}^m, \quad \Sigma_{\text{new}}^{(i)} = \text{Diag}\left[\boldsymbol{\sigma}'\left(\boldsymbol{W}^{(i)}\varphi_{\text{new}}^{(i-1)}\right)\right] \in \mathbb{R}^{m \times m}. \tag{45}$$

For another point $\tilde{\boldsymbol{w}} \in \mathbb{R}^{d'}$, we use $\tilde{\varphi}_{\text{new}}^{(i)}, \tilde{\psi}_{\text{new}}^{(i)}, \tilde{\Sigma}_{\text{new}}^{(i)}$ to denote the corresponding terms with respect to $\tilde{w}$. For the model defined in (8) or equally (44), we have a similar result as Lemma 1 in the following.

**Lemma 13.** *Generally, for any $r > 2/l$, let $\|\boldsymbol{W}^{(i)}\| \le r, \forall i \in [L]$ and $\|\boldsymbol{a}\| \le r$ . Also, let Assumptions 1 and 6 hold. When $\varphi_{\text{new}}^{(i)}, \varphi_{\text{new}}^{(i)}, \Sigma_{\text{new}}^{(i)}$ are defined in (44), for any $i \in [L]$, it holds that with $R_1' = R(lr)^L$,*

$$\left\|\varphi_{\text{new}}^{(i)}\right\| \le R_1', \quad \left\|\Sigma_{\text{new}}^{(i)}\right\| \le l, \quad \left\|\psi_{\text{new}}^{(i)}\right\| \le \frac{(lr)^L r}{\sqrt{m}}.$$

**Lemma 14.** *Given any $r > 2/l$, let*

$$R_2' = \sqrt{L}R_1'(lr)^{L+1}, \quad R_3' = \sqrt{L}(lr)^{2L+1}\left[(lr)^{L-1}\tilde{l}r^2 R_1' + 1\right].$$

*If Assumptions 1 and 6 hold, $\Phi_{\text{new}}(\boldsymbol{w}, \boldsymbol{x})$ is as in (8), $\|\boldsymbol{W}^{(i)}\| \le \frac{r}{2}, \forall i \in [L]$ and $\|\boldsymbol{a}\| \le r$, then*

$$\|\nabla_{\boldsymbol{w}}\Phi_{\text{new}}(\boldsymbol{w}, \boldsymbol{x})\| < \frac{R_2'}{\sqrt{m}}, \quad \|\nabla_{\boldsymbol{w}}^2\Phi_{\text{new}}(\boldsymbol{w}, \boldsymbol{x})\| < \frac{R_3'}{\sqrt{m}}, \quad \forall \boldsymbol{x} \in \mathbb{R}^d.$$

Note that $f$ is now logistic loss, which satisfies both Assumptions 2 and 3. Then, based on Lemma 14, we can derive the following results.

**Lemma 15.** *Under the same conditions in Lemma 14, if $f$ is logistic loss and $F_t(\boldsymbol{w}) = f(\Phi_{\text{new}}(\boldsymbol{w}, \boldsymbol{x}_t))$, then for any $t \ge 1$,*

$$\|\nabla F_t(\boldsymbol{w})\| \le \frac{R_2'}{\sqrt{m}} \cdot F_t(\boldsymbol{w}), \quad \lambda_{\min}(\nabla^2 F_t(\boldsymbol{w})) > -\frac{R_3'}{\sqrt{m}} \cdot F_t(\boldsymbol{w}).$$

*Also,*

$$\|\nabla F_t(\boldsymbol{w})\| \le \frac{R_2'}{2\sqrt{m}}\sqrt{F_t(\boldsymbol{w})}, \quad \lambda_{\min}(\nabla^2 F_t(\boldsymbol{w})) > -\frac{R_3'}{\sqrt{2m}} \cdot \sqrt{F_t(\boldsymbol{w})}.$$

Consequently, based on the above weakly-convex-like property, we can get the following result.

**Lemma 16.** *Under the same conditions of Lemma 14, let $\|\boldsymbol{W}^{(i)}\|, \|\boldsymbol{W}_t^{(i)}\| \le \frac{r}{2}, \forall i \in [L]$ and $\|\boldsymbol{w}_t - \boldsymbol{w}\| \le D$. If $\sqrt{m} \ge \frac{9D^2 R_3'}{2}$, then,*

$$F_t(\boldsymbol{w}_t) - F_t(\boldsymbol{w}) \le \langle \boldsymbol{w}_t - \boldsymbol{w}, \nabla F_t(\boldsymbol{w}_t) \rangle + \frac{F_t(\boldsymbol{w}_t) + F_t(\boldsymbol{w})}{8}.$$

We notice that Proposition 2 holds for any algorithms with the form in (9). Then, we should also estimate terms in the RHS of (11). Given this, we will provide three lemmas based on the following assumptions.

**Assumption 8.** *(a). Suppose that $\Phi(\boldsymbol{w}, \boldsymbol{x})$ is as in (8) or equally (44), $\{\boldsymbol{w}_s\}$ is generated by Algorithm 1, and $f$ is logsitic loss.*
*(b). Suppose that the following inequalities hold with probability at least $1 - 2(L+1)\exp(-m/2)$ with regard to the initialization: given some constants $D_{\ell_2} > 0, r > 2/l$ and some $\boldsymbol{w} \in \mathcal{X}$,*

$$\|\boldsymbol{w}_s - \boldsymbol{w}\| \le D_{\ell_2}, \quad \|\boldsymbol{W}^{(i)}\| \le \frac{r}{2}, \quad \|\boldsymbol{W}_s^{(i)}\| \le \frac{r}{2}, \quad \forall s \in [t], i \in [L]. \tag{46}$$

*Consequently Lemma 14 holds.*

**Lemma 17.** *Generally, let Assumption 8 hold. Then,*

$$\frac{\beta_1}{1 - \beta_1}\langle \boldsymbol{w}_t - \boldsymbol{w}, \boldsymbol{m}_t\rangle \le \frac{D_{\ell_2}\beta_1 R_2'}{\sqrt{m}}\sum_{s=1}^{t} F_s(\boldsymbol{w}_s).$$

**Lemma 18.** *Generally, let Assumption 8 hold. Then,*

$$(*) \le \frac{D_{\ell_2}^2 R_2'\sqrt{1-\beta_2}}{C_0\eta\sqrt{m}(1-\sqrt{\beta_2})}\sum_{s=1}^{t} F_s(\boldsymbol{w}_s) + \frac{\epsilon\|\boldsymbol{w} - \boldsymbol{w}_1\|^2}{2C_0\eta} - \frac{\|\boldsymbol{w} - \boldsymbol{w}_{t+1}\|_{\boldsymbol{\Lambda}_t}^2}{2C_0}.$$

**Lemma 19.** *Generally, let Assumption 8 hold, and $0 \le \beta_1 \le \beta_2 < 1$. Then,*

$$\frac{2C_0}{(1-\beta_1)^2}\sum_{s=1}^{t}\|\boldsymbol{m}_s\|_{\boldsymbol{\Lambda}_s^{-1}}^2 \le \frac{C_0\eta(R_2')^2}{2\epsilon m(1-\beta_1)^2}\sum_{s=1}^{t} F_s(\boldsymbol{w}_s).$$

Based on the above lemmas and Proposition 2, we can prove the main regret bound in Theorems 3 and 7.

**Proof of Theorem 3 and the detailed version Theorem 7.** We can use an induction argument to prove (46) with $D_{\ell_2}$ and $r$ defined in Theorem 7. First, we set $\boldsymbol{w} = \boldsymbol{w}^{(\epsilon/T)} \in \mathcal{X}$ satisfying Assumption 4.

**Case $k = 1$.** Using Lemma 2 and Assumption 4, we get that $\|\boldsymbol{w} - \boldsymbol{w}_1\| \le D_{\ell_2}$ and $\|\boldsymbol{W}^{(i)}\|, \|\boldsymbol{W}_1^{(i)}\| \le \frac{r}{4}, \forall i \in [L]$. Also, we have $\|\boldsymbol{a}\| \le 3\sqrt{2} < r$ from Lemma 2.

**Case $k = t + 1$.** Suppose that (46) holds for any $s \le t$ with $D_{\ell_2}$, $r$ and $\boldsymbol{w}$ defined above. Then, Assumption 8 is satisfied. Based on Assumption 8 and the requirement of $m$ in (17), we can get from Lemma 16 that

$$\sum_{s=1}^{t}\frac{7F_s(\boldsymbol{w}_s)}{8} - \sum_{s=1}^{t}\frac{9F_s(\boldsymbol{w})}{8} \le \sum_{s=1}^{t}\langle \boldsymbol{w}_s - \boldsymbol{w}, \nabla F_s(\boldsymbol{w}_s)\rangle.$$

Similarly, we get from Lemmas 17 to 19 that

$$\frac{\beta_1}{1-\beta_1}\langle \boldsymbol{w}_t - \boldsymbol{w}, \boldsymbol{m}_t\rangle \le \sum_{s=1}^{t}\frac{F_s(\boldsymbol{w}_s)}{8},$$

$$(*) \le \sum_{s=1}^{t}\frac{F_s(\boldsymbol{w}_s)}{8} + \frac{\epsilon\|\boldsymbol{w} - \boldsymbol{w}_1\|^2}{2C_0\eta} - \frac{\|\boldsymbol{w} - \boldsymbol{w}_{t+1}\|_{\boldsymbol{\Lambda}_t}^2}{2C_0},$$

$$\frac{2C_0}{(1-\beta_1)^2}\sum_{s=1}^{t}\|\boldsymbol{m}_s\|_{\boldsymbol{\Lambda}_s^{-1}}^2 \le \sum_{s=1}^{t}\frac{F_s(\boldsymbol{w}_s)}{8}.$$

Using Proposition 2 (since $\boldsymbol{w} \in \mathcal{X}$) with $\phi = 0$, and combining the above results, we get that

$$\sum_{s=1}^{t}\frac{F_s(\boldsymbol{w}_s)}{2} \le \frac{9}{8}\sum_{s=1}^{t} F_s(\boldsymbol{w}) + \frac{\epsilon\|\boldsymbol{w} - \boldsymbol{w}_1\|^2}{2C_0\eta} - \frac{\|\boldsymbol{w} - \boldsymbol{w}_{t+1}\|_{\boldsymbol{\Lambda}_t}^2}{2C_0}. \tag{47}$$

Re-arranging (47), and using $F_s(\cdot) \ge 0$ and $\boldsymbol{\Lambda}_t \succeq \frac{\epsilon}{\eta}\boldsymbol{I}$, we get that

$$\frac{\epsilon\|\boldsymbol{w} - \boldsymbol{w}_{t+1}\|^2}{2C_0\eta} \le \frac{\|\boldsymbol{w} - \boldsymbol{w}_{t+1}\|_{\boldsymbol{\Lambda}_t}^2}{2C_0} \le \frac{9}{8}\sum_{s=1}^{t} F_s(\boldsymbol{w}) + \frac{\epsilon\|\boldsymbol{w} - \boldsymbol{w}_1\|^2}{2C_0\eta} \le \frac{9\epsilon}{8} + \frac{\epsilon\|\boldsymbol{w} - \boldsymbol{w}_1\|^2}{2C_0\eta},$$

where the last inequality uses $\sum_{s=1}^{t} F_s(\boldsymbol{w}) \leq \sum_{s=1}^{T} F_s(\boldsymbol{w}) \leq \epsilon$ from Assumption 4. Based on $C_0\eta \leq 1$ and the norm inequality, we get that $\|\boldsymbol{W}^{(i)} - \boldsymbol{W}_{t+1}^{(i)}\|^2 \leq \|\boldsymbol{w} - \boldsymbol{w}_{t+1}\|^2 \leq \frac{9}{4} + g^2\left(\frac{\epsilon}{T}\right) \leq D_{\ell_2}^2 \leq \frac{r^2}{16}$, and $\|\boldsymbol{W}_{t+1}^{(i)}\| \leq \|\boldsymbol{W}^{(i)}\| + \|\boldsymbol{W}^{(i)} - \boldsymbol{W}_{t+1}^{(i)}\| \leq \frac{r}{2}, \forall i \in [L]$. With the induction, we prove that (46) holds for any $t \in [T]$. Based on (46) and $\inf_{\boldsymbol{w} \in \mathcal{X}} \sum_{s=1}^{t} F_s(\boldsymbol{w}) \geq 0$, we can use (47) to get the desired result.

*Proof of Remark 2.* We first prove that Assumption 4 can be implied by Assumption 5. Following the construction of $\boldsymbol{w}^{(\varepsilon)}$ in the proof of Proposition 1, we let $r = \frac{4(2C+\log(1/\varepsilon))}{\gamma} + \max\{12\sqrt{2}, 2/l\}$. We then get from Lemma 2 and Assumption 4 that for any $\boldsymbol{w} \in [\boldsymbol{w}^{(\varepsilon)}, \boldsymbol{w}_1]$, $\|\boldsymbol{W}^{(i)}\| \leq \frac{r}{2}, \forall i \in [L]$. In addition, since $r \geq 2/l$, we can use Lemma 14 (in Section F) to get that for any $\boldsymbol{w} \in [\boldsymbol{w}^{(\varepsilon)}, \boldsymbol{w}_1]$,

$$\|\nabla_{\boldsymbol{w}}\Phi(\boldsymbol{w}, \boldsymbol{x}_t)\| < \frac{R_2'}{\sqrt{m}}, \quad \|\nabla_{\boldsymbol{w}}^2\Phi(\boldsymbol{w}, \boldsymbol{x}_t)\| < \frac{R_3'}{\sqrt{m}}, \quad \forall t \geq 1,$$

where $R_2', R_3'$ are as in Lemma 14 with $r$ defined above. Then, with $m \geq \left(\frac{R_3'(2C+\log(1/\varepsilon))^2}{2\gamma^2}\right)^2$, we can follow the similar deduction to lower bound $y_t\Phi(\boldsymbol{w}^{(\varepsilon)}, \boldsymbol{x}_t)$ as in the proof of Proposition 1, which leads to that $y_t\Phi(\boldsymbol{w}^{(\varepsilon)}, \boldsymbol{x}_t) \geq \log(1/\varepsilon)$. Hence, we can prove that $F_t(\boldsymbol{w}^{(\varepsilon)}) \leq \varepsilon$, and $\|\boldsymbol{w}^{(\varepsilon)} - \boldsymbol{w}_1\| = \frac{2C+\log(1/\varepsilon)}{\gamma}$ given that $f$ is logistic loss. Since $R_3' \sim \mathcal{O}(r^L) \sim \mathcal{O}(\log^L(1/\varepsilon))$, we get that

$$g\left(\frac{\epsilon}{T}\right) = \mathcal{O}\left(\log^L\left(\frac{T}{\epsilon}\right)\right), \quad \text{when} \quad m \geq \mathcal{O}\left(\log^L(T/\epsilon)\right).$$

Combining with Theorem 3, we can easily obtain the desired result. $\qquad\square$

### F.1 DETAILED PROOF OF LEMMAS IN THIS SECTION

*Proof of Lemma 13.* For the model in (8) or equally (44) with $\sigma(0) = 0$, we get that for any $i \in [L]$,

$$\left\|\boldsymbol{\varphi}_{\text{new}}^{(i)}\right\| = \left\|\boldsymbol{\sigma}\left(\boldsymbol{W}^{(i)}\boldsymbol{\varphi}_{\text{new}}^{(i-1)}\right)\right\| \leq \left\|\boldsymbol{\sigma}\left(\boldsymbol{W}^{(i)}\boldsymbol{\varphi}_{\text{new}}^{(i-1)}\right) - \boldsymbol{\sigma}(\boldsymbol{0}_m)\right\| + \|\boldsymbol{\sigma}(\boldsymbol{0}_m)\|$$

$$\leq l\left\|\boldsymbol{W}^{(i)}\boldsymbol{\varphi}_{\text{new}}^{(i-1)}\right\| \leq (lr)^i\|\boldsymbol{x}\| \leq R(lr)^L = R_1',$$

where we use $\|\boldsymbol{x}\| \leq R$ from Assumption 1. Using the definition of $\Sigma_{\text{new}}^{(i)}$, and noting that $\sigma$ is $l$-Lipschitz continuous, we get that

$$\left\|\Sigma_{\text{new}}^{(i)}\right\| = \left\|\boldsymbol{\sigma}'\left(\boldsymbol{W}^{(i)}\boldsymbol{\varphi}_{\text{new}}^{(i-1)}\right)\right\|_{\infty} \leq l.$$

Using the definition of $\boldsymbol{\psi}_{\text{new}}^{(i)}$, we get that

$$\boldsymbol{\psi}_{\text{new}}^{(i)} = \left(\boldsymbol{W}^{(i)}\right)^{\top}\Sigma_{\text{new}}^{(i)}\boldsymbol{\psi}_{\text{new}}^{(i+1)}, \quad \forall i \in [L-1], \quad \boldsymbol{\psi}_{\text{new}}^{(L)} = \frac{1}{\sqrt{m}}\boldsymbol{a}^{\top}.$$

Then, using the norm inequality, $r \geq 2/l$ and $\|\boldsymbol{a}\| \leq r$, for any $i \in [L-1]$,

$$\left\|\boldsymbol{\psi}_{\text{new}}^{(i)}\right\| \leq \left\|\boldsymbol{W}^{(i)}\right\|\left\|\Sigma_{\text{new}}^{(i)}\boldsymbol{\psi}_{\text{new}}^{(i+1)}\right\| \leq \left\|\boldsymbol{W}^{(i)}\right\|\left\|\Sigma_{\text{new}}^{(i)}\right\|\left\|\boldsymbol{\psi}_{\text{new}}^{(i+1)}\right\|$$

$$< lr\left\|\boldsymbol{\psi}^{(i+1)}\right\| \leq \frac{1}{\sqrt{m}}(lr)^{L-i}\|\boldsymbol{a}\| \leq \frac{(lr)^L r}{\sqrt{m}}.$$

For $i = L$, we have

$$\left\|\boldsymbol{\psi}_{\text{new}}^{(L)}\right\| = \frac{1}{\sqrt{m}}\|\boldsymbol{a}^{\top}\| \leq \frac{r}{\sqrt{m}}.$$

We then derive the desired bound. $\qquad\square$

*Proof of Lemma 14.* During the following proof, we use $\varphi, \psi, \Sigma, \tilde{\varphi}, \tilde{\psi}, \tilde{\Sigma}$ without "new" to denote the terms in (44) and (45) for the notation simplicity. By the chain rule, we have for any $i \in [L]$,

$$\frac{\partial \Phi_{\text{new}}(\boldsymbol{w}, \boldsymbol{x})}{\partial \boldsymbol{W}^{(i)}} = \Sigma^{(i)} \boldsymbol{\psi}^{(i)} \left( \boldsymbol{\varphi}^{(i-1)} \right)^{\top} \in \mathbb{R}^{m \times m}.$$

Then, using the norm inequality, we get that for any $i \in [L]$,

$$\left\| \frac{\partial \Phi_{\text{new}}(\boldsymbol{w}, \boldsymbol{x})}{\partial \boldsymbol{W}^{(i)}} \right\|_F \leq \left\| \Sigma^{(i)} \right\| \left\| \boldsymbol{\psi}^{(i)} \left( \boldsymbol{\varphi}^{(i-1)} \right)^{\top} \right\|_F \leq \left\| \Sigma^{(i)} \right\| \left\| \boldsymbol{\psi}^{(i)} \right\| \left\| \boldsymbol{\varphi}^{(i-1)} \right\|.$$

Using Lemma 13, we get that

$$\max_{i \in [L]} \left\| \frac{\partial \Phi_{\text{new}}(\boldsymbol{w}, \boldsymbol{x})}{\partial \boldsymbol{W}^{(i)}} \right\|_F \leq \frac{R_1'(lr)^{L+1}}{\sqrt{m}}.$$

Note that $\nabla_{\boldsymbol{w}} \Phi_{\text{new}}(\boldsymbol{w}, \boldsymbol{x})$ is a vector and its $\ell_2$-norm is bounded by

$$\| \nabla_{\boldsymbol{w}} \Phi_{\text{new}}(\boldsymbol{w}, \boldsymbol{x}) \| = \sqrt{\sum_{i=1}^{L} \left\| \frac{\partial \Phi_{\text{new}}(\boldsymbol{w}, \boldsymbol{x})}{\partial \boldsymbol{W}^{(i)}} \right\|_F^2} < \frac{\sqrt{L} R_1'(lr)^{L+1}}{\sqrt{m}} = \frac{R_2'}{\sqrt{m}}.$$

For any $\boldsymbol{w}$ such that $\| \boldsymbol{W}^{(i)} \| \leq \frac{r}{2}, \forall i \in [L]$, we let $\tilde{w} \in \mathcal{B}(\boldsymbol{w}, \Delta), \Delta \leq \frac{r}{2}$. Then, we get that $\| \tilde{\boldsymbol{W}}^{(i)} \| \leq r, \forall i \in [L]$. Then, for each layer $i \in [L]$, we have the same decomposition as (22),

$$\begin{aligned}
\left\| \frac{\partial \Phi_{\text{new}}(\boldsymbol{w}, \boldsymbol{x})}{\partial \boldsymbol{W}^{(i)}} - \frac{\partial \Phi_{\text{new}}(\tilde{\boldsymbol{w}}, \boldsymbol{x})}{\partial \tilde{\boldsymbol{W}}^{(i)}} \right\|_F &\leq \left\| \left( \Sigma^{(i)} - \tilde{\Sigma}^{(i)} \right) \boldsymbol{\psi}^{(i)} \left( \boldsymbol{\varphi}^{(i-1)} \right)^{\top} \right\|_F \\
&+ \left\| \tilde{\Sigma}^{(i)} \left( \boldsymbol{\psi}^{(i)} - \tilde{\boldsymbol{\psi}}^{(i)} \right) \left( \boldsymbol{\varphi}^{(i-1)} \right)^{\top} \right\|_F \\
&+ \left\| \tilde{\Sigma}^{(i)} \tilde{\boldsymbol{\psi}}^{(i)} \left( \boldsymbol{\varphi}^{(i-1)} - \tilde{\boldsymbol{\varphi}}^{(i-1)} \right)^{\top} \right\|_F. \quad (48)
\end{aligned}$$

Using the definition of $\boldsymbol{\varphi}^{(i)}$ from (44), the norm inequality and Lemma 13, we get that for any $i \in [L]$,

$$\begin{aligned}
\left\| \boldsymbol{\varphi}^{(i)} - \tilde{\boldsymbol{\varphi}}^{(i)} \right\| &= \left\| \boldsymbol{\sigma} \left( \boldsymbol{W}^{(i)} \boldsymbol{\varphi}^{(i-1)} \right) - \boldsymbol{\sigma} \left( \tilde{\boldsymbol{W}}^{(i)} \tilde{\boldsymbol{\varphi}}^{(i-1)} \right) \right\| \\
&\leq l \left\| \boldsymbol{W}^{(i)} \boldsymbol{\varphi}^{(i-1)} - \tilde{\boldsymbol{W}}^{(i)} \tilde{\boldsymbol{\varphi}}^{(i-1)} \right\| \\
&\leq l \left\| \boldsymbol{W}^{(i)} \left( \boldsymbol{\varphi}^{(i-1)} - \tilde{\boldsymbol{\varphi}}^{(i-1)} \right) \right\| + l \left\| \left( \boldsymbol{W}^{(i)} - \tilde{\boldsymbol{W}}^{(i)} \right) \tilde{\boldsymbol{\varphi}}^{(i-1)} \right\| \\
&\leq lr \left\| \boldsymbol{\varphi}^{(i-1)} - \tilde{\boldsymbol{\varphi}}^{(i-1)} \right\| + l \Delta R_1' \leq l \Delta R_1' \sum_{j=0}^{i-1} (lr)^j,
\end{aligned}$$

where we use that $\sigma$ is $l$-Lipschitz continuous in the first inequality, Lemma 13 in the third inequality, and $\boldsymbol{\varphi}^{(0)} = \tilde{\boldsymbol{\varphi}}^{(0)} = \boldsymbol{x}$ in the last one. With $r \geq 2/l$, we thereby derive that

$$\max_{i \in [L]} \left\| \boldsymbol{\varphi}^{(i)} - \tilde{\boldsymbol{\varphi}}^{(i)} \right\| \leq \frac{(lr)^L l \Delta R_1'}{lr - 1} \leq (lr)^L l \Delta R_1'.$$

Using that $\sigma'$ is $\tilde{l}$-Lipschitz continuous and following a similar deduction for bounding $\left\| \boldsymbol{\varphi}^{(i)} - \tilde{\boldsymbol{\varphi}}^{(i)} \right\|$ above, we get that

$$\begin{aligned}
\left\| \Sigma^{(i)} - \tilde{\Sigma}^{(i)} \right\| &= \left\| \sigma' \left( \boldsymbol{W}^{(i)} \boldsymbol{\varphi}^{(i-1)} \right) - \sigma' \left( \tilde{\boldsymbol{W}}^{(i)} \tilde{\boldsymbol{\varphi}}^{(i-1)} \right) \right\|_\infty \\
&\leq \tilde{l} \left\| \boldsymbol{W}^{(i)} \boldsymbol{\varphi}^{(i-1)} - \tilde{\boldsymbol{W}}^{(i)} \tilde{\boldsymbol{\varphi}}^{(i-1)} \right\|_\infty \\
&\leq \tilde{l} \left\| \boldsymbol{W}^{(i)} \boldsymbol{\varphi}^{(i-1)} - \tilde{\boldsymbol{W}}^{(i)} \tilde{\boldsymbol{\varphi}}^{(i-1)} \right\| \leq (lr)^L \tilde{l} \Delta R_1'.
\end{aligned}$$

Then, we have

$$\left\| \boldsymbol{\psi}^{(i)} - \tilde{\boldsymbol{\psi}}^{(i)} \right\| \leq \left\| \left( \Sigma^{(i)} - \tilde{\Sigma}^{(i)} \right) \boldsymbol{W}^{(i)} \boldsymbol{\psi}^{(i+1)} \right\| + \left\| \tilde{\Sigma}^{(i)} \left( \boldsymbol{W}^{(i)} - \tilde{\boldsymbol{W}}^{(i)} \right) \boldsymbol{\psi}^{(i+1)} \right\|$$

$$+ \left\| \tilde{\Sigma}^{(i)} \tilde{\boldsymbol{W}}^{(i)} \left( \boldsymbol{\psi}^{(i+1)} - \tilde{\boldsymbol{\psi}}^{(i+1)} \right) \right\|$$

$$\leq \frac{(lr)^{2L} \tilde{l} r^2 \Delta R_1'}{\sqrt{m}} + \frac{(lr)^{L+1} \Delta}{\sqrt{m}} + lr \left\| \boldsymbol{\psi}^{(i+1)} - \tilde{\boldsymbol{\psi}}^{(i+1)} \right\| |$$

$$\leq \frac{\left( (lr)^{2L} \tilde{l} r^2 R_1' + (lr)^{L+1} \right) \Delta}{\sqrt{m}} \cdot \sum_{j=0}^{L-i-1} (lr)^j + (lr)^{L-i} \| \boldsymbol{a} - \boldsymbol{a} \|$$

$$\leq \frac{\left( (lr)^{2L} \tilde{l} r^2 R_1' + (lr)^{L+1} \right) (lr)^L \Delta}{\sqrt{m}}.$$

For $i = L$, we get that

$$\left\| \boldsymbol{\psi}^{(L)} - \tilde{\boldsymbol{\psi}}^{(L)} \right\| = \frac{1}{\sqrt{m}} \| \boldsymbol{a} - \boldsymbol{a} \| = 0.$$

Based on the above results and (48), we get that if $\| \boldsymbol{w} - \tilde{\boldsymbol{w}} \| \leq \Delta < \frac{r}{2}$, then

$$\| \nabla_{\boldsymbol{w}} \Phi_{\text{new}}(\boldsymbol{w}, \boldsymbol{x}) - \nabla_{\tilde{\boldsymbol{w}}} \Phi_{\text{new}}(\tilde{\boldsymbol{w}}, \boldsymbol{x}) \|_F \leq \sqrt{\sum_{i=1}^{L} \left\| \frac{\partial \Phi_{\text{new}}(\boldsymbol{w}, \boldsymbol{x})}{\partial \boldsymbol{W}^{(i)}} - \frac{\partial \Phi_{\text{new}}(\tilde{\boldsymbol{w}}, \boldsymbol{x})}{\partial \tilde{\boldsymbol{W}}^{(i)}} \right\|_F^2}$$

$$\leq \frac{\Delta \sqrt{L} (lr)^{2L} \left[ (lr)^L \tilde{l} r^2 R_1' + lr \right]}{\sqrt{m}}$$

$$= \frac{\Delta R_3'}{\sqrt{m}}.$$

Using the definition of the Hessian matrix and the definition of the induced $\ell_2$-norm, we get that

$$\left\| \nabla_{\boldsymbol{w}}^2 \Phi_{\text{new}}(\boldsymbol{w}, \boldsymbol{x}) \right\| = \sup_{\| \boldsymbol{v} \| = 1} \left\| \nabla_{\boldsymbol{w}}^2 \Phi_{\text{new}}(\boldsymbol{w}, \boldsymbol{x}) \boldsymbol{v} \right\| \leq \frac{R_3'}{\sqrt{m}}.$$

$\square$

*Proof of Lemma 15.* Since $f$ is logistic loss, which is 1-self-bounded and $1/4$-smooth. Then, we can follow the proof of Lemma 3, and combine Lemma 14 to get that

$$\| \nabla F_t(\boldsymbol{w}) \| \leq |F_t'(\boldsymbol{w})| \| \nabla_{\boldsymbol{w}} \Phi_{\text{new}}(\boldsymbol{w}, \boldsymbol{x}_t) \| \leq \frac{\alpha_f R_2'}{\sqrt{m}} \cdot F_t(\boldsymbol{w})$$

$$\lambda_{\min}(\nabla_{\text{new}}^2 F_t(\boldsymbol{w})) \geq - \sqrt{|y_t F_t'(\boldsymbol{w})|^2 \| \nabla_{\boldsymbol{w}}^2 \Phi_{\text{new}}(\boldsymbol{w}, \boldsymbol{x}_t) \|^2} \geq - \frac{\alpha_f R_3'}{\sqrt{m}} \cdot F_t(\boldsymbol{w}).$$

Setting $\alpha_f = 1$, we get the desired result. Similarly, we can follow the proof of Lemma 4 and combine Lemma 14 and $L_f = 1/4$ to get the second desired result. $\square$

*Proof of Lemma 16.* First, we follow the proof of Lemma 11. Let $\boldsymbol{w}_a = \arg\max_{\boldsymbol{w} \in [\boldsymbol{w}, \boldsymbol{w}_t]} F_t(\boldsymbol{w})$. If $\boldsymbol{w}_a$ is the end point, we get that $\max_{\boldsymbol{w} \in [\boldsymbol{w}, \boldsymbol{w}_t]} F_t(\boldsymbol{w}) \leq \max\{F_t(\boldsymbol{w}), F_t(\boldsymbol{w}_t)\}$. Suppose that $\boldsymbol{w}_a \in (\boldsymbol{w}_1, \boldsymbol{w}_2)$. Using the optimal condition, we also get that $\langle \nabla F_t(\boldsymbol{w}_a), \boldsymbol{w}_2 - \boldsymbol{w}_1 \rangle = 0$. Then, using Taylor's expansion theorem, Lemma 15, and denoting $M_3 = \frac{R_3'}{\sqrt{m}}$,

$$F_t(\boldsymbol{w}) \geq F_t(\boldsymbol{w}_a) - \frac{M_3 D^2}{2} \cdot F_t(\boldsymbol{w}_a) = \left( 1 - \frac{M_3 D^2}{2} \right) F_t(\boldsymbol{w}_a).$$

Using the same deduction, we can also get that

$$F_t(\boldsymbol{w}_t) \geq \left( 1 - \frac{M_3 D^2}{2} \right) F_t(\boldsymbol{w}_a).$$

Combining the above, we get that

$$\max_{\boldsymbol{w} \in [\boldsymbol{w}, \boldsymbol{w}_t]} F_t(\boldsymbol{w}) \leq \left(1 - \frac{M_3 D^2}{2}\right)^{-1} \max\{F_t(\boldsymbol{w}), F_t(\boldsymbol{w}_t)\}. \tag{49}$$

Then, following the proof of Lemma 5 under self-bounded losses, and combining Lemma 14, we have

$$F_t(\boldsymbol{w}) \geq F_t(\boldsymbol{w}_t) + \langle \nabla F_t(\boldsymbol{w}_t), \boldsymbol{w} - \boldsymbol{w}_t \rangle - \frac{\|\boldsymbol{w} - \boldsymbol{w}_t\|^2}{2} \cdot \frac{R_3'}{\sqrt{m}} \cdot \max_{\boldsymbol{w} \in [\boldsymbol{w}, \boldsymbol{w}_t]} F_t(\boldsymbol{w}).$$

With the requirement of $m$ and (49), we can easily obtain that

$$\frac{\|\boldsymbol{w} - \boldsymbol{w}_t\|^2}{2} \cdot \frac{R_3'}{\sqrt{m}} \cdot \max_{\boldsymbol{w} \in [\boldsymbol{w}, \boldsymbol{w}_t]} F_t(\boldsymbol{w}) \leq \frac{D^2 R_3'}{2\sqrt{m}} \left(1 - \frac{R_3' D^2}{2\sqrt{m}}\right)^{-1} \max\{F_t(\boldsymbol{w}), F_t(\boldsymbol{w}_t)\}$$

$$\leq \frac{F_t(\boldsymbol{w}_t) + F_t(\boldsymbol{w})}{8},$$

which leads to the result. $\qquad\square$

*Proof of Lemma 17.* We use Cauchy-Schwarz inequality, Lemma 14 and $\|\boldsymbol{w}_t - \boldsymbol{w}\| \leq D_{\ell_2}$ from Assumption 8 to get that

$$\frac{\beta_1}{1 - \beta_1} \langle \boldsymbol{w}_t - \boldsymbol{w}, \boldsymbol{m}_t \rangle \leq \frac{D_{\ell_2} \beta_1}{1 - \beta_1} \left\| (1 - \beta_1) \sum_{s=1}^{t} \beta_1^{t-s} \boldsymbol{g}_s \right\|$$

$$\leq D_{\ell_2} \beta_1 \sum_{s=1}^{t} \|\boldsymbol{g}_s\| \leq \frac{D_{\ell_2} \beta_1 R_2'}{\sqrt{m}} \sum_{s=1}^{t} F_s(\boldsymbol{w}_s).$$

$\qquad\square$

*Proof of Lemma 18.* We first note that (39) and (41) in the proof of Lemma 7 remain unchanged. Also, (42) still holds under Assumption 2. Then, we sum up (39) and (41) and use Lemma 7 to get that

$$(*) \leq \frac{\epsilon \|\boldsymbol{w} - \boldsymbol{w}_1\|^2}{2C_0 \eta} + \frac{D_{\ell_2}^2 \sqrt{1 - \beta_2}}{C_0 \eta (1 - \sqrt{\beta_2})} \sum_{s=1}^{t} \|\boldsymbol{g}_s\| - \frac{\|\boldsymbol{w} - \boldsymbol{w}_{t+1}\|_{\boldsymbol{\Lambda}_t}^2}{2C_0}$$

$$\leq \frac{\epsilon \|\boldsymbol{w} - \boldsymbol{w}_1\|^2}{2C_0 \eta} + \frac{D_{\ell_2}^2 \sqrt{1 - \beta_2}}{C_0 \eta (1 - \sqrt{\beta_2})} \cdot \frac{R_2'}{\sqrt{m}} \sum_{s=1}^{t} F_s(\boldsymbol{w}_s) - \frac{\|\boldsymbol{w} - \boldsymbol{w}_{t+1}\|_{\boldsymbol{\Lambda}_t}^2}{2C_0},$$

where we use Lemmas 14 and 15 instead of Lemmas 1 and 3. $\qquad\square$

*Proof of Lemma 19.* The proof is a bit different from Lemma 8. Here, we directly have

$$\|\boldsymbol{m}_s\|_{\boldsymbol{\Lambda}_s^{-1}}^2 = \sum_{i=1}^{d'} \frac{\eta m_{s,i}^2}{\epsilon + \sqrt{v_{s,i}}} \leq \frac{\eta \|\boldsymbol{m}_s\|^2}{\epsilon} \leq \frac{\eta}{\epsilon} \left\| (1 - \beta_1) \sum_{j=1}^{s} \beta_1^{s-j} \boldsymbol{g}_j \right\|^2 \leq \frac{\eta(1 - \beta_1)}{\epsilon} \sum_{j=1}^{s} \beta_1^{s-j} \|\boldsymbol{g}_j\|^2,$$

where the last inequality uses the convexity of $\|\cdot\|^2$. Summing up over $s \in [t]$, and using Lemma 15, we have

$$\sum_{s=1}^{t} \|\boldsymbol{m}_s\|_{\boldsymbol{\Lambda}_s^{-1}}^2 \leq \frac{\eta(1 - \beta_1)}{\epsilon} \sum_{s=1}^{t} \sum_{j=1}^{s} \beta_1^{s-j} \|\boldsymbol{g}_j\|^2$$

$$= \frac{\eta(1 - \beta_1)}{\epsilon} \sum_{j=1}^{t} \|\boldsymbol{g}_j\|^2 \sum_{s=j}^{t} \beta_1^{s-j}$$

$$\leq \frac{\eta}{\epsilon} \sum_{s=1}^{t} \|\boldsymbol{g}_s\|^2 \leq \frac{\eta(R_2')^2}{4\epsilon m} \sum_{s=1}^{t} F_s(\boldsymbol{w}_s).$$

Finally, multiplying $\frac{2C_0}{(1 - \beta_1)^2}$ on both sides, we get the desired result. $\qquad\square$

# G  DETAILED REGRET BOUNDS IN SECTION 6 AND THEIR PROOF

We first provide the detailed version of Theorem 4.

**Theorem 8** (Under self-bounded losses). *Let $\{\boldsymbol{w}_t\}_{t\in[T]}$ be generated by Algorithm 1 with $\boldsymbol{\Lambda}_t$ replaced by* (14). *Following the same conditions of Theorem 1, for any $0 < \beta_1 \leq \beta_2 < 1$ and $C_0, \eta, \epsilon > 0$, we define $D_{\ell_2} = \frac{3\sqrt{C_0\eta}}{2} + g\left(\frac{\epsilon}{T}\right)$ and let $R_1, R_2, R_3$ be the forms in* (15) *with $r = 4D_{\ell_2} + 12\sqrt{2}$. If $m \geq \max\{2\log(2(L+1)), 4l^2r^2, d\}$ and*

$$m \geq \max\left\{8D_{\ell_2}\alpha_f R_2 \max\left\{\beta_1, \frac{D_{\ell_2}\sqrt{1-\beta_2}}{C_0\eta(1-\sqrt{\beta_2})}\right\}, \frac{16C_0\eta\alpha_f R_2}{(1-\beta_1)(1-\sqrt{\beta_2})^2}, \frac{9\alpha_f D_{\ell_2}^2 R_3}{2}\right\}, \quad (50)$$

*then, with probability at least $1 - 2(L+1)\exp(-m/2)$ with regard to the initialization*

$$R(T) \leq \sum_{t=1}^{T} F_t(\boldsymbol{w}_t) \leq \frac{9\epsilon}{4} + \frac{g^2(\epsilon/T)\epsilon}{C_0\eta}.$$

**Theorem 9** (Under smooth losses). *Let $\{\boldsymbol{w}_t\}_{t\in[T]}$ be generated by Algorithm 1 with $\boldsymbol{\Lambda}_t$ replaced by* (14). *Following the same conditions of Theorem 2, for any $C_0, \eta, \epsilon > 0$ and $0 \leq \beta_1 < \beta_2 < 1$, we define*

$$D_{\ell_2}^2 = \frac{9}{4} + g^2\left(\frac{\epsilon}{T}\right) + \frac{1}{\epsilon}\left(\frac{8C_0\eta\beta_1^2 L_f}{1-\beta_1} + \frac{2L_f}{C_0\eta\beta_2} + \frac{32(C_0\eta)^3 L_f}{(1-\beta_1)^2(1-\beta_2)^2} + C_0\eta(4+\sqrt{2})\right).$$

*Let $R_1, R_2, R_3$ be the forms in* (15) *with $r = 4D_{\ell_2} + 12\sqrt{2}$. It's required that*

$$m \geq \max\left\{\max\{D_{\ell_2}^2, 1\}R_2\sqrt{T}, D_{\ell_2}R_2, D_{\ell_2}^2 R_3\sqrt{L_f T}, 2\log(2(L+1)), , 4l^2r^2, d\right\}. \quad (51)$$

*Then, with probability at least $1 - 2(L+1)\exp(-m/2)$ with regard to the initialization*

$$R(T) \leq \sum_{t=1}^{T} F_t(\boldsymbol{w}_t) \leq \frac{\epsilon D_{\ell_2}^2}{C_0\eta}.$$

Then, we provide the detailed proof. We note that Proposition 2 holds since it's algorithm-dependent. Then, we also have the following assumption based on the high probability event in Lemma 2.

**Assumption 9.** *(a). Suppose that $\Phi(\boldsymbol{w}, \boldsymbol{x})$ is as in* (3), *$\{\boldsymbol{w}_s\}$ is generated by Algorithm 1 with $\boldsymbol{\Lambda}_t$ replaced by* (14).
*(b). Suppose that the following inequalities hold with probability at least $1 - 2L\exp(-m/2)$ with regard to the initialization: given some constants $D_{\ell_2}, r > 0$ and some $\boldsymbol{w} \in \mathcal{X}$,*

$$\|\boldsymbol{w}_s - \boldsymbol{w}\| \leq D_{\ell_2}, \quad \|\boldsymbol{W}^{(i)}\| \leq \frac{r}{2}, \quad \|\boldsymbol{W}_s^{(i)}\| \leq \frac{r}{2}, \quad \forall s \in [t], i \in [L]. \quad (52)$$

*Also, let $\sqrt{m} \geq \max\{1, 2lr\}$. Consequently, Lemma 1 holds.*

We also note that under Assumption 9, Lemmas 5 to 7 remain unchanged since these results are independent of the specific form of $\boldsymbol{\Lambda}_t$. The major difference is that we provide the following lemma to replace Lemma 8.

**Lemma 20.** *Generally, let Assumption 9 hold and $0 \leq \beta_1 \leq \beta_2 < 1$. When Assumption 2 holds, it holds that*

$$\frac{2C_0}{(1-\beta_1)^2}\sum_{s=1}^{t}\|\boldsymbol{m}_s\|_{\boldsymbol{\Lambda}_s^{-1}}^2 \leq \frac{2C_0\eta\alpha_f R_2}{(1-\beta_1)(1-\sqrt{\beta_2})^2 m}\sum_{s=1}^{t}F_s(\boldsymbol{w}_s).$$

*When Assumption 3 holds, it holds that*

$$\frac{2C_0}{(1-\beta_1)^2}\sum_{s=1}^{t}\|\boldsymbol{m}_s\|_{\boldsymbol{\Lambda}_s^{-1}}^2 \leq \sum_{s=1}^{t}\frac{F_s(\boldsymbol{w}_s)}{8} + \frac{16(C_0\eta)^2 L_f t}{(1-\beta_1)^2(1-\beta_2)^2}\left(\frac{R_2}{m}\right)^2.$$

Then, we can prove the main regret bounds based on an induction argument.

*Proof of Theorem 8.* We set $\boldsymbol{w} = \boldsymbol{w}^{(\epsilon/T)} \in \mathcal{X}$ satisfying Assumption 4.

**Case $k = 1$.** Using Lemma 2 and Assumption 4, and $D_{\ell_2}, r$ defined in Theorem 8, we get that $\|\boldsymbol{w} - \boldsymbol{w}_1\| \le g\left(\frac{\epsilon}{T}\right) \le D_{\ell_2}$ and $\|\boldsymbol{W}_1^{(i)}\|, \|\boldsymbol{W}^{(i)}\| \le \frac{r}{4}, \forall i \in [L]$. Also, we have $\|\boldsymbol{a}\| \le 3\sqrt{2} < r$.

**Case $k = t + 1$.** For some $t \in [T]$, suppose that (52) holds for any $s \le t$ with $D_{\ell_2}$, $r$ and $\boldsymbol{w}$ defined above. Then, based on Assumption 9, we get that Lemmas 5 to 7 and 20 hold under self-bounded losses with $D = D_{\ell_2}$, and $r, R_1, R_2, R_3$ defined in Theorem 8. Since $\boldsymbol{w} \in \mathcal{X}$, we apply Proposition 2 with $\phi(\boldsymbol{w}) = 0$ and combining the above four lemmas, to get that when $m$ satisfies requirements in (50),

$$\frac{1}{2}\sum_{s=1}^{t} F_s(\boldsymbol{w}_s) \le \frac{9}{8}\sum_{s=1}^{t} F_s(\boldsymbol{w}) + \frac{\epsilon\|\boldsymbol{w} - \boldsymbol{w}_1\|^2}{2C_0\eta} - \frac{\|\boldsymbol{w} - \boldsymbol{w}_{t+1}\|_{\boldsymbol{\Lambda}_t}^2}{2C_0}. \tag{53}$$

Re-arranging (53) with $F_s(\cdot) \ge 0$, we get that

$$\frac{\epsilon\|\boldsymbol{w} - \boldsymbol{w}_{t+1}\|^2}{2C_0\eta} \le \frac{\|\boldsymbol{w} - \boldsymbol{w}_{t+1}\|_{\boldsymbol{\Lambda}_t}^2}{2C_0} \le \frac{9}{8}\sum_{s=1}^{t} F_s(\boldsymbol{w}) + \frac{\epsilon\|\boldsymbol{w} - \boldsymbol{w}_1\|^2}{2C_0\eta}.$$

With $\sum_{s=1}^{t} F_s(\boldsymbol{w}) \le \sum_{s=1}^{T} F_s(\boldsymbol{w}) \le \epsilon$ from Assumption 4, we get that

$$\|\boldsymbol{w} - \boldsymbol{w}_{t+1}\|^2 \le \frac{2C_0\eta}{\epsilon}\left(\frac{9\epsilon}{8} + \frac{\epsilon\|\boldsymbol{w} - \boldsymbol{w}_1\|^2}{2C_0\eta}\right) \le \frac{9C_0\eta}{4} + g^2\left(\frac{\epsilon}{T}\right) \le D_{\ell_2}^2.$$

Using the norm inequality, we get that for any $i \in [L]$,

$$\|\boldsymbol{W}^{(i)} - \boldsymbol{W}_{t+1}^{(i)}\| \le \|\boldsymbol{w} - \boldsymbol{w}_{t+1}\| \le \frac{r}{4}, \quad \|\boldsymbol{W}_{t+1}^{(i)}\| \le \|\boldsymbol{W}^{(i)} - \boldsymbol{W}_{t+1}^{(i)}\| + \|\boldsymbol{W}^{(i)}\| \le \frac{r}{2}.$$

Then, we prove that (52) holds for any $t \in [T]$ with $D_{\ell_2}$ and $r$ defined in Theorem 8. Based on (52), and $\inf_{\boldsymbol{w} \in \mathcal{X}} \sum_{t=1}^{T} F_t(\boldsymbol{w}) \ge 0$, we can use (53) to get that:

$$R(t) \le \sum_{s=1}^{t} F_s(\boldsymbol{w}_s) \le \frac{9\epsilon}{4} + \frac{\|\boldsymbol{w} - \boldsymbol{w}_1\|^2\epsilon}{C_0\eta} \le \frac{9\epsilon}{4} + \frac{g^2(\epsilon/T)\epsilon}{C_0\eta}.$$

$\square$

*Proof of Theorem 9.* We set $\boldsymbol{w} = \boldsymbol{w}^{(\epsilon/T)} \in \mathcal{X}$ satisfying Assumption 4.

**Case $k = 1$.** Using Lemma 2 and Assumption 4, and $D_{\ell_2}, r$ defined in Theorem 9, we get that $\|\boldsymbol{w} - \boldsymbol{w}_1\| \le g\left(\frac{\epsilon}{T}\right) \le D_{\ell_2}$ and $\|\boldsymbol{W}_1^{(i)}\|, \|\boldsymbol{W}^{(i)}\| \le \frac{r}{4}, \forall i \in [L]$. Also, we have $\|\boldsymbol{a}\| \le 3\sqrt{2} < r$.

**Case $k = t + 1$.** For some $t \in [T]$, suppose that (52) holds for any $s \le t$ with $D_{\ell_2}$, $r$ and $\boldsymbol{w}$ defined above. Then, based on Assumption 9, we get that Lemmas 5 to 7 and 20 hold under smooth losses with $D = D_{\ell_2}$, and $r, R_1, R_2, R_3$ defined in Theorem 9. Also, we use Proposition 2 with $\phi(\boldsymbol{w}) = 0$. Then, combining the above, we get that

$$\frac{1}{2}\sum_{s=1}^{t} F_s(\boldsymbol{w}_s) \le \frac{9}{8}\sum_{s=1}^{t} F_s(\boldsymbol{w}) - \frac{\|\boldsymbol{w} - \boldsymbol{w}_{t+1}\|_{\boldsymbol{\Lambda}_t}^2}{2C_0} + \frac{\epsilon\|\boldsymbol{w} - \boldsymbol{w}_1\|^2}{2C_0\eta} + \frac{4(D_{\ell_2}\beta_1)^2 L_f}{1 - \beta_1}\left(\frac{R_2}{m}\right)^2$$

$$+ \left(\frac{D_{\ell_2}^2 R_2}{C_0\eta m}\right)^2 \frac{L_f t}{\beta_2} + \frac{16(C_0\eta)^2 L_f t}{(1 - \beta_1)^2(1 - \beta_2)^2}\left(\frac{R_2}{m}\right)^2 + \frac{4 + \sqrt{2}}{2}. \tag{54}$$

We also get that $\frac{\|\boldsymbol{w} - \boldsymbol{w}_{t+1}\|_{\boldsymbol{\Lambda}_t}^2}{2C_0} \ge \frac{\epsilon\|\boldsymbol{w} - \boldsymbol{w}_{t+1}\|^2}{2C_0\eta}$. Then, with the learning rate setup and the width requirement in (51), we get from (54) that

$$\|\boldsymbol{w} - \boldsymbol{w}_{t+1}\|^2 \le \frac{C_0\eta}{\epsilon}\sum_{s=1}^{t}\left(\frac{9F_s(\boldsymbol{w})}{4} - F_s(\boldsymbol{w}_s)\right) + \|\boldsymbol{w} - \boldsymbol{w}_1\|^2 + \frac{8C_0\eta(D_{\ell_2}\beta_1)^2 L_f}{\epsilon(1 - \beta_1)}\left(\frac{R_2}{m}\right)^2$$

$$+ \left(\frac{D_{\ell_2}^2 R_2}{m}\right)^2 \frac{2L_f t}{\epsilon C_0\eta\beta_2} + \frac{32(C_0\eta)^3 L_f t}{\epsilon(1 - \beta_1)^2(1 - \beta_2)^2}\left(\frac{R_2}{m}\right)^2 + \frac{C_0\eta(4 + \sqrt{2})}{\epsilon}.$$

Using $\sum_{s=1}^{t} F_s(\boldsymbol{w}) \leq \sum_{s=1}^{T} F_s(\boldsymbol{w}) \leq \epsilon$ and the width requirement (6),

$$\|\boldsymbol{w} - \boldsymbol{w}_{t+1}\|^2 \leq D_{\ell_2}^2 - \frac{C_0\eta}{\epsilon} \sum_{s=1}^{t} F_s(\boldsymbol{w}_s) \leq D_{\ell_2}^2. \tag{55}$$

It's then derived that $\|\boldsymbol{w} - \boldsymbol{w}_{t+1}\| \leq D_{\ell_2} \leq \frac{r}{4}$. Using the norm inequality, one can also easily derive that $\|\boldsymbol{W}_{t+1}^{(i)}\| \leq \frac{r}{2}, \forall i \in [L]$. With the induction, we prove that (52) holds for any $t \in [T]$ with $D_{\ell_2}$ and $r$ defined in Theorem 9. Then, we derive the regret bound through (55) that

$$R(t) \leq \sum_{s=1}^{t} F_s(\boldsymbol{w}_s) \leq \frac{\epsilon D_{\ell_2}^2}{C_0\eta}, \quad \forall t \in [T].$$

□

*Proof of Lemma 20.* Recalling the definitions of $\boldsymbol{m}_s$ from Algorithm 1, and $\boldsymbol{\Lambda}_s^{-1}$ from (14), we get that

$$\|\boldsymbol{m}_s\|_{\boldsymbol{\Lambda}_s^{-1}}^2 = \frac{\eta\|\boldsymbol{m}_s\|^2}{\epsilon + \sqrt{v_s}} \leq \frac{\eta\left\|(1-\beta_1)\sum_{j=1}^{s}\beta_1^{s-j}\boldsymbol{g}_j\right\|^2}{\sqrt{(1-\beta_2)\sum_{j=1}^{s}\beta_2^{s-j}\|\boldsymbol{g}_j\|^2}} \leq \frac{\eta(1-\beta_1)}{\sqrt{1-\beta_2}} \frac{\sum_{j=1}^{s}\beta_1^{s-j}\|\boldsymbol{g}_j\|^2}{\sqrt{\sum_{j=1}^{s}\beta_2^{s-j}\|\boldsymbol{g}_j\|^2}}$$

$$\leq \frac{\eta(1-\beta_1)}{\sqrt{1-\beta_2}}\sqrt{\sum_{j=1}^{s}\beta_2^{s-j}\|\boldsymbol{g}_j\|^2} \leq \frac{\eta(1-\beta_1)}{\sqrt{1-\beta_2}}\sum_{j=1}^{s}\beta_2^{\frac{s-j}{2}}\|\boldsymbol{g}_j\|, \tag{56}$$

where the second inequality uses the convexity of $\|\cdot\|^2$ and $\beta_1 \in [0,1)$, and the third one applies $\beta_1 \leq \beta_2$. Then, summing up both sides over $s \in [t]$, and using Lemma 3,

$$\sum_{s=1}^{t}\|\boldsymbol{m}_s\|_{\boldsymbol{\Lambda}_s^{-1}}^2 \leq \frac{\eta(1-\beta_1)}{\sqrt{1-\beta_2}}\sum_{s=1}^{t}\sum_{j=1}^{s}\beta_2^{\frac{s-j}{2}}\|\boldsymbol{g}_j\| \leq \frac{\eta(1-\beta_1)}{\sqrt{1-\beta_2}(1-\sqrt{\beta_2})}\sum_{s=1}^{t}\|\boldsymbol{g}_s\|$$

$$\leq \frac{\eta(1-\beta_1)}{(1-\sqrt{\beta_2})^2} \cdot \frac{\alpha_f R_2}{m}\sum_{s=1}^{t}F_s(\boldsymbol{w}_s).$$

Then, multiplying $\frac{2C_0}{(1-\beta_1)^2}$ on both sides, and using Young's inequality, we get the desired result. When Assumption 3 holds, we get from (56) that

$$\sum_{s=1}^{t}\|\boldsymbol{m}_s\|_{\boldsymbol{\Lambda}_s^{-1}}^2 \leq \frac{\eta(1-\beta_1)}{\sqrt{1-\beta_2}}\sum_{s=1}^{t}\sqrt{\sum_{j=1}^{s}\beta_2^{s-j}\|\boldsymbol{g}_j\|^2}$$

$$\leq \frac{\eta(1-\beta_1)}{\sqrt{1-\beta_2}}\sqrt{\sum_{s=1}^{t}\sum_{j=1}^{s}\beta_2^{s-j}\|\boldsymbol{g}_j\|^2 \cdot \sqrt{t}}$$

$$\leq \frac{\eta(1-\beta_1)\sqrt{t}}{\sqrt{1-\beta_2}}\sqrt{\sum_{s=1}^{t}\sum_{j=1}^{s}\beta_2^{s-j} \cdot 2L_f\left(\frac{R_2}{m}\right)^2 F_j(\boldsymbol{w}_j)}$$

$$\leq \frac{\eta(1-\beta_1)\sqrt{2L_f t}}{\sqrt{1-\beta_2}}\sqrt{\frac{\sum_{j=1}^{t}F_j(\boldsymbol{w}_j)}{1-\beta_2}\left(\frac{R_2}{m}\right)^2}.$$

Then, multiplying $\frac{2C_0}{(1-\beta_1)^2}$ on both sides, we get that

$$\frac{2C_0}{(1-\beta_1)^2}\sum_{s=1}^{t}\|\boldsymbol{m}_s\|_{\boldsymbol{\Lambda}_s^{-1}}^2 \leq \frac{2C_0\eta\sqrt{2L_f t}}{(1-\beta_1)(1-\beta_2)}\sqrt{\sum_{j=1}^{t}F_j(\boldsymbol{w}_j)\left(\frac{R_2}{m}\right)^2}$$

$$\leq \sum_{j=1}^{t}\frac{F_j(\boldsymbol{w}_j)}{8} + \frac{16(C_0\eta)^2 L_f t}{(1-\beta_1)^2(1-\beta_2)^2} \cdot \left(\frac{R_2}{m}\right)^2.$$

□

# H  EXPERIMENT

We provide two simple experiments to complement our theoretical assumptions and regret bounds. We choose model (8) with the logistic loss and sigmoid activation. The initialization $\boldsymbol{w}_1$ follows the setup in (4). We use the optimizer Adam with $\beta_1 = 0.9, \beta_2 = 0.999$ to train neural networks. The learning rate is set through the grid search on $[10^{-6}, 10^{-5}, 10^{-4}, 10^{-3}, 10^{-2}, 10^{-1}]$.

**Supplement of Assumption 4.** We first use the noisy XOR data distrbution defined in (25). We generate a dataset with each sample of $d = 10$ dimensions and a total of $n = 10^7$ samples. Then, we set the total iteration number $T = n = 10^7$. At each iteration, we randomly select one data without replacement and use Adam to update the weights, following the online learning setting. The loss is denoted by $F_t(\boldsymbol{w})$ at $t$-th iteration. In this experiment, we fix $m = 25, L = 2$. Then, we set different accuracy $\varepsilon = (\varepsilon_i)_i = (10^{-2}, 10^{-3}, 10^{-4}, 10^{-5}, 10^{-6}, 10^{-7})$. We record the iteration $\boldsymbol{w}_{1/\varepsilon_i}$ which is the weight at $1/\varepsilon_i$-th iteration, and calculate $\sum_{t=1}^{T} F_t(\boldsymbol{w}_{1/\varepsilon_i})$ and $\|\boldsymbol{w}_{1/\varepsilon_i} - \boldsymbol{w}_1\|$. The results are listed as follows:

Table 1: Training a neural network of $m = 25, L = 2$ on noisy XOR data

| $\boldsymbol{w}_{1/\varepsilon_i}$ | $\sum_{t=1}^{T} F_t(\boldsymbol{w}_{1/\varepsilon_i})$ | $\|\boldsymbol{w}_{1/\varepsilon_i} - \boldsymbol{w}_1\|$ | $d_0$ |
|---|---|---|---|
| $\boldsymbol{w}_{10^2}$ | $7.045 \times 10^4$ | 10.439 | 2.269 |
| $\boldsymbol{w}_{10^3}$ | $2.023 \times 10^3$ | 47.751 | 6.913 |
| $\boldsymbol{w}_{10^4}$ | 30.623 | 56.258 | 6.108 |
| $\boldsymbol{w}_{10^5}$ | $6.043 \times 10^{-6}$ | 69.105 | 6.002 |
| $\boldsymbol{w}_{10^6}$ | $3.847 \times 10^{-7}$ | 71.180 | 5.152 |
| $\boldsymbol{w}_{10^7}$ | $3.842 \times 10^{-13}$ | 79.701 | 4.945 |

where $d_0 = \lfloor \frac{\|\boldsymbol{w}_{1/\varepsilon_i} - \boldsymbol{w}_1\|}{\log(1/\varepsilon_i)} \rfloor + 1$. The goal is to use $\boldsymbol{w}_{\varepsilon_i}$ as the approximation of $\boldsymbol{w}^{(\varepsilon_i)}$ defined in Assumption 4. The results indicate that for given $\varepsilon_i \geq 1/T$, $\boldsymbol{w}_{1/\varepsilon_i}$ satisfies the requirements of Assumption 4 since $d_0 \sim \mathcal{O}(1)$. Note that since $\sum_{t=1}^{T} F_t(\boldsymbol{w}_{10^7})/T \sim \mathcal{O}(10^{-20})$, the experiment results allow for $\varepsilon \geq 10^{-20}$ in Assumption 4, though still leaving a gap to arbitrary small $\varepsilon$ required in this assumption.

We also plot the curves of $\|\boldsymbol{w}_{1/\varepsilon_i} - \boldsymbol{w}_1\|$ and $\sum_{t=1}^{T} F_t(\boldsymbol{w}_{1/\varepsilon_i})$ in Figures 1 and 2. To better show that $\|\boldsymbol{w}_{1/\varepsilon_i} - \boldsymbol{w}_1\| = g(\varepsilon_i) \sim \mathcal{O}(\log(1/\varepsilon_i))$, we also plot a line $f(\varepsilon) = 7\log(1/\varepsilon)$ for comparison.

**Supplement of regret bounds.** We also record the regret bounds in the first experiment, where $\sum_{t=1}^{T} F_t(\boldsymbol{w}_t) = 407.248$ and $\sum_{t=1000}^{T} F_t(\boldsymbol{w}_t) = 7.166, T = 10^7$, which roughly complement the regret bound order of $\mathcal{O}(\text{poly}(L, \log T)$ in Theorems 1 and 3.

We also use the MNIST dataset with sample number $n = 70000$ and set the total iteration number $T = n/4$. At each iteration, we randomly select 4 data without replacement. Similarly, we record regret bounds under different width and depth neural networks as follows:

| $L$ | $m$ | $\sum_{t=1}^{T} F_t(\boldsymbol{w}_t)$ | $(d_1, d_2)$ |
|---|---|---|---|
| 2 | 92 | 2.094 | (10, 1) |
| 5 | 92 | 43.144 | (10, 5) |
| 10 | 92 | 94.221 | (10, 56) |

where $d_1 = \lfloor \frac{m}{\log(T)} \rfloor + 1$ and $d_2 = \lfloor \frac{\sum_{t=1}^{T} F_t(\boldsymbol{w}_t)}{\log(T)} \rfloor + 1$. These results roughly show that when $m \sim \mathcal{O}(\log(T))$, $\sum_{t=1}^{T} F_t(\boldsymbol{w}_t) \leq \mathcal{O}(\text{poly}(L, \log T)$, which complements our regret bounds.

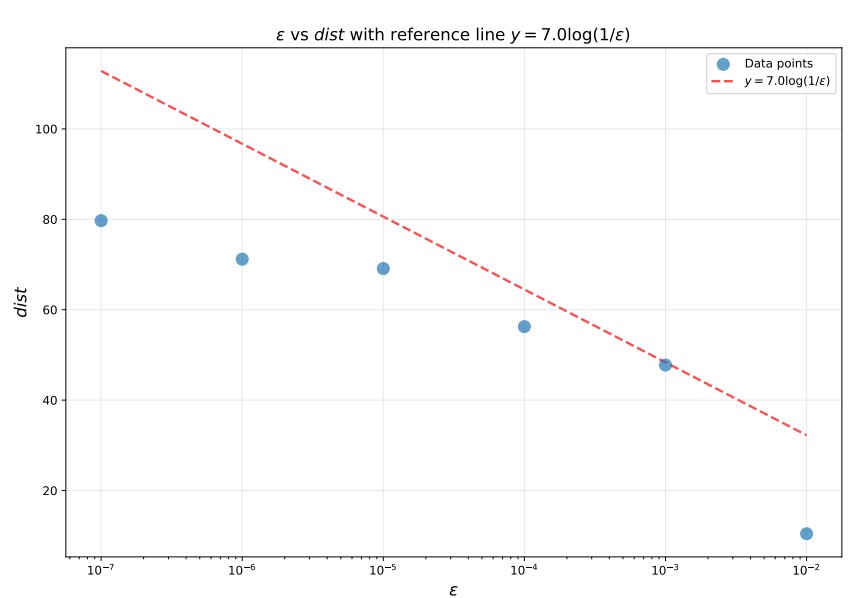

Figure 1: $\|\boldsymbol{w}_{1/\varepsilon_i} - \boldsymbol{w}_1\|$ vs. $\varepsilon_i$

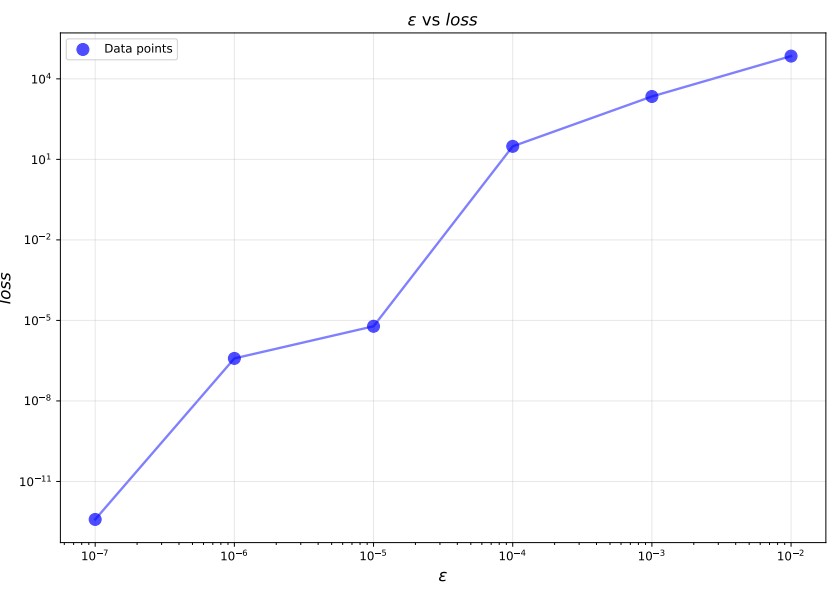

Figure 2: $\sum_{t=1}^{T} F_t(\boldsymbol{w}_{1/\varepsilon_i})$ vs. $\varepsilon_i$

