# OpenReview forum: "Regret Analysis of RMSProp and AdamNC for Training Deep Interpolating Neural Networks"
_ICLR.cc/2026/Conference — Submitted to ICLR 2026_

### Official Review · Reviewer_Csr8 · 2025-10-30

**Soundness:** 3
**Presentation:** 3
**Contribution:** 2
**Rating:** 6
**Confidence:** 3

**Summary:**

This paper gives a regret analysis of RMSProp/AdamNC when training deep smooth neural networks in an online manner specifically in the NTK interpolating setting. The main results are sufficient conditions on the width of the network and iterations numbers that ensure regret analysis and the derivation of the regret bounds which parallel online SGD bounds. The approach is based on exploiting the weak-convexity of the objective trained by losses such as logistic loss in the kernel regime.

**Strengths:**

The majority of the paper and the stated results are well-written. The studied problem is new, whereas prior works mainly tackled the convex case. The results show that even small networks of poly-logarithmic width can have favorable regret bounds. The analysis nicely extends the current GD analyses to ADAM and covers the constant step-size for this algorithm.

**Weaknesses:**

-The take-away message of the paper is not well-stated. In particular, a brief discussion on the following questions seems lacking from the current version: what are the distinctions between the bounds resulting from this over bounds resulting from using S/GD? Can ADAM improve upon GD in width requirements or final regret bounds? Are the current bounds tight? How does the analysis stand compared to the analysis of convex objectives?

-The analysis seems to rely on the known methods (especially (Taheri and Thrampoulidis, 2024)). It can be clarifying if you can include a discussion on the distinct steps from the current SGD analyses.


-Do experiments verify the bounds on the regret bounds or the width conditions?

**Questions:**

please see above section.

---

> ### Author Response · Authors · 2025-11-22
> **Responses to Weaknesses and Questions**
>
> We thank the reviewers for their valuable suggestions. We highlight the revised parts with **blue color** in the revised version.
>
> > *what are the distinctions between the bounds resulting from this over bounds resulting from using S/GD?* and *Can ADAM improve upon GD in width requirements or final regret bounds?*
>
> **Re. We have added the following sentences in Section 3.2.**
>
> **(Chen et al., 2021) required a minimal width of $\mathcal{O}(\text{poly}(L,\log T))$ comparable to the one in Theorem 1 whereas Taheri & Thrampoulidis (2024) required at least $\mathcal{O}(\text{poly}(\log^L T))$ width comparable that in Theorem 3. Note that Theorem 3 only requires parameter-free learning rates, while the two existing works require prior knowledge of problem parameters for learning rate setting, an adaptive advantage for Adam over GD/SGD that has been verified in stochastic optimizations. Both papers consider the logistic loss, and we further extend the regret bound to more general smooth or self-bounded losses. In addition, our regret bounds are implicitly better than GD when the gradients are sparse, which is also observed in AdaGrad (Duchi et al., 2011).**
>
> ---
>
> > *Are the current bounds tight?*
>
> **Re.** Our training loss bounds are comparable in order to the ones for GD in neural network optimization. However, since neural network training is a non-convex non-smooth optimization,  at this moment it remains unclear about the lower bound rate and whether the upper bound rate is tight or not.
>
> ---
>
> > *How does the analysis stand compared to the analysis of convex objectives?*
>
> **Re.** Some of the major proof differences lie in the following:
>
> - **The absence of bounded gradient and bounded iterate assumptions.** Most existing works for adaptive methods in online-convex optimization rely on the bounded gradient and iteration assumptions, including Duchi et al. (2011) and Alacaoglu et al. (2020). In comparison, we do not assume either of them. We rely on the more delicate bounds for gradient and Hessian norm of neural networks, such as Lemmas 3, 4, and Lemma 15, as well as other related lemmas, and an induction argument, to derive that the iteration remains in a bounded region. Further, combining a proper setup of the width, we can get the regret bounds without requiring bounded gradients.
>
> - **The non-monotonic step-sizes.** Duchi et al., (2011) and Alacaoglu et al. (2020) rely on the non-increasing step-sizes of AdaGrad/AMSGrad or AdamNC (using $\beta_{2t}=1-1/t$) to estimate the term $\sum\_{s=1}^t (B\_{\psi\_s}(w\_s,w)-B\_{\psi\_s}(w\_{s+1},w))$ in Proposition 2. In comparison, the step-sizes of RMSProp and AdamNC are non-monotonic when $\beta_1,\beta_2$ and $C_0\eta$ are constants. We use new bounds in Lemmas 7 and 18, as well as other related lemmas, to control this term, relying on the exponential moving average of RMSProp and AdamNC.
>
> ---
>
> > *The analysis seems to rely on the known methods (especially (Taheri and Thrampoulidis, 2024)). It can be clarifying if you can include a discussion on the distinct steps from the current SGD analyses.*
>
> **Re.** Some of the major proof differences lie in the following:
>
> - **Non-monotonic step-sizes.** Unlike a constant or non-increasing step-size of GD, the step-size of RMSProp and AdamNC is non-monotonic given only constant learning rates and $\beta_1,\beta_2$, leading to a challenge in controlling $\sum_{s=1}^t (B_{\psi_s}(w_s,w)-B_{\psi_s}(w_{s+1},w))$ in Proposition 2. We propose new bounds in Lemmas 7 and 18 to tackle this problem based on the exponential moving average of RMSProp and AdamNC.
>
> - **Gradient/Hessian norm.** Taheri and Thrampoulidis (2024) derive $\mathcal{O}(1)$ and $\mathcal{O}({1 \over \sqrt{m}})$ bounds for gradient and Hessian norm of shallow neural networks, respectively. We also note that Taheri et al., (2025) derive the same order bounds for deep neural networks. In comparison, we derive $\mathcal{O}({1\over m})$ bounds of both norms for model (3), and $\mathcal{O}({1\over \sqrt{m}})$ bounds for model (8).
>
> - **Curse of dimension.** When bounding the iteration norm, there is an additional $\mathcal{O}(d') \sim \mathcal{O}(m^2)$ factor since Adam is a coordinate-wise algorithm while GD/SGD use uniform step-sizes across all dimensions. To tackle this challenge, we use the $\mathcal{O}({1\over m})$ gradient bound to cancel this effect for model (3). For model (8), we derive a dimension-free bound for $\sum_{s=1}^t \|{\rm m}_s \|^2_{{\rm \Lambda}_s^{-1}}$.
>
> - **Unique lemmas for AdamNC.** Since AdamNC incorporates adaptive step-sizes which are determined by historical gradients, and additional momentum, there are some new bounds in comparison to GD, such as Lemmas 6 to 8.
>
> **We have added two paragraphs in Section 5 and Section B to clarify in detail the new challenges and solutions that arise when analyzing Adam in comparison to GD and other adaptive methods in neural network optimization or online convex optimization.**

---

> ### Author Response · Authors · 2025-11-22
> **Brief Presentation of Experiments**
>
> > *Do experiments verify the bounds on the regret bounds or the width conditions?*
>
> **Re. We also provide some simple experiments to complement this assumption, which are briefly presented as follows. All details are in Section H.**
>
> **Experiment**
>
> We use a fully-connected neural network with the logistic loss and sigmoid activation. The initialization $w_1$ follows the setup in our paper. We fix the desired accuracy $\varepsilon = 10^{-5}$ defined in Assumption 4. For each experiment, we have the input $(T, L, m)$ where $T$ is the total iteration number, $L$ is the layer number, and the width $m \sim\mathcal{O}( \log (1/\varepsilon))$. We use Adam to train neural networks. At $t$-th iteration, we uniformly sample data from the dataset, and the loss is $F_t(w)$.
>
> We first generate a dataset from the noisy XOR data distribution (Wei et al., 2019), which is a commonly used example satisfying NTK-separability (please refer to Examples 2 and 3). We record the last iteration $w_T$ and calculate terms listed below under different inputs:
>
> |   $(T,  L)$   | $({m},c_0)$ | $(\sum_{t=1}^T F_t(w_T),{\|w_{T}-w_1\|},d_1)$ | $(\sum_{t=1}^T F_t(w_t),d_2)$ |
> | :-----------: | :---------: | :-------------------------------------------: | :---------------------------: |
> | $(10^{6},2)$  |   $(9,1)$   |              $(31.794,32.871,4)$              |        $(424.557,33)$         |
> | $(10^{6},5)$  |  $(46,5)$   |              $(31.66,26.729,3)$               |        $(1102.49,120)$        |
> | $(10^{6},10)$ |  $(92,10)$  |              $(31.309,26.935,3)$              |        $(621.312, 67)$        |
>
> where $c_0 = \lfloor{m \over \log(1/\varepsilon)}\rfloor+1, d_1 = \lfloor {\|w_{T}-w_1\|\over\log(1/\varepsilon)}\rfloor +1, d_2 = \lfloor \sum_{t=1}^T F_t(w_t)/\log(1/\varepsilon)\rfloor+1$.
>
> We also test assumptions on the MNIST dataset. Similarly, we record the distance and the cumulative training loss as follows.
>
> |   $(T, L)$   | $({m},c\_0)$ | $(\sum\_{t=1}^T F\_t(w\_T), \|w\_{t}-w\_1\|,d\_1)$ | $(\sum\_{t=1}^T F\_t(w\_t),d\_2)$ |
> | :----------: | :---------: | :-------------------------------------------: | :---------------------------: |
> | $(10^{6},2)$ |  $(92,10)$  |              $(15.299,35.261,4)$              |          $(2.094,1)$          |
> | $(10^{6},5)$ |  $(92,10)$  |              $(15.862,18.434,2)$              |         $(43.144, 5)$         |
> | $(10^6,10)$  |  $(92,10)$  |              $(16.042,23.149,3)$              |        $(519.839, 56)$        |
>
> These results show that when $m \sim \mathcal{O}(\log(1/\varepsilon))$, we get that at least one point $w_T$ such that
>
> - $\sum_{t=1}^T F_t(w_T) \sim \mathcal{O}(1)$ and $\\|w_{T}-w_1\\| \le \mathcal{O}(\log(1/\varepsilon))$;
> - $\sum_{t=1}^T F_t(w_t) \le \mathcal{O}(\log(1/\varepsilon)) \sim \mathcal{O}(\log T)$ since $d_2 \sim \mathcal{O}(1)$.
>
> There exists one point $w_T$ such that $\sum_{t=1}^T F_t(w_T)/T \sim \mathcal{O}(1/T) $, which can be smaller than $\varepsilon$ when $T$ is larger than $1/\varepsilon$, such as $1/\varepsilon =10^{5}$ in this example. Hence, Assumption 4 holds, and the regret bounds match the order in our results.
>
> **For more details of these experiments, we refer to Section H in the revised version.**

---

> ### Author Response · Authors · 2025-11-28
> **correction on experiment related to noisy XOR data**
>
> Dear Reviewer Csr8,
>
> We apologized for a misleading between online learning and the random shuffle setting in the experiment using the noisy XOR dataset.
>
> **We have corrected this experiment in an online learning setting.** We first generate a dataset under a noisy XOR distribution with a sample number $10^7$, and set the total iteration number $T=10^7$. At each iteration, we randomly select a data without replacement and use Adam to update the weight. Then, we set different accuracy $\varepsilon =(\varepsilon_i)\_i =(10^{-2},10^{-3},10^{-4}, 10^{-5},10^{-6}, 10^{-7})$, and record the iteration $w_{1/\varepsilon_i}$ which is the weight at $1/\varepsilon_i$-th iteration during training. The goal is to use $w_{\varepsilon_i}$ as the approximation of $w^{(\varepsilon)}$ defined in Assumption 4. We test with $ \\|w_{\varepsilon_i} -w_1\\|$ and $\sum_{t=1}^T F_t(w_{\varepsilon_i})$. We also plot the curve $\\|w^\varepsilon-w_1\\|$ and  $\varepsilon$ along with $\sum_{t=1}^T F_t(w^\varepsilon)$.
> **For more details, please refer to Section H in the new revision.**
>
> At last, we acknowledge that these experiments could not fully verify but could only complement the assumptions and our regret bounds. The main contribution of this paper focuses on deriving, maybe (to our knowledge)  the first regret bound for AdamNC with constants $\beta_1 $ and $\beta_2$ in neural network optimization. We follow the interpolating assumption in some literature.
>
> We truly appreciate your valuable time and comments. If you have an opportunity, we would be grateful for any further comments or clarifications. Your feedback is highly valued. Thank you for your time and consideration.

---

> ### Author Response · Authors · 2025-12-01
> **Summary of Major Weaknesses from Reviewer Csr8 and Our Responses**
>
> **W1.**  **a brief discussion on the following seems lacking**: distinction and advantages of the derived results for AdamNC in this paper over GD in neural network optimization; analysis compared to that of convex objectives.
>
> **Re.** **We added more related discussions to fix this presentation issue.**  See (Lines 267-276; Lines 913-917) and  (Lines 918-952) of the revision, correspondingly.
>
> **W2.**  **analysis seems to rely on the know methods (especially (Taheri and Thrampoulidis, 2024) for GD)**
>
> **Re.** **Our proof relies on existing analysis while there are several challenges**:  non-monotonic step-sizes of AdamNC (with constants $\beta_1$ and $\beta_2$ that could lead to divergence in online convex optimization) compared to monotonic step-sizes of GD on neural network optimization; unbounded gradient and iterate norms that are usually considered bounded in online convex optimization, etc. We added two paragraphs (Lines 420–431 and Lines 918–952) in the revision.
>
> **W3.** Do experiments verify the bounds on the regret bounds or the width conditions?
>
> **Re.** Yes. We added two simple experiments (Section H) to complement them.

---

### Official Review · Reviewer_cAqU · 2025-10-31

**Soundness:** 3
**Presentation:** 3
**Contribution:** 2
**Rating:** 6
**Confidence:** 2

**Summary:**

This paper provides convergence guarantees of training a deep neural network with RMSProp and AdamNC under interpolation assumption. Their assumptions and proof resemble [1] in fashion: the two important points are using an approximate convexity of deep neural network, and using the interpolation point as a "reference point" that has low training loss. With the interpolation assumption the regret becomes a function of $g$ that dictates the interpolation property, which is potentially better than standard results for well-behaving $g$.

[1] Taheri, Hossein, and Christos Thrampoulidis. "Generalization and stability of interpolating neural networks with minimal width." Journal of Machine Learning Research 25.156 (2024): 1-41.

**Strengths:**

It is a solid contribution to extend certain results to different optimizers. Convergence guarantee better than O(\sqrt(T)) can be attained by certain assumptions look interesting.

Also, the paper is very well written and easy to understand the mathematical formalism. The lemmas are well stated, with exact assumptions, with theorems that look valid.

**Weaknesses:**

It would be better if more motivation was given for studying neural networks in the interpolation setting. Especially Assumption 4 seems like a very strong assumption to me, and I was a bit confused because in line 57-58 it states that the setting has been studied in different papers, whereas when I read the papers they do have min-margin assumptions but not exactly the one discussed in Assumption 4, except for [1]. So I have two questions:

- Is this theoretical assumption widely used in the exact form proposed in the paper? e.g. are there different papers that exactly show this form of assumption? If yes, it would be good to mention how they are used in different papers. If not but they are associated somehow, it would also be good to clarify it. It could be the case that I am missing something apparent.

- Is this theoretical assumption valid? e.g. is it verifyable by experiments? Are there any experiments that support this assumption?

Clarifying the questions would make the paper stronger.

**Questions:**

See weaknesses

---

> ### Author Response · Authors · 2025-11-22
> **Responses to Weaknesses and Questions**
>
> We thank the reviewers for their valuable suggestions. We highlight the revised parts with **blue color** in the revised version.
>
> > *Is this theoretical assumption widely used in the exact form proposed in the paper? e.g. are there different papers that exactly show this form of assumption? If yes, it would be good to mention how they are used in different papers. If not but they are associated somehow, it would also be good to clarify it. It could be the case that I am missing something apparent.*
>
> **Re.** We briefly discuss some literature that uses the same or similar realizable assumptions as ours.
>
> - Works like (Jacot et al., 2018, Cao and Gu, 2019, Ji and Telgarsky, 2020, Chen et al., 2023,Taheri et al., 2025) considered NTK-separability the same as Assumption 5 in our paper.
>
> - Chen et al., (2023) further used the following interpolating assumption: when $R=\\|w-w_1\\|\le \mathcal{O}(\log(1/\epsilon))$,
> $$
> \inf\_{F \in \mathcal{F}(w\_1,R)} \frac{1}{n}\sum\_{i=1}^n f(y_iF(x_i)) \le \epsilon + \rho\mathcal{O}(R).
> $$
> The assumption with $\rho=0$ is the same as ours (Assumption 4).  They considered offline finite-sum loss, while we consider the average regret bound, which in general is exchangeable.
>
> -  Taheri and Thrampoulidis,(2024) also considered the interpolating assumption the same as ours, also in the case of the offline finite-sum loss.
>
>
> In general, the interpolating realizable assumption in (Chen et al., 2023, Taheri and Thrampoulidis, 2024) is exchangeable to ours, and the NTK-separability assumption in the literature is the same as ours. In addition, in the offline setting, our results can easily lead to convergence results for full-batch Adam under exactly the same assumption in the literature.
>
>
> ---
>
> > *Is this theoretical assumption valid? e.g. is it verifiable by experiments? Are there any experiments that support this assumption?*
>
> **Re. We have added some more examples satisfying the interpolating assumption, such as Example 1  (linearly separable data), Examples 2 and 3 (the noisy XOR data) in Section D.**
>
> **We also provide some simple experiments to complement this assumption, which are briefly presented as follows. All details are in Section H.**

---

> > ### Author Response · Authors · 2025-11-22
> > **Brief Presentation of Experiments**
> >
> > **Experiments**
> >
> > We use a fully-connected neural network with the logistic loss and sigmoid activation. The initialization $w_1$ follows the setup in our paper. We fix the desired accuracy $\varepsilon = 10^{-5}$ defined in Assumption 4. For each experiment, we have the input $(T, L, m)$ where $T$ is the total iteration number, $L$ is the layer number, and the width $m \sim\mathcal{O}( \log (1/\varepsilon))$. We use Adam to train neural networks. At $t$-th iteration, we uniformly sample data from the dataset, and the loss is $F_t(w)$.
> >
> > We first generate a dataset from the noisy XOR data distribution (Wei et al., 2019), which is a commonly used example satisfying NTK-separability (please refer to Examples 2 and 3). We record the last iteration $w_T$ and calculate terms listed below under different inputs:
> >
> > |   $(T,  L)$   | $({m},c_0)$ | $(\sum_{t=1}^T F_t(w_T),{\|w_{T}-w_1\|},d_1)$ | $(\sum_{t=1}^T F_t(w_t),d_2)$ |
> > | :-----------: | :---------: | :-------------------------------------------: | :---------------------------: |
> > | $(10^{6},2)$  |   $(9,1)$   |              $(31.794,32.871,4)$              |        $(424.557,33)$         |
> > | $(10^{6},5)$  |  $(46,5)$   |              $(31.66,26.729,3)$               |        $(1102.49,120)$        |
> > | $(10^{6},10)$ |  $(92,10)$  |              $(31.309,26.935,3)$              |        $(621.312, 67)$        |
> >
> > where $c_0 = \lfloor{m \over \log(1/\varepsilon)}\rfloor+1, d_1 = \lfloor {\|w_{T}-w_1\|\over\log(1/\varepsilon)}\rfloor +1, d_2 = \lfloor \sum_{t=1}^T F_t(w_t)/\log(1/\varepsilon)\rfloor+1$.
> >
> > We also test assumptions on the MNIST dataset. Similarly, we record the distance and the cumulative training loss as follows.
> >
> > |   $(T, L)$   | $({m},c\_0)$ | $(\sum\_{t=1}^T F\_t(w\_T), \|w\_{t}-w\_1\|,d\_1)$ | $(\sum\_{t=1}^T F\_t(w\_t),d\_2)$ |
> > | :----------: | :---------: | :-------------------------------------------: | :---------------------------: |
> > | $(10^{6},2)$ |  $(92,10)$  |              $(15.299,35.261,4)$              |          $(2.094,1)$          |
> > | $(10^{6},5)$ |  $(92,10)$  |              $(15.862,18.434,2)$              |         $(43.144, 5)$         |
> > | $(10^6,10)$  |  $(92,10)$  |              $(16.042,23.149,3)$              |        $(519.839, 56)$        |
> >
> > These results show that when $m \sim \mathcal{O}(\log(1/\varepsilon))$, we get that at least one point $w_T$ such that
> >
> > - $\sum_{t=1}^T F_t(w_T) \sim \mathcal{O}(1)$ and $\\|w_{T}-w_1\\| \le \mathcal{O}(\log(1/\varepsilon))$;
> > - $\sum_{t=1}^T F_t(w_t) \le \mathcal{O}(\log(1/\varepsilon)) \sim \mathcal{O}(\log T)$ since $d_2 \sim \mathcal{O}(1)$.
> >
> > There exists one point $w_T$ such that $\sum_{t=1}^T F_t(w_T)/T \sim \mathcal{O}(1/T) $, which can be smaller than $\varepsilon$ when $T$ is larger than $1/\varepsilon$, such as $1/\varepsilon =10^{5}$ in this example. Hence, Assumption 4 holds, and the regret bounds match the order in our results.
> >
> > **For more details of these experiments, we refer to Section H in the revised version.**

---

> ### Comment · Reviewer_cAqU · 2025-11-26
>
> Dear reviewers,
>
> Thank you for the response and additional experiments. I have a few more questions / recommendations before I make my final decision.
>
> - In the current version of the manuscript it is still written that Ji and Telgarsky 2020 has a similar assumption with the realizable setup. Is this true or is Ji and Telgarsky only related with Assumption 5? It would be good to clarify.
>
> - Is it fair to understand assumption 4 as:
>
> for a certain initialization $w_1$, no matter what $\epsilon > 0$ is, we can find $w^{(\epsilon)}$ that has low regret and is "close enough" to $w_1$ (in the sense of g)?
>
> - How should we understand the experiment in Section H? I see that $\sum_{t=1}^{T} F_{t}(w_t)$ increases as $L$ increses but $T$ is fixed for MNIST. Can we say that $\sum_{t=1}^{T} F_{t}(w_t) \sim O(\log T)$?
>
> Also it seems to me like the experiments suggest that we can find a parameter $w_T$ with low risk and if we set $T$ sufficiently large we can find the parameter that is close to $w_1$ and having low risk. Assumption 4 is written as if $\epsilon$ and $T$ are independent. Can $T$ depend on $\epsilon$?
>
> At last, it would be helpful to see a curve $||w^{\epsilon} - w||$ for different $\epsilon$ along with $\sum_{t=1}^{T} F_t(w^{\epsilon})/T$ to see how $g$ looks in reality.

---

> ### Author Response · Authors · 2025-11-28
> **We truly appreciate your valuable time for the discussions and your helpful comments**
>
> We understand how busy you are with a heavy review load, and we sincerely thank you for making time to join the discussion and for giving us the opportunity to clarify. We also thank you again for your valuable suggestions.
>
> > *In the current version of the manuscript it is still written that Ji and Telgarsky 2020 has a similar assumption with the realizable setup. Is this true or is Ji and Telgarsky only related with Assumption 5? It would be good to clarify.*
>
> **Re.** Ji and Telgarsky, (2024) only used the NTK-separability (i.e., Assumption 5 in our paper). We have corrected the related statments and citations accordingly, such as in Line 57, Line 128, and Line 136. Also, we add some literature that using the NTK-separability in Line 142.
>
> ---
>
> > *Is it fair to understand assumption 4 as...*
>
> **Re.** We think that the understanding is correct and accurate. It indicates that the neural network model can interpolate data.
>
> ---
>
> > *How should we understand the experiment in Section H? I see that $\sum_{t=1}^T F_t(w_t)$ increases as $L$ increases but $T$ is fixed for MNIST. Can we say that $\sum_{t=1}^T F_t(w_t) \sim O(\log T)$.?*
>
> **Re.** We have clarified as follows:
>
> - The dominating factor in our regret bounds is $T$ whereas $L$ can be regarded as a constant factor. Given this, the experiments could somehow complement our theoretical regret bounds since the ratio $d_2$ is of constant order in comparison to $T$.
> - Our regret bounds are $\mathcal{O}(\text{poly}(L,\log T))$, such as $\mathcal{O}(L^2\log^3 T)$ in both Theorem 1 and Theorem 3, which are increasing as $L$ increases.
>
> ---
>
> > *Assumption 4 is written as if $T$ and $\epsilon$ are independent. Can $T$ depend on $\varepsilon$?*
>
> **Re.** We have clarified as follows:
>
> - The horizon $T$ is fixed in Assumption 4. We rewrote Assumption 4 accordingly as follows:
>
> **Assumption 4. Given $T \ge 1$ and $w_1 \in \mathbb{R}^{d'}$, for any $\varepsilon >0$, there always exists $w^{(\epsilon)}$ such that $\\|w^{(\epsilon)}-w_1\\| \le g(1/\varepsilon)$ and ${1 \over T}\sum_{t=1}^T F_t(w^{(\varepsilon)}) \le \varepsilon.$**
>
> - Assumption 4 is proposed under the online learning setting. The fixed $T$ denotes both the total iteration number and the number of training points (considering the without replacement sampling scheme). Given this, $T$ and $\varepsilon$ are independent.
>
> To help better understand this point, recall the interpolating assumption under the offline setting (Chen et al., 2023, Taheri and Thrampoulidis, 2024): for any $\varepsilon > 0$, there always exists $w^{(\epsilon)}$ such that
> $$
> {1 \over n}\sum_{i=1}^n F_i(w^{(\varepsilon)}) \le \varepsilon.
> $$
> Here, $T$ is equivalent to $n$.
>
> We apologized for a misleading between online learning and the random shuffle setting in the formal experiment using the noisy XOR dataset. **We have corrected this experiment in an online learning setting.** We first generate a dataset under a noisy XOR distribution with a sample number $10^7$, and fix the total iteration number $T=10^7$. At each iteration, we randomly select a data without replacement and use Adam to update the weight. Then, we set different accuracy $\varepsilon =(\varepsilon_i)\_i =(10^{-2},10^{-3},10^{-4}, 10^{-5},10^{-6}, 10^{-7})$, and record the iteration $w_{1/\varepsilon_i}$ which is the weight at $1/\varepsilon_i$-th iteration during training. The goal is to use $w_{\varepsilon_i}$ as the approximation of $w^{(\varepsilon)}$ defined in Assumption 4. We test with $ \\|w_{\varepsilon_i} -w_1\\|$ and $\sum_{t=1}^T F_t(w_{\varepsilon_i})$. **For more details, please refer to Section H in the new revision.**
>
> ---
>
> > *At last, it would be helpful to see a curve $ \\|w^{\varepsilon} -w\\|$ for different $\varepsilon$ along with $\sum_{t=1}^T F_t(w^\varepsilon)$* .
>
> We plot the curve $\\|w^\varepsilon-w_1\\|$ and  $\varepsilon$ along with $\sum_{t=1}^T F_t(w^\varepsilon)$. Please refer to Section H in the new revision.
>
>
> At last, we acknowledge that these experiments could not fully verify but could only complement the assumptions and our regret bounds. The main contribution of this paper focuses on deriving, maybe (to our knowledge)  the first regret bound for AdamNC with constants $\beta_1 $ and $\beta_2$ in neural network optimization. We follow the interpolating assumption in some literature.

---

> ### Author Response · Authors · 2025-12-01
> **Summary of Major Weaknesses from Reviewer cAqU and Our Responses**
>
> **W1.** Is this theoretical assumption (Assumption 4) widely used in the exact form proposed in the paper?
>
> **Re.** Assumption 4 is the same as, e.g. (Chen et al., 2021; Taheri & Thrampoulidis, 2024).  The NTK-separability in Assumption 5 (which can imply Assumption 4) is the same as, e.g. (Ji & Telgarsky, 2020; Chen et al., 2021; Taheri & Thrampoulidis, 2024; Taheri et al., 2025).
>
> **W2.** Are there any experiments that support this assumption?
>
> **Re.** We added examples related to linear separable data and the noisy XOR data (Lines 1142-1168), and two simple experiments (Section H), to support Assumption 4.

---

### Official Review · Reviewer_U4Xb · 2025-11-01

**Soundness:** 3
**Presentation:** 3
**Contribution:** 3
**Rating:** 6
**Confidence:** 2

**Summary:**

This paper studies the regret analysis of AdamNC (without debiasing correction) and RMSPropNC (as a special case of AdamNC without first momentum) with a structured loss function which is constructed as $F_t(w)=f(y_t\Phi(w,x_t))$ where $\Phi$ denotes a fully connected MLP neural network for binary classification tasks with data $\\{(x_t,y_t)\\}$. The main assumptions are that $f$ is self-bounded or smooth and that the model $\Phi$ has interpolation ability such that there exists a decreasing function $g(\epsilon)$ such that any $\epsilon$ corresponds to a nearly optimal parameter $w^{(\epsilon)}$ such that $\sum F_t(w^{(\epsilon)}) / T \le \epsilon$ and $\\|w^{(\epsilon)}-w_1\\|\le g(\epsilon)$. Under these assumptions, this paper provides a convergence guarantee of AdamNC and shows that it achieves $O(g^3(\epsilon/T))$ for self-bounded or smooth $f$ when the model width is larger than certain threshold. In particular, if the model is NTK-separable and $f$ is the exponential loss or logistic loss, then $g(\epsilon)$ has an explicit form $g(\epsilon) \sim \log(1/\epsilon)$, and the previous regret bound becomes $\mathrm{polylog}(T)$. The analysis is also extended to AdamNC-Norm, where the preconditioner aggregates the norm of gradient instead of per coordinate.

**Strengths:**

The main strength of this paper lies in its novelty significance. In particular, this paper provides a novel theoretical analysis of regret bound of training neural networks with the AdamNC optimizer, which is rarely studied in any prior work. This helps to better understand the empirical effective of the popular Adam optimizer from a different perspective. Moreover, the technical results on the theoretical analysis is very concrete. It provides systematic analysis under different assumption, e.g., the loss being self-bounded or smooth, and different model structures, e.g., with and without the dimension normalization per-layer.

**Weaknesses:**

One limitation is that the convergence results in this paper requires a minimum model width to be true, and that threshold is usually asymptotically larger than the convergence rate (e.g., the width needs to be $O(g^4(\epsilon/T))$ to achieve $g^3(\epsilon/T))$ in Thm 1 and 2). This setup does not reflect the practical setting of training neural networks, where the total iteration usually has larger orders compared to the model width.

**Questions:**

- In general (without NTK-separability), is there an explicit form for $g(\epsilon)$? Could the author provides some example to help understand the shape of this function and how it's related to practical training in real life?
- Assumption 4: does it implicitly assumes $\inf f = 0$ so that $F(w)\le \epsilon$ is always achievable for any small $\epsilon$?

---

> ### Author Response · Authors · 2025-11-22
> **Responses to Weaknesses and Questions**
>
> We thank the reviewers for their valuable suggestions. We highlight the revised parts with **blue color** in the revised version.
>
> > *The convergence results in this paper require a minimum model width, and that threshold is usually asymptotically larger than the convergence rate. This setup does not reflect the practical setting of training neural networks, where the total iteration usually has larger orders compared to the model width.*
>
> **Re.** As far as we understand, the "convergence rate" in the review seems to be referred to as the regret bound instead of the average regret bound divided by $T$. The concern may lie in the case where the minimal width $g(\epsilon/T)$ is larger than $T$.
>
> We remind that in many cases, such as the NTK-separability (Assumption 5) and the linearly separable data (**please refer to Examples 1, 2, 3 in Section D**), $g(\epsilon/T)$ is a logarithmic function. Thus, the width requirement in Theorems 1 and 3 is only larger by **one $\mathcal{O}(\log T)$ factor** than the convergence rate, and both quantities are *poly-logarithmic* in the number of iterations $T$. In this case,  the model width is usually not larger than the total number of iterations.  For example, in practical scenarios where $T$ can be on the order of  $10^{5} - 10^{6}$, a width requirement of order $\mathcal{O}(\log^4(T))$ (Theorem 1) or $\mathcal{O}(\log^L(T))$ (Theorem 3), is mild and easily satisfied by typical neural network widths.
>
> In the smooth case (Theorem 2), there is an additional $\mathcal{O}(T)$ factor inside the minimal width, which could cause the width to be larger than the total iteration. We suspect that the reason may come from the rather weaker smooth loss assumption and the technical artifact of our proof. For example, the upper bound of regret is determined by $\mathcal{O}(\sqrt{F(w)})$ due to the weakly-convex-like parameter and smooth-like parameters in Lemmas 4 and 5. When using Young's inequality to derive $\mathcal{O}(F(w))$, it is required that $m \ge \mathcal{O}(T)$ to control the additional tail term, as shown in Lemma 5. However, for several commonly used loss functions, including the logistic loss and exponential loss, both Theorem 1 and Theorem 3 hold where the width requirements are of order $\mathcal{O}(\log T)$ under NTK-separability.
>
> ---
>
> > *In general (without NTK-separability), is there an explicit form for $g(\epsilon)$? Could the author provides some example to help understand the shape of this function and how it's related to practical training in real life?*
>
> **Re. We have added some more examples where $g(\epsilon/T)$ is a logarithm function, such as Example 1  (linearly separable data), Examples 2 and 3 (the noisy XOR data) in Section D. Please refer to the new revision.**
>
> **We also add some simple experiments in Section H to show that a $\mathcal{O}(\log T)$ order width is enough to achieve the $\mathcal{O}(\log T)$ regret bound, which may help to address the reviewer's concern.**
>
> ---
>
> > *Assumption 4: does it implicitly assumes $\inf f = 0$ so that $F(w)<\epsilon$ is always achievable for any small $\epsilon$?*
>
> **Re.** The implicit assumption of $\inf f = 0$  lies in **Line 110-111**, where the range of $f$ is positive. The positive loss functions with $\inf f=0$ are common, including $\ell_2$-loss, logistic loss, and exponential loss. Since $f$ is positive, the objective function $F(w) =f(y\Phi(w,x))$ is also non-negative.
>
> In Assumption 4, we assume for any $\epsilon > 0$, there exists $w\^{\*}$ such that $\sum\_{t=1}^T F\_t(w\^*)/T < \epsilon$.

---

> ### Author Response · Authors · 2025-11-28
> **correction on experiment related to noisy XOR data**
>
> Dear Reviewer U4Xb,
>
> We apologized for a misleading between online learning and the random shuffle setting in the experiment using the noisy XOR dataset.
>
> **We have corrected this experiment in an online learning setting.** We first generate a dataset under a noisy XOR distribution with a sample number $10^7$, and set the total iteration number $T=10^7$. At each iteration, we randomly select a data without replacement and use Adam to update the weight. Then, we set different accuracy $\varepsilon =(\varepsilon_i)\_i =(10^{-2},10^{-3},10^{-4}, 10^{-5},10^{-6}, 10^{-7})$, and record the iteration $w_{1/\varepsilon_i}$ which is the weight at $1/\varepsilon_i$-th iteration during training. The goal is to use $w_{\varepsilon_i}$ as the approximation of $w^{(\varepsilon)}$ defined in Assumption 4. We test with $ \\|w_{\varepsilon_i} -w_1\\|$ and $\sum_{t=1}^T F_t(w_{\varepsilon_i})$. We also plot the curve $\\|w^\varepsilon-w_1\\|$ and  $\varepsilon$ along with $\sum_{t=1}^T F_t(w^\varepsilon)$.
> **For more details, please refer to Section H in the new revision.**
>
> We truly appreciate your valuable time and comments. We would greatly appreciate any additional feedback or clarifications you can offer. Thank you for your time and consideration.

---

> ### Author Response · Authors · 2025-12-01
> **Summary of Major Weaknesses from Reviewer U4Xb and Our Responses**
>
> **W1.** Concern about the minimal width (being larger than the total number of iterations $T$).
>
> **Re.** Under NTK-separability, the required minimal width scales polylogarithmically with $T$, and can therefore be much smaller than $T$. To complement this, we added examples related to linear separable data and the noisy XOR data (Lines 1142 to 1168), and two simple experiments (Section H).

---

### Official Review · Reviewer_jdbY · 2025-11-03

**Soundness:** 3
**Presentation:** 3
**Contribution:** 2
**Rating:** 4
**Confidence:** 3

**Summary:**

The paper studies the convergence of adaptive methods (RMSProp and AdamNC) in deep fully connected networks for online binary classification with smooth activation functions. The authors prove an $O(polylog(T))$ regret bound for sufficiently wide networks in the NTK regime. This is comparable to rate for strongly convex online optimization. They also analyze scalar variants (RMSProp-Norm and AdamNC-Norm) that achieve similar bounds without requiring prior knowledge of problem constants. The proof builds on the idea of Bregman Proximal Gradient and applies it within the NTK framework.

**Strengths:**

-	Analyzing the convergence of adaptive methods is an important research question.
-	The paper is overall clearly written and includes a helpful proof sketch to illustrate the main idea.
-	The paper establishes the convergence of adaptive methods for deep networks in the NTK regime, which appears to be new.

**Weaknesses:**

-	The paper focuses on the NTK regime, where neural networks are known to behave similarly to linear or kernel methods. However, this setting does not always reflect the behavior of practical networks.
-	The main technique appears similar to those in Duchi et al. (2011) and Alacaoglu et al. (2020) for handling adaptive methods, and is applied here to the specific NTK setting. It would be helpful to clarify whether any new challenges arise in this context or if additional techniques were required to address them.

**Questions:**

- For Theorems 1 and 2, I wonder why the stepsize $\eta$ must have the exact order specified in the statement, rather than simply being any sufficiently small value. What is the intuition behind this requirement?
- Can Assumption 4, which assumes the existence of such a function $g(\epsilon)$, hold in more interesting regimes beyond the NTK setting (Assumption 5)?
- Do the results provide any insight into the potential advantages of using adaptive methods over vanilla gradient descent?

---

> ### Author Response · Authors · 2025-11-22
> **Responses to Weaknesses**
>
> We thank the reviewers for their valuable suggestions. We highlight the revised parts with **blue color** in the revised version.
>
> > *The paper focuses on the NTK regime, where neural networks are known to behave similarly to linear or kernel methods. However, this setting does not always reflect the behavior of practical networks.*
>
> **Re.** We agree that the NTK regime may not represent all aspects of practical deep networks. We have added the statement in the **Limitation** part:
>
> **The results rely on the NTK regime. Extending our results beyond this setting is left as a future problem.
> Also, it may be beneficial to provide more experiments in practical cases to verify the realizable assumptions and regret bounds.**
>
> We also note that the NTK-based analysis may still be common, particularly for theoretical understanding of neural network training, such as (Ji & Telgarsky, 2020; Chen et al., 2021; Taheri & Thrampoulidis, 2024).   Following this line, the main contribution of this paper is to derive theoretical regret bounds (which may be the first ones to our knowledge) of Adam-type methods for training deep neural networks. We believe that such progress may be valuable in further deriving the regret bound of Adam in other types of neural networks under more general settings.
>
> To complement the potential limitation of NTK-based analysis, we also further consider the weaker interpolating realizable setup (Assumption 4), which may be more practical in the behavior of neural networks.
>
> **We also add some experiments in Section H to complement the interpolating assumption and our regret bounds.**
>
> ---
>
> > *The main technique appears similar to those in Duchi et al. (2011) and Alacaoglu et al. (2020) for handling adaptive methods, and is applied here to the specific NTK setting. It would be helpful to clarify whether any new challenges arise in this context or if additional techniques were required to address them.*
>
> **Re.** We borrow some technical results from adaptive methods in standard online-convex optimization, including (Duchi et al., 2011; Alacaoglu et al., 2020), as we indicated before Proposition 2. In addition, some of the major differences lie in the following three parts:
>
> - **Neural networks optimization.**
>
> The objective function of training a neural network is potentially non-smooth and non-convex. We derive that the objective function exhibits some unique weakly-convex-like and smooth-like properties, such as in Lemmas 3 to 5. In comparison, works like (Duchi et al., 2011; Alacaoglu et al., 2020) considered a more general convex objective function.
>
> - **The absence of bounded gradient and bounded iterate assumptions.**
>
> Most existing works for adaptive methods in online-convex optimization rely on the bounded gradient and iteration assumptions, including (Duchi et al., 2011; Alacaoglu et al., 2020). In comparison, we do not assume either of them. We rely on the more benign bounds for gradient and Hessian norm of neural networks, such as Lemmas 3, 4, and Lemma 15, as well as other related lemmas, and an induction argument, to derive that the iteration remains in a bounded region. Further, combining a proper setup of the width, we can get the regret bounds without requiring bounded gradients.
>
> - **The non-monotonic step-sizes.**
>
> Duchi et al., (2011) and Alacaoglu et al. (2020) rely on the non-increasing step-sizes of AdaGrad/AMSGrad or AdamNC (using $\beta_{2t}=1-1/t$) to estimate the term $\sum\_{s=1}^t (B\_{\psi\_s}(w\_s,w)-B\_{\psi\_s}(w\_{s+1},w))$ in Proposition 2. In comparison, the step-sizes of RMSProp and AdamNC are non-monotonic when $\beta\_1,\beta\_2$ and $C\_0\eta$ are constants. We use new bounds in Lemma 7 and Lemma 18, as well as other related lemmas, to control $\sum\_{s=1}^t (B\_{\psi\_s}(w\_s,w)-B\_{\psi\_s}(w\_{s+1},w))$, relying on the exponential moving average of RMSProp and AdamNC.
>
> **We have added more discussions in Section 5 and Section B.3 to clarify in detail the new challenges and solutions that arise when analyzing Adam compared to GD and other adaptive methods in neural network optimization or online convex optimization.**

---

> ### Author Response · Authors · 2025-11-22
> **Responses to Questions**
>
> > *For Theorems 1 and 2, I wonder why the stepsize must have the exact order specified in the statement, rather than simply being any sufficiently small value. What is the intuition behind this requirement?*
>
> **Re. We have clarified these requirements in the revised version.**
> - In Theorem 1, $C_0\eta \le \mathcal{O}({1 \over g(\epsilon /T)L^2})$;
> - In Theorem 2, $\tilde{C}_0 \le \mathcal{O}({1 \over g(\epsilon_0 /T)L^{1.5}})$.
>
> These requirements can be reduced to $\mathcal{O}({1 \over \log T})$ order under NTK-separability. Also, these requirements may indicate that the learning rate should be moderately small to ensure the regret bound. We also notice that Theorems 3 and 4 allow for parameter-free learning rates due to the adaptive step-sizes of AdamNC (compared to constant step-sizes of GD).
>
> ---
>
> > *Can Assumption 4, which assumes the existence of such a function $g(\epsilon)$, hold in more interesting regimes beyond the NTK setting (Assumption 5)?*
>
> **Re. We have added some more examples where $g(\epsilon)$ is a logarithm function $\log(1/\epsilon)$, such as Example 1  (linearly separable data), Examples 2 and 3 (the noisy XOR data distribution) in Section D. Please refer to the new revision.**
>
> **We also add some simple experiments in Section H where we train deep fully connected neural networks with Adam on noisy XOR data distribution and the MNIST dataset. We test the interpolating assumptions and regret bounds in these experiments, which help to complement our realizable assumptions and regret bounds.**
>
> ---
>
> > *Do the results provide any insight into the potential advantages of using adaptive methods over vanilla gradient descent?*
>
> **Re.** The potential advantages mainly lie in the following two points,
>
> - **Parameter-free learning rates.** Theorems 3 and 4 only require learning rates $0< C_0\eta \le 1$, whereas several regret bounds for GD in neural network optimization require the learning rates to be determined by problem-dependent parameters, such as (Chen et al., 2021; Taheri and Thrampoulidis, 2024; Taheri et al., 2025).
> - **Implicitly better regret bounds when gradients are sparse.** We notice that in the proof of Lemmas 7, 8, 18 and 19, the upper bounds are $\mathcal{O}(\sum\_{i=1}^{d'}\sqrt{(1-\beta\_2)\sum\_{s=1}^t \beta\_2^{t-s} g\_{s,i}^2})$. When the gradient $g_s$ is sparse, this term is smaller than $\mathcal{O}(\sqrt{T})$ obtained by GD. The advantage is also observed in online-convex optimization for AdaGrad (Duchi et al., 2011) or Adam (Reddi et al., 2019).
>
> **We also add more discussions to clarify the potential advantages of RMSProp and AdamNC over GD in Section 3 and Remark 4 in Section B.**

---

> ### Author Response · Authors · 2025-11-28
> **We truly appreciate your valuable time and consideration**
>
> Dear Reviewer jdbY,
>
> We would like to respectfully follow up regarding our rebuttal. We understand the significant workload during the review period and appreciate the time you devote to evaluating submissions. If you have an opportunity, we would be grateful for any further comments or clarifications. Your feedback is highly valued. Thank you for your time and consideration.
>
> The newly update revision including:
>
> - a presentation revision on Assumption 4 to make it clearer.
> - a correction on the experiment related to the noisy XOR data, please refer to Section H.
>
> Warm regards,
>
> Authors

---

> ### Author Response · Authors · 2025-12-01
> **Summary of Major Weaknesses from Reviewer jdbY and Our Responses**
>
> **W1.** The paper focuses on the NTK regime (which does not always reflect the behavior of practical networks).
>
> **Re.**  The  NTK-regime is used in, e.g., (Ji & Telgarsky, 2020; Chen et al., 2021; Taheri & Thrampoulidis, 2024, Taheri et al., 2025). AdamNC with constant $\beta_1$ and $\beta_2$ could lead to diverge in online convex optimization. We believe that the analysis for AdamNC in nonconvex neural network optimization is not easy even in the NTK regime.  We provide such an analysis.
>
>
> **W2.** clarify whether any new challenges
>
> **Re.**   Proof challenges:  non-monotonic step-sizes of AdamNC (with constants $\beta_1$ and $\beta_2$ that could lead to divergence in online convex optimization); unbounded gradient and iterate norms, etc. See (Lines 420–431 and Lines 918–952) in the revision.

---

### Author Response · Authors · 2025-12-01
**Final Remark**

We would like to sincerely thank the Area Chair and all reviewers for their thoughtful feedback and efforts in handling our manuscript.

Our work provides, to the best of our knowledge, the first regret bounds for RMSProp and AdamNC in training deep neural networks with constant $\beta_1$ and $\beta_2$, under the interpolating data assumption or NTK-separability. We believe that the analysis is challenging, without relying on bounded gradients and iteration in comparison to classical online convex optimization (where AdamNC with constant $\beta_1$ and $\beta_2$ could diverge in online convex optimization). It's also essentially different from Gradient Descent in neural network optimization due to the adaptive non-monotonic step-sizes of AdamNC.

We carefully considered and addressed the raised concerns (that are mostly about the presentation clarity and the NTK-separability assumption) during the Rebuttal and Discussion periods.  We made some changes accordingly (marked with blue) in the revision.

**Reviewer Csr8** raises seemly concerns about the potential limitation of NTK-based analysis and the similarity of adaptive methods in online learning. In response,  we clarified the NTK-regime is used in, e.g., (Ji & Telgarsky, 2020; Chen et al., 2021; Taheri & Thrampoulidis, 2024, Taheri et al., 2025). We believe that our analysis for AdamNC with constant $\beta_1$ and $\beta_2$ (that could lead to diverge in online convex optimization) in nonconvex neural network optimization is not easy even in the NTK regime. We added paragraphs (Lines 420–431 and Lines 918–952) to clarify the major proof challenging  and new techniques in the revision.

**Reviewer cAqU** notes that the minimal width in regret bounds may be larger than the iteration number $T$. In response, we indicate that the main results only require a polylogarithm order width with respect to $T$ (Remarks 1-2). To complement this, we added some examples (Lines 1142 to 1168) and simple experiments (Section H).

**Reviewer U4Xb** raises concerns about the interpolating assumption (Assumption 4). In response, we rewrote the assumption to make it clearer. We clarified that Assumption 4 is relatively common in the literature. We provided some examples (Lines 1142 to 1168 ) and simple experiments (Section H) to further support this assumption.

**Reviewer jdbY** mainly raises concerns about the lack of discussions of distinctions of regret bounds and the proof novelty of AdamNC over GD. In response, we added more discussions on our regret bounds and existing ones, and the potential advantages of AdamNC over GD in our bounds, including the parameter-free learning rates and better order bounds when gradients are sparse (Lines 267 to 276). We also added more details of the proof distinctions (Lines 420–431and Lines 918–952).

---

### Meta-Review · Area_Chair_yKK2 · 2026-01-07

**Summary:**

This work studies the theoretical analysis of RMSProp and Adam without corrective terms, for deep networks and smooth activations. One reviewer suggested that the NTK regime which the paper is based on does not reflect practical networks' behavior. Also, some reviewers questioned the novelty of the analysis, as there have already been many convergence analyses for momentum-based methods, and the analysis seems to rely on the known methods.

**Reviewer Concerns:**

The reviewers share the concerns regarding the practicality and novelty of the results. While the authors provided some answers, the justifications are not very convincing.

**Reviewer Scores:**

Most reviewers did not reply to the rebuttal, and there are no major misunderstandings of the paper. Thus, I do not expect the reviewers' scores would change much after the rebuttal.

---

### Decision · Program_Chairs · 2026-01-26

Reject